# Doubly Adaptive Scaled Algorithm for Machine Learning Using $2^{nd}$ Order Information

**Majid Jahani**[1], **Sergey Rusakov**[1], **Zheng Shi**[1], **Peter Richtárik**[2], **Michael W. Mahoney**[3],
**Martin Takáč**[4,1]
[1]Lehigh University [2]KAUST [3]University of California, Berkeley, USA
[4]Mohamed bin Zayed University of Artificial Intelligence (MBZUAI)
majidjahani89@gmail.com, ser318@lehigh.edu,
shi.zheng.tfls@gmail.com, peter.richtarik@kaust.edu.sa,
mmahoney@stat.berkeley.edu, takac.MT@gmail.com

## Abstract

We present a novel adaptive optimization algorithm for large-scale machine learning problems. Equipped with a low-cost estimate of local curvature and Lipschitz smoothness, our method dynamically adapts the search direction and step-size. The search direction contains gradient information preconditioned by a well-scaled diagonal preconditioning matrix that captures the local curvature information. Our methodology does not require the tedious task of learning rate tuning, as the learning rate is updated automatically without adding an extra hyperparameter. We provide convergence guarantees on a comprehensive collection of optimization problems, including convex, strongly convex, and nonconvex problems, in both deterministic and stochastic regimes. We also conduct an extensive empirical evaluation on standard machine learning problems, justifying our algorithm's versatility and demonstrating its strong performance compared to other start-of-the-art first-order and second-order methods.

## 1 Introduction

This paper presents an algorithm for solving empirical risk minimization problems of the form:

$$\min_{w\in\mathbb{R}^d} F(w) := \tfrac{1}{n}\sum_{i=1}^n f(w; x^i, y^i) = \tfrac{1}{n}\sum_{i=1}^n f_i(w), \tag{1}$$

where $w$ is the model parameter/weight vector, $\{(x_i, y_i)\}_{i=1}^n$ are the training samples, and $f_i : \mathbb{R}^d \to \mathbb{R}$ is the loss function. Usually, the number of training samples, $n$, and dimension, $d$, are large, and the loss function $F$ is potentially nonconvex, making (1) difficult to solve.

In the past decades, significant effort has been devoted to developing optimization algorithms for machine learning. Due to easy implementation and low per-iteration cost, (stochastic) first-order methods (Robbins & Monro, 1951; Duchi et al., 2011; Schmidt et al., 2017; Johnson & Zhang, 2013; Nguyen et al., 2017; 2019; Kingma & Ba, 2014; Jahani et al., 2021a; Recht et al., 2011) have become prevalent approaches for many machine learning applications. However, these methods have several drawbacks: ($i$) they are highly sensitive to the choices of hyperparameters, especially learning rate; ($ii$) they suffer from ill-conditioning that often arises in large-scale machine learning; and ($iii$) they offer limited opportunities in distributed computing environments since these methods usually spend more time on "communication" instead of the true "computation." The main reasons for the aforementioned issues come from the fact that first-order methods only use the gradient information for their updates.

On the other hand, going beyond first-order methods, Newton-type and quasi-Newton methods (Nocedal & Wright, 2006; Dennis & Moré, 1977; Fletcher, 1987) are considered to be a strong family of optimizers due to their judicious use of the curvature information in order to scale the gradient. By exploiting the curvature information of the objective function, these methods mitigate many of the issues inherent in first-order methods. In the deterministic regime, it is known that these methods are relatively insensitive to the choices of the hyperparameters, and they handle ill-conditioned problems with a fast convergence rate. Clearly, this does not come for free, and these methods can have memory requirements up to $\mathcal{O}(d^2)$ with computational

complexity up to $\mathcal{O}(d^3)$ (e.g., with a naive use of the Newton method). There are, of course, efficient ways to solve the Newton system with significantly lower costs (e.g., see Nocedal & Wright (2006)). Moreover, quasi-Newton methods require lower memory and computational complexities than Newton-type methods. Recently, there has been shifted attention towards stochastic second-order (Roosta-Khorasani & Mahoney, 2018; Byrd et al., 2011; Martens, 2010; Jahani et al., 2020a; Xu et al., 2017; Roosta et al., 2018; Yao et al., 2018) and quasi-Newton methods (Curtis, 2016; Berahas et al., 2016; Mokhtari & Ribeiro, 2015; Jahani et al., 2021b; Berahas et al., 2019; Jahani et al., 2020b) in order to approximately capture the local curvature information.

These methods have shown good results for several machine learning tasks (Xu et al., 2020; Berahas et al., 2020; Yao et al., 2019). In some cases, however, due to the noise in the Hessian approximation, their performance is still on par with the first-order variants. One avenue for reducing the computational and memory requirements for capturing curvature information is to consider just the diagonal of the Hessian. Since the Hessian diagonal can be represented as a vector, it is affordable to store its moving average, which is useful for reducing the impact of noise in the stochastic regime. To exemplify this, AdaHessian Algorithm (Yao et al., 2020) uses Hutchinson's method (Bekas et al., 2007) to approximate the Hessian diagonal, and it uses a second moment of the Hessian diagonal approximation for preconditioning the gradient. AdaHessian achieves impressive results on a wide range of state-of-the-art tasks. However, its preconditioning matrix approximates the Hessian diagonal only very approximately, suggesting that improvements are possible if one can better approximate the Hessian diagonal.

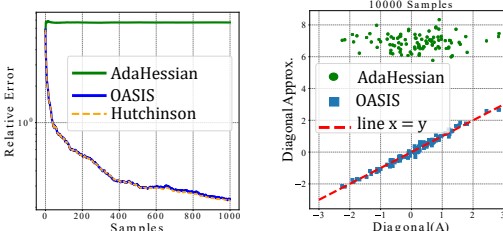

Figure 1: Comparison of the diagonal approximation by AdaHessian and `OASIS` over a random symmetric matrix $A$ ($100 \times 100$). **left:** Relative error (in Euclidean norm) between the true diagonal of matrix $A$ and the diagonal approximation by AdaHessian, Hutchinson's method, and `OASIS` (the $x$-axis shows the number of random vectors sampled from the Rademacher distribution, see Section 2. Moreover, this plot can be considered as the representation of the error of the Hessian diagonal approximation's evolution over the iterations of minimizing $w^T A w$, since $A$ is fixed and symmetric.); **right:** Diagonal approximation scale for AdaHessian and `OASIS` ($y$-axis), in comparison to the true diagonal of matrix $A$ ($x$-axis).

In this paper, we propose the d**O**ubly **A**daptive **S**caled algor**I**thm for machine learning using **S**econd-order information (`OASIS`). `OASIS` approximates the Hessian diagonal in an efficient way, providing an estimate whose scale much more closely approximates the scale of the true Hessian diagonal (see Figure 1). Due to this improved scaling, the search direction in `OASIS` contains gradient information, in which the components are well-scaled by the novel preconditioning matrix. Therefore, every gradient component in each dimension is adaptively scaled based on the approximated curvature for that dimension. For this reason, there is no need to tune the learning rate, as it would be updated automatically based on a local approximation of the Lipschitz smoothness parameter (see Figure 2). The well-scaled preconditioning matrix coupled with the adaptive learning rate results in a fully adaptive step for updating the parameters. Here, we provide a brief summary of our main contributions:

- *Novel Optimization Algorithm.* We propose `OASIS` as a fully adaptive method that preconditions the gradient information by a well-scaled Hessian diagonal approximation. The gradient component in each dimension is adaptively scaled by the corresponding curvature approximation.
- *Adaptive Learning Rate.* Our methodology does not require us to tune the learning rate, as it is updated automatically via an adaptive rule. The rule approximates the Lipschitz smoothness parameter, and it updates the learning rate accordingly.
- *Comprehensive Theoretical Analysis.* We derive convergence guarantees for `OASIS` with respect to different settings of learning rates, namely the case with adaptive learning rate for convex and strongly convex cases. We also provide the convergence guarantees with respect to fixed learning rate and line search for both strongly convex and nonconvex settings.
- *Competitive Numerical Results.* We investigate the empirical performance of `OASIS` on a variety of standard machine learning tasks, including logistic regression, nonlinear least squares problems, and image classification. Our proposed method consistently shows competitive or superior performance in comparison to many first- and second-order state-of-the-art methods.

**Notation.** By considering the positive definite matrix $D$, we define the weighted Euclidean norm of vector $x \in \mathbb{R}^d$ with $\|x\|_D^2 = x^T D x$. Its corresponding dual norm is shown as $\| \cdot \|_D^*$. The operator

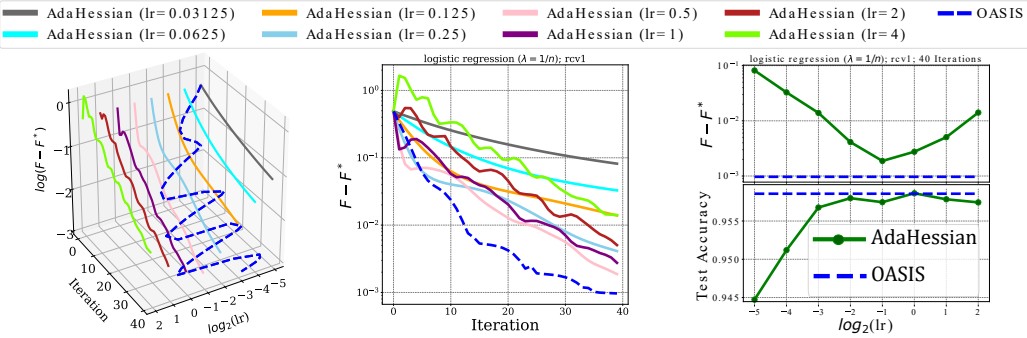

Figure 2: Adaptive Learning Rate (`Logistic Regression` with strong-convexity parameter $\frac{1}{n}$ over `rcv1` dataset). **left and middle:** Comparison of optimality gap for AdaHessian Algorithm with multiple learning-rate choices vs. `OASIS` Algorithm with adaptive learning rate (dashed-blue line); **right:** Comparison of the best optimality gap and test accuracy for AdaHessian Algorithm w.r.t. each learning rate shown on $x$-axis after 40 iterations vs. the optimality gap and test accuracy for our `OASIS` Algorithm with adaptive learning rate after 40 iteration (dashed-blue line).

$\odot$ is used as a component-wise product between two vectors. Given a vector $v$, we represent the corresponding diagonal matrix of $v$ with `diag`($v$).

## 2 RELATED WORK

In this paper, we analyze algorithms with the generic iterate updates:
$$w_{k+1} = w_k - \eta_k \hat{D}_k^{-1} m_k, \tag{2}$$
where $\hat{D}_k$ is the preconditioning matrix, $m_k$ is either $g_k$ (the true gradient or the gradient approximation) or the first moment of the gradient with momentum parameter $\beta_1$ or the bias corrected first moment of the gradient, and $\eta_k$ is the learning rate. The simple interpretation is that, in order to update the iterates, the vector $m_k$ would be rotated and scaled by the inverse of preconditioning matrix $\hat{D}_k$, and the transformed information would be considered as the search direction. Due to limited space, here we consider only some of the related studies with a diagonal preconditioner. For more general preconditioning, see Nocedal & Wright (2006). Clearly, one of the benefits of a well-defined diagonal preconditioner is the easy calculation of its inverse.

There are many optimization algorithms that follow the update in (2). A well-known method is stochastic gradient descent (SGD).The idea behind SGD is simple yet effective: the preconditioning matrix is set to be $\hat{D}_k = I_d$, for all $k \geq 0$. There are variants of SGD with and without momentum. The advantage of using momentum is to smooth the gradient (approximation) over the past iterations, and it can be useful in the noisy settings. In order to converge to the stationary point(s), the learning rate in SGD needs to decay. Therefore, there are many important hyperparameters that need to be tuned, e.g., learning rate, learning-rate decay, batch size, and momentum. Among all of them, tuning the learning rate is particularly important and cumbersome since the learning rate in SGD is considered to be the same for all dimensions. To address this issue, one idea is to use an adaptive diagonal preconditioning matrix, where its elements are based on the local information of the iterates. One of the initial methods with a non-identity preconditioning matrix is Adagrad (Duchi et al., 2011; McMahan & Streeter, 2010). In Adagrad, the momentum parameter is set to be zero ($m_k = g_k$), and the preconditioning matrix is defined as:
$$\hat{D}_k = \mathtt{diag}\left(\sqrt{\sum_{i=1}^{k} g_k \odot g_k}\right). \tag{3}$$

In the preconditioning matrix $\hat{D}_k$ in (3), every gradient component is scaled with the accumulated information of all the past squared gradients. It is advantageous in the sense that every component is scaled adaptively. However, a significant drawback of $\hat{D}_k$ in (3) has to do with the progressive increase of its elements, which leads to rapid decrease of the learning rate. To prevent Adagrad's aggressive, monotonically decreasing learning rate, several approaches, including Adadelta (Zeiler, 2012) and RMSProp (Tieleman & Hinton, 2012), have been developed. Specifically, in RMSProp, the momentum parameter $\beta_1$ is zero (or $m_k = g_k$) and the preconditioning matrix is as follows:
$$\hat{D}_k = \sqrt{\beta_2 \hat{D}_{k-1}^2 + (1 - \beta_2)\mathtt{diag}(g_k \odot g_k)}, \tag{4}$$
where $\beta_2$ is the momentum parameter used in the preconditioning matrix. As we can see from the difference between the preconditioning matrices in (3) and (4), in RMSProp an exponentially decaying average of squared gradients is used, which prevents rapid increase of preconditioning

components in (3).

Another approach for computing the adaptive scaling for each parameter is Adam (Kingma & Ba, 2014). Besides storing an exponentially decaying average of past squared gradients like Adadelta and RMSprop, Adam also keeps first moment estimate of gradient, similar to SGD with momentum. In Adam, the bias-corrected first and second moment estimates, i.e., $m_k$ and $\hat{D}_k$ in (2), are as follows:

$$m_k = \frac{1-\beta_1}{1-\beta_1^k}\sum_{i=1}^{k}\beta_1^{k-i}g_i, \qquad \hat{D}_k = \sqrt{\frac{1-\beta_2}{1-\beta_2^k}\sum_{i=1}^{k}\beta_2^{k-i}\texttt{diag}(g_i \odot g_i)}. \qquad (5)$$

There have been many other first-order methods with adaptive scaling (Loshchilov & Hutter, 2017; Chaudhari et al., 2019; Loshchilov & Hutter, 2016; Shazeer & Stern, 2018).

The methods described so far have only used the information of the gradient for preconditioning $m_k$ in (2). The main difference of second-order methods is to employ higher order information for scaling and rotating the $m_k$ in (2). To be precise, besides the gradient information, the (approximated) curvature information of the objective function is also used. As a textbook example, in Newton's method $\hat{D}_k = \nabla^2 F(w_k)$ and $m_k = g_k$ with $\eta_k = 1$.

**Diagonal Approx.** Recently, using methods from randomized numerical linear algebra, the AdaHessian method was developed (Yao et al., 2020). AdaHessian approximates the diagonal of the Hessian, and it uses the second moment of the diagonal Hessian approximation as the preconditioner $\hat{D}_k$ in (2). In AdaHessian, Hutchinson's method[1] is used to approximate the Hessian diagonal as follows:

$$D_k \approx \texttt{diag}(\mathbb{E}[z_k \odot \nabla^2 F(w_k)z_k]), \qquad (6)$$

where $z_k$ is a random vector with Rademacher distribution. Needless to say,[2] the oracle $\nabla^2 F(w_k)z_k$, or Hessian-vector product, can be efficiently calculated; in particular, for AdaHessian, it is computed with two back-propagation rounds without constructing the Hessian explicitly. In a nutshell, the first momentum for AdaHessian is the same as (5), and its second order momentum is:

$$\hat{D}_k = \sqrt{\frac{1-\beta_2}{1-\beta_2^k}\sum_{i=1}^{k}\beta_2^{k-i}D_i^2}. \qquad (7)$$

The intuition behind AdaHessian is to have a larger step size for the dimensions with shallow loss surfaces and smaller step size for the dimensions with sharp loss surfaces. The results provided by AdaHessian show its strength by using curvature information, in comparison to other adaptive first-order methods, for a range of state-of-the-art problems in computer vision, natural language processing, and recommendation systems (Yao et al., 2020). However, even for AdaHessian, the preconditioning matrix $\hat{D}_k$ in (7) does not approximate the scale of the actual diagonal of the Hessian particularly well (see Figure 1). One might hope that a better-scaled preconditioner would enable better use of curvature information. This is one of the main focuses of this study.

**Adaptive Learning Rate.** In all of the methods discussed previously, the learning rate $\eta_k$ in (2) is still a hyperparameter which needs to be manually tuned, and it is a critical and sensitive hyperparameter. It is also necessary to tune the learning rate in methods that use approximation of curvature information (such as quasi-Newton methods like BFGS/LBFGS, and methods using diagonal Hessian approximation like AdaHessian). The studies (Loizou et al., 2020; Vaswani et al., 2019; Chandra et al., 2019; Baydin et al., 2017; Malitsky & Mishchenko, 2020) have tackled the issue regarding tuning learning rate, and have developed methodologies with adaptive learning rate, $\eta_k$, for first-order methods. Specifically, the work (Malitsky & Mishchenko, 2020) finds the learning rate by approximating the Lipschitz smoothness parameter in an affordable way without adding a tunable hyperparameter which is used for GD-type methods (with identity norm). Extending the latter approach to the weighted-Euclidean norm is not straightforward. In the next section, we describe how we can extend the work (Malitsky & Mishchenko, 2020) for the case with weighted Euclidean norm. This is another main focus of this study. In fact, while we focus on AdaHessian, any method with a positive-definite preconditioning matrix and bounded eigenvalues can benefit from our approach.

## 3   OASIS

In this section, we present our proposed methodology. First, we focus on the deterministic regime, and then we describe the stochastic variant of our method.

---

[1]For a general symmetric matrix A, $\mathbb{E}[z \odot Az]$ equals the diagonal of $A$ (Bekas et al., 2007).

[2]Actually, it needs to be said: many within the machine learning community still maintain the incorrect belief that extracting second order information "requires inverting a matrix." It does not.

## 3.1 DETERMINISTIC OASIS

Similar to the methods described in the previous section, our OASIS Algorithm generates iterates according to (2).

Motivated by AdaHessian, and by the fact that the loss surface curvature is different across different dimensions, we use the curvature information for preconditioning the gradient. We now describe how the preconditioning matrix $\hat{D}_k$ can be adaptively updated at each iteration as well as how to update the learning rate $\eta_k$ automatically for performing the step. To capture the curvature information, we also use Hutchinson's method and update the diagonal approximation as follows:

---

**Algorithm 1** OASIS

**Input:** $w_0, \eta_0, D_0, \theta_0 = +\infty$

1: $w_1 = w_0 - \eta_0 \hat{D}_0^{-1} \nabla F(w_0)$
2: **for** $k = 1, 2, \dots$ **do**
3:     Form $D_k$ via (8) and $\hat{D}_k$ via (9)
4:     Update $\eta_k$ based on (10)
5:     Set $w_{k+1} = w_k - \eta_k \hat{D}_k^{-1} \nabla F(w_k)$
6:     Set $\theta_k = \frac{\eta_k}{\eta_{k-1}}$
7: **end for**

---

$$D_k = \beta_2 D_{k-1} + (1 - \beta_2) \, \mathtt{diag}(v_k), \quad \text{where } v_k := z_k \odot \nabla^2 F(w_k) z_k. \tag{8}$$

Before we proceed, we make a few more comments about the Hessian diagonal $D_k$ in (8). As is clear from (8), a decaying exponential average of Hessian diagonal is used, which can be very useful in the noisy settings for smoothing out the Hessian noise over iterations. Moreover, it approximates the scale of the Hessian diagonal with a satisfactory precision, unlike AdaHessian Algorithm (see Figure 1). More importantly, the modification is simple yet very effective. Similar simple and efficient modification happens in the evolution of adaptive first-order methods (see Section 2). Further, motivated by (Paternain et al., 2019; Jahani et al., 2021b), in order to find a well-defined preconditioning matrix $\hat{D}_k$, we truncate the elements of $D_k$ by a positive truncation value $\alpha$. To be more precise:

$$(\hat{D}_k)_{i,i} = \max\{|D_k|_{i,i}, \alpha\}, \quad \forall i \in [d]. \tag{9}$$

The goal of the truncation described above is to have a well-defined preconditioning matrix that results in a descent search direction; note the parameter $\alpha$ is equivalent to $\epsilon$ in Adam and AdaHessian). Next, we discuss the adaptive strategy for updating the learning rate $\eta_k$ in (2). By extending the adaptive rule in (Malitsky & Mishchenko, 2020) and by defining $\theta_k := \frac{\eta_k}{\eta_{k-1}}, \; \forall k \geq 1$, our learning rate needs to satisfy the inequalities: i) $\eta_k^2 \leq (1 + \theta_{k-1}) \eta_{k-1}^2$, and ii) $\eta_k \leq \frac{\|w_k - w_{k-1}\|_{\hat{D}_k}}{2\|\nabla F(w_k) - \nabla F(w_{k-1})\|_{\hat{D}_k}^*}$. (These inequalities come from the theoretical results.) Thus, the learning rate can be adaptively calculated as follows:

$$\eta_k = \min\{\sqrt{1 + \theta_{k-1}} \, \eta_{k-1}, \, \|w_k - w_{k-1}\|_{\hat{D}_k} / (2\|\nabla F(w_k) - \nabla F(w_{k-1})\|_{\hat{D}_k}^*)\}. \tag{10}$$

As is clear from (10), it is only required to store the previous iterate with its corresponding gradient and the previous learning rate (a scalar) in order to update the learning rate. Moreover, the learning rate in (10) is controlled by the gradient and curvature information. As we will see later in the theoretical results, $\eta_k \geq \frac{\alpha}{2L}$ where $L$ is the Lipschitz smoothness parameter of the loss function. It is noteworthy to highlight that due to usage of weighted-Euclidean norms the required analysis for the cases with adaptive learning rate is non-trivial (see Section 4). Also, by setting $\beta_2 = 1$, $\alpha = 1$, and $D_0 = I_d$, our OASIS algorithm covers the algorithm in (Malitsky & Mishchenko, 2020).

## 3.2 STOCHASTIC OASIS

In every iteration of OASIS, as presented in the previous section, it is required to evaluate the gradient and Hessian-vector product on the whole training dataset. However, these computations are prohibitive in the large-scale setting, i.e., when $n$ and $d$ are large. To address this issue, we present a stochastic variant of OASIS (see Appendix B for details) that only considers a small mini-batch of training data in each iteration.

The Stochastic OASIS chooses sets $\mathcal{I}_k, \mathcal{J}_k \subset [n]$ randomly and independently, and the new iterate is computed as: $w_{k+1} = w_k - \eta_k \hat{D}_k^{-1} \nabla F_{\mathcal{I}_k}(w_k)$, where $\nabla F_{\mathcal{I}_k}(w_k) = \frac{1}{|\mathcal{I}_k|} \sum_{i \in \mathcal{I}_k} \nabla F_i(w_k)$ and $\hat{D}_k$ is the truncated variant of $D_k = \beta_2 D_{k-1} + (1 - \beta_2) \, \mathtt{diag}(z_k \odot \nabla^2 F_{\mathcal{J}_k}(w_k) z_k)$ with $\nabla^2 F_{\mathcal{J}_k}(w_k) = \frac{1}{|\mathcal{J}_k|} \sum_{j \in \mathcal{J}_k} \nabla^2 F_j(w_k)$.

## 3.3 WARMSTARTING AND COMPLEXITY OF `OASIS`

Both deterministic and stochastic variants of our methodology share the need to obtain an initial estimate of $D_0$. The importance of this is illustrated by the rule (10), which regulates the choice of $\eta_k$, which is highly dependent on $D_k$. In order to have a better approximation of $D_0$, we propose to sample some predefined number of Hutchinson's estimates before the training process.

The main overhead of our methodology is the Hessian-vector product used in Hutchinson's method for approximating the Hessian diagonal. With the current advanced hardware and packages, this computation is *not* a bottleneck anymore. To be more specific, the Hessian-vector product can be efficiently calculated by *two* rounds of back-propagation. We also present how to calculate the Hessian-vector product efficiently for various well-known machine learning tasks in Appendix B.

## 4 THEORETICAL ANALYSIS

In this section, we present our theoretical results for `OASIS`. We show convergence guarantees for different settings of learning rates, i.e., $(i)$ adaptive learning rate, $(ii)$ fixed learning rate, and $(iii)$ with line search. Before the main theorems are presented, we state the following assumptions and lemmas that are used throughout this section. For brevity, we present the theoretical results using line search in Appendix A. The proofs and other auxiliary lemmas are in Appendix A.

**Assumption 4.1.** *(Convex). The function $F$ is convex.*

**Assumption 4.2.** *($L-$smooth). The gradients of $F$ are $L-$Lipschitz continuous for all $w \in \mathbb{R}^d$, i.e., $\exists L > 0$ such that $\forall w, w' \in \mathbb{R}^d, F(w) \leq F(w') + \langle \nabla F(w'), w - w' \rangle + \frac{L}{2}\|w - w'\|^2$.*

**Assumption 4.3.** *The function $F$ is twice continuously differentiable.*

**Assumption 4.4.** *($\mu-$strongly convex). The function $F$ is $\mu-$strongly convex, i.e., there exists a constant $\mu > 0$ such that $\forall w, w' \in \mathbb{R}^d$, $F(w) \geq F(w') + \langle \nabla F(w'), w - w' \rangle + \frac{\mu}{2}\|w - w'\|^2$.*

Here is a lemma regarding the bounds for Hutchinson's approximation and the diagonal differences.

**Lemma 4.5.** *(Bound on change of $D_k$). Suppose that Assumption 4.2 holds, then i) $|(v_k)_i| \leq \Gamma \leq \sqrt{d}L$, where $v_k = z_k \odot \nabla^2 F(w_k)z_k$; ii) $\exists \delta \leq 2(1 - \beta_2)\Gamma$ such that $\forall k : \|D_{k+1} - D_k\|_\infty \leq \delta$.*

## 4.1 ADAPTIVE LEARNING RATE

Here, we present theoretical convergence results for the case with adaptive learning rate using (10).

**Theorem 4.6.** *Suppose that Assumptions 4.1, 4.2 and 4.3 hold. Let $\{w_k\}$ be the iterates generated by Algorithm `OASIS`, then we have: $F(\hat{w}_k) - F^* \leq \frac{LC}{k} + 2L(1 - \beta_2)\Gamma\frac{Q_k}{k}$, where $C = \frac{2\|w_1 - w^*\|_{\hat{D}_0}^2 + \|w_1 - w_0\|_{\hat{D}_0}^2}{2} + 2\eta_1\theta_1(F(w_0) - F(w^*))$, and $Q_k = \sum_{i=1}^k (\frac{2\eta_i\theta_i + \alpha}{2\alpha}\|w_{i-1} - w_i\|^2 + \frac{L^2\eta_i\theta_i + \alpha}{\alpha}\|w_i - w^*\|^2)$.*

**Remark 4.7.** *The following remarks are made regarding Theorem 4.6:*

1. *If $\beta_2 = 1$, $\alpha = 1$, and $D_0 = I_d$, the results in Theorem 4.6 completely match with (Malitsky & Mishchenko, 2020).*

2. *By considering an extra assumption as in (Reddi et al., 2019; Duchi et al., 2011) regarding bounded iterates, i.e., $\|w_k - w^*\|^2 \leq B \quad \forall k \geq 0$, one can easily show the convergence of `OASIS` to the neighborhood of optimal solution(s).*

The following lemma provides the bounds for the adaptive learning rate for smooth and strongly-convex loss functions. The next theorem provides the linear convergence rate for the latter setting.

**Lemma 4.8.** *Suppose that Assumptions 4.2, 4.3, and 4.4 hold, then $\eta_k \in [\frac{\alpha}{2L}, \frac{\Gamma}{2\mu}]$.*

**Theorem 4.9.** *Suppose that Assumptions 4.2, 4.3, and 4.4 hold, and let $w^*$ be the unique solution for (1). Let $\{w_k\}$ be the iterates generated by Algorithm 1. Then, for all $k \geq 0$ and $\beta_2 \geq \max\{1 - \frac{\alpha^4\mu^4}{4L^2\Gamma^2(\alpha^2\mu^2 + L\Gamma^2)}, 1 - \frac{\alpha^3\mu^3}{4L\Gamma(2\alpha^2\mu^2 + L^3\Gamma^2)}\}$ we have: $\Psi^{k+1} \leq (1 - \frac{\alpha^2}{2\Gamma^2\kappa^2})\Psi^k$, where $\Psi^{k+1} = \|w_{k+1} - w^*\|_{\hat{D}_k}^2 + \frac{1}{2}\|w_{k+1} - w_k\|_{\hat{D}_k}^2 + 2\eta_k(1 + \theta_k)(F(w_k) - F(w^*))$.*

## 4.2 FIXED LEARNING RATE

Here, we provide theoretical results for fixed learning rate for deterministic and stochastic regimes.

**Remark 4.10.** *For any $k \geq 0$, we have $\alpha I \preceq \hat{D}_k \preceq \Gamma I$ where $0 < \alpha \leq \Gamma$.*

### 4.2.1 DETERMINISTIC REGIME

**Strongly Convex.** The following theorem provides the linear convergence for the smooth and strongly-convex loss functions with fixed learning rate.

**Theorem 4.11.** *Suppose that Assumptions 4.2, 4.3, and 4.4 hold, and let $F^* = F(w^*)$, where $w^*$ is the unique minimizer. Let $\{w_k\}$ be the iterates generated by Algorithm 1, where $0 < \eta_k = \eta \leq \frac{\alpha^2}{L\Gamma}$, and $w_0$ is a starting point. Then, for all $k \geq 0$ we have: $F(w_k) - F^* \leq (1 - \frac{\eta\mu}{\Gamma})^k [F(w_0) - F^*]$.*

**Nonconvex.** The following theorem provides the convergence to the stationary points for the nonconvex setting with fixed learning rate.

**Assumption 4.12.** *The function $F(.)$ is bounded below by a scalar $\hat{F}$.*

**Theorem 4.13.** *Suppose that Assumptions 4.2, 4.3, and 4.12 hold. Let $\{w_k\}$ be the iterates generated by Algorithm , where $0 < \eta_k = \eta \leq \frac{\alpha^2}{L\Gamma}$, and $w_0$ is a starting point. Then, for all $T > 1$ we have:*

$$\frac{1}{T}\sum_{k=1}^{T} \|\nabla F(w_k)\|^2 \leq \frac{2\Gamma[F(w_0) - \hat{F}]}{\eta T} \xrightarrow{T \to \infty} 0. \tag{11}$$

### 4.2.2 STOCHASTIC REGIME

Here, we use $\mathbb{E}_{\mathcal{I}_k}[.]$ to denote conditional expectation given $w_k$, and $\mathbb{E}[.]$ to denote the full expectation over the full history. The following standard assumptions as in (Bollapragada et al., 2019; Berahas et al., 2016) are considered for this section.

**Assumption 4.14.** *There exist a constant $\gamma$ such that $\mathbb{E}_{\mathcal{I}}[\|\nabla F_{\mathcal{I}}(w) - \nabla F(w)\|^2] \leq \gamma^2$.*

**Assumption 4.15.** *There exist a constant $\sigma^2 < \infty$ such that $\mathbb{E}_{\mathcal{I}}[\|\nabla F_{\mathcal{I}}(w^*)\|^2] \leq \sigma^2$.*

**Assumption 4.16.** *$\nabla F_{\mathcal{I}}(w)$ is an unbiased estimator of the gradient, i.e., $\mathbb{E}_{\mathcal{I}}[\nabla F_{\mathcal{I}}(w)] = \nabla F(w)$.*

**Strongly Convex.** The following theorem presents the convergence to the neighborhood of the optimal solution for the smooth and strongly-convex case in the stochastic setting.

**Theorem 4.17.** *Suppose that Assumptions 4.2, 4.3, 4.4, 4.15 and 4.16 hold. Let $\{w_k\}$ be the iterates generated by Algorithm 1 with $\eta_k = \eta \in (0, \frac{\alpha^2\mu}{\Gamma L^2})$, then, for all $k \geq 0$,*

$$\mathbb{E}[F(w_k) - F^*] \leq (1 - c)^k (F(w_0) - F^*) + \frac{\eta^2 L\sigma^2}{c\alpha^2}, \tag{12}$$

*where $c = \frac{2\eta\mu}{\Gamma} - \frac{2\eta^2 L^2}{\alpha^2} \in (0, 1)$. Moreover, if $\eta_k = \eta \in (0, \frac{\alpha^2\mu}{2\Gamma L^2})$ then*

$$\mathbb{E}[F(w_k) - F^*] \leq \left(1 - \frac{\eta\mu}{\Gamma}\right)^k (F(w_0) - F^*) + \frac{\eta\Gamma L\sigma^2}{\alpha^2\mu}. \tag{13}$$

**Nonconvex.** The next theorem provides the convergence to the stationary point for the nonconvex loss functions in the stochastic regime.

**Theorem 4.18.** *Suppose that Assumptions 4.2, 4.3, 4.12, 4.14 and 4.16 hold. Let $\{w_k\}$ be the iterates generated by Algorithm 1, where $0 < \eta_k = \eta \leq \frac{\eta^2}{L\Gamma}$, and $w_0$ is the starting point. Then, for all $k \geq 0$,*

$$\mathbb{E}\left[\frac{1}{T}\sum_{k=0}^{T-1} \|\nabla F(w_k)\|^2\right] \leq \frac{2\Gamma[F(w_0) - \hat{F}]}{\eta T} + \frac{\eta\Gamma\gamma^2 L}{\alpha^2} \xrightarrow{T \to \infty} \frac{\eta\Gamma\gamma^2 L}{\alpha^2}.$$

The previous two theorems provide the convergence to the neighborhood of the stationary points. One can easily use either a variance reduced gradient approximation or a decaying learning rate strategy to show the convergence to the stationary points (in expectation). OASIS's analyses are similar to those of limited-memory quasi-Newton approaches which depends on $\lambda_{max}$ and $\lambda_{min}$ (largest and smallest eigenvalues) of the preconditioning matrix (while in (S)GD $\lambda_{max} = \lambda_{min} = 1$). These methods in theory are not better than GD-type methods. In practice, however, they have shown their strength.

## 5 EMPIRICAL RESULTS

In this section, we present empirical results for several machine learning problems to show that our `OASIS` methodology outperforms state-of-the-art first- and second-order methods in both deterministic and stochastic regimes. We considered: (1) deterministic $\ell_2$-regularized logistic regression (strongly convex); (2) deterministic nonlinear least squares (nonconvex), and we report results on 2 standard machine learning datasets `rcv1` and `ijcnn1`[3]; and (3) image classification tasks on

---

[3]https://www.csie.ntu.edu.tw/ cjlin/libsvmtools/datasets/

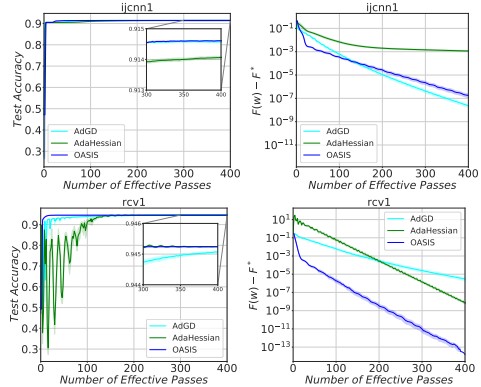

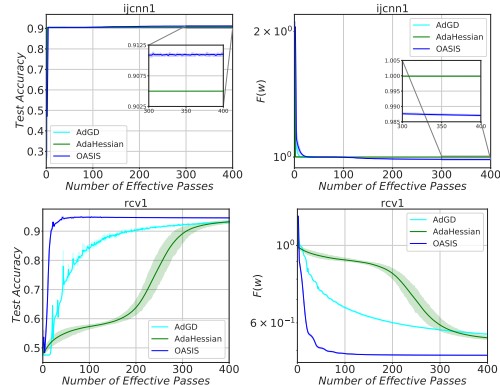

Figure 3: Comparison of optimality gap and Test Accuracy for different algorithms on Logistic Regression Problems.

Figure 4: Comparison of objective function ($F(w)$) and Test Accuracy for different algorithms on Non-linear Least Square Problems.

`MNIST`, `CIFAR10`, and `CIFAR100` datasets on standard network structures. In the interest of space, we report only a subset of the results in this section. The rest can be found in Appendix C.

To be clear, we compared the empirical performance of `OASIS` with algorithms with diagonal preconditioners. In the deterministic regime, we compared the performance of `OASIS` with AdGD (Malitsky & Mishchenko, 2020) and AdaHessian (Yao et al., 2020). Further, for the stochastic regime, we provide experiments comparing SGD (Robbins & Monro, 1951), Adam (Kingma & Ba, 2014), AdamW (Loshchilov & Hutter, 2017), and AdaHessian. For the logistic regression problems, the regularization parameter was chosen from the set $\lambda \in \{\frac{1}{10n}, \frac{1}{n}, \frac{10}{n}\}$. It is worth highlighting that we ran each method for each of the following experiments from 10 different random initial points. Moreover, we separately tuned the hyperparameters for each algorithm, if needed. See Appendix C for details. The proposed `OASIS` is robust with respect to different choices of hyperparameters, and it has a narrow spectrum of changes (see Appendix C).

**Logistic Regression.** We considered $\ell_2$-regularized logistic regression problems, $F(w) = \frac{1}{n}\sum_{i=1}^{n} \log(1 + e^{-y_i x_i^T w}) + \frac{\lambda}{2}\|w\|^2$. Figure 3 shows the performance of the methods in terms of optimality gap and test accuracy versus number of effective passes (number of gradient and Hessian-vector evaluations). As is clear, the performance of `OASIS` (with adaptive learning rate and without any hyperparameter tuning) is on par or better than that of the other methods.

**Non-linear Least Square.** We considered non-linear least squares problems (described in Xu et al. (2020)): $F(w) = \frac{1}{n}\sum_{i=1}^{n}(y_i - 1/(1 + e^{-x_i^T w}))^2$. Figure 4 shows that our `OASIS` Algorithm always outperforms the other methods in terms of training loss function and test accuracy. Moreover, the behaviour of `OASIS` is robust with respect to the different initial points.

**Image Classification.** We illustrate the performance of `OASIS` on standard bench-marking neural network training tasks: `MNIST`, `CIFAR10`, and `CIFAR100`. The results for `MNIST` and the details of the problems are given in Appendix C. We present the results regarding `CIFAR10`/`CIFAR100`.

**CIFAR10.** We use standard `ResNet-20` and `ResNet-32` (He et al., 2015) architectures for comparing the performance of `OASIS` with SGD, Adam, AdamW and AdaHessian[4]. Specifically, we report 3 variants of `OASIS`: ($i$) adaptive learning rate, ($ii$) fixed learning rate (without first moment) and ($iii$) fixed learning rate with gradient momentum tagged with "Adaptive LR," "Fixed LR," and "Momentum," respectively. For settings with fixed learning rate, no warmstarting is used in order to obtain an initial $D_0$ approximation. For the case with adaptive learning case, we used the warmstarting strategy to approximate the initial Hessian diagonal. More details regarding the exact parameter values and hyperparameter search can be found in the Appendix C. The results on `CIFAR10` are shown in the Figure 5 (the left and middle columns) and Table 1. As is clear, the simplest variant of `OASIS` with fixed learning rate achieves significantly better results, as compared to Adam, and a performance comparable to SGD. For the variant with an added momentum, we get similar accuracy as AdaHessian, while getting better or the same loss values, highlighting the advantage of using different preconditioning schema. Another important observation is that `OASIS`-Adaptive LR, without too much tuning efforts, has better performance than Adam with sensitive hyperparameters. All in all, the performance of `OASIS` variants is on par or better than the other state-of-the-art methods especially

---

[4]Note that we follow the same Experiment Setup as in (Yao et al., 2020), and the codes for other algorithms and structures are brought from https://github.com/amirgholami/adahessian.

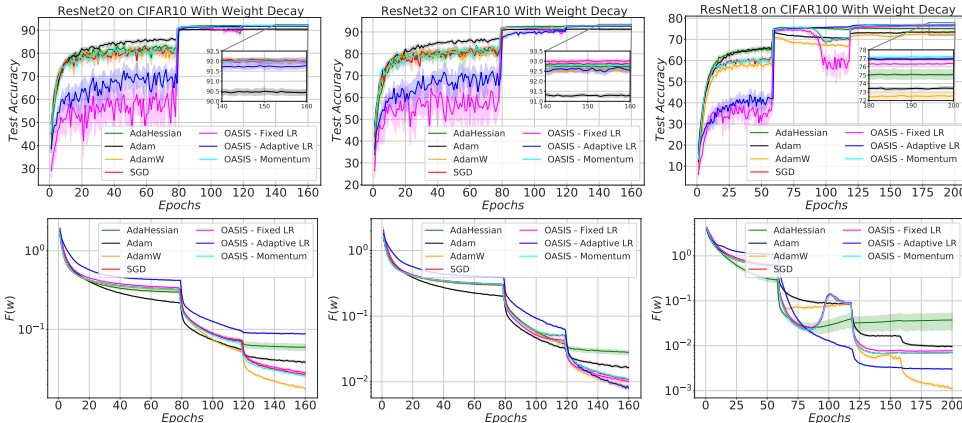

Figure 5: Performance of SGD, Adam, AdamW, Adehessian and different variants of `OASIS` on `CIFAR10` (left and middle columns) and `CIFAR100` (right column) problems on `ResNet-20` (left column), `ResNet-32` (middle column) and `ResNet-18` (right column).

SGD. As we see from Figure 5 (left and middle columns), the lack of momentum produces a slow, noisy training curve in the initial stages of training, while `OASIS` with momentum works better than the other two variants in the early stages. All three variants of `OASIS` get satisfactory results in the end of training. More results and discussion regarding `CIFAR10` dataset are in Appendix C.

Table 1: Results of `ResNet-20/32` on `CIFAR10`

| Setting | ResNet-20 | ResNet-32 |
|---|---|---|
| SGD | $92.02 \pm 0.22$ | $92.85 \pm 0.12$ |
| Adam | $90.46 \pm 0.22$ | $91.30 \pm 0.15$ |
| AdamW | $91.99 \pm 0.17$ | $92.58 \pm 0.25$ |
| AdaHessian | $\mathbf{92.03 \pm 0.10}$ | $92.71 \pm 0.26$ |
| `OASIS`- Adaptive LR | $91.20 \pm 0.20$ | $92.61 \pm 0.22$ |
| `OASIS`- Fixed LR | $91.96 \pm 0.21$ | $\mathbf{93.01 \pm 0.09}$ |
| `OASIS`- Momentum | $92.01 \pm 0.19$ | $92.77 \pm 0.18$ |

Table 2: Results of `ResNet-18` on `CIFAR100`.

| Setting | ResNet-18 |
|---|---|
| SGD | $76.57 \pm 0.24$ |
| Adam | $73.40 \pm 0.31$ |
| AdamW | $72.51 \pm 0.76$ |
| AdaHessian | $75.71 \pm 0.47$ |
| `OASIS`- Adaptive LR | $\mathbf{76.93 \pm 0.22}$ |
| `OASIS`- Fixed LR | $76.28 \pm 0.21$ |
| `OASIS`- Momentum | $76.89 \pm 0.34$ |

**CIFAR-100.** We use the hyperparameter settings obtained by training on `CIFAR10` on `ResNet-20/32` to train `CIFAR100` on `ResNet-18` network structure.[5] We similarly compare the performance of our method and its variants with SGD, Adam, AdamW and AdaHessian. The results are shown in Figure 5 (right column) and Table 2. In this setting, without any hyperparameter tuning, fully adaptive version of our algorithm immediately produces results surpassing other state-of-the-art methods especially SGD.

## 6  FINAL REMARKS

This paper presents a fully adaptive optimization algorithm for empirical risk minimization. The search direction uses the gradient information, well-scaled with a novel Hessian diagonal approximation, which itself can be calculated and stored efficiently. In addition, we do not need to tune the learning rate, which instead is automatically updated based on a low-cost approximation of the Lipschitz smoothness parameter. We provide comprehensive theoretical results covering standard optimization settings, including convex, strongly convex and nonconvex; and our empirical results highlight the efficiency of `OASIS` in large-scale machine learning problems.

Future research avenues include: (1) deriving the theoretical results for stochastic regime with adaptive learning rate; (2) employing variance reduction schemes in order to reduce further the noise in the gradient and Hessian diagonal estimates; and (3) providing a more extensive empirical investigation on other demanding machine learning problems such as those from natural language processing and recommendation system (such as those from the original AdaHessian paper (Yao et al., 2020)).

---

[5]https://github.com/uoguelph-mlrg/Cutout.

## Acknowledgements

MT was partially supported by the NSF, under award numbers CCF:1618717/CCF:1740796. PR was supported by the KAUST Baseline Research Funding Scheme. MM would like to acknowledge the US NSF and ONR via its BRC on RandNLA for providing partial support of this work. Our conclusions do not necessarily reflect the position or the policy of our sponsors, and no official endorsement should be inferred.

## ETHICS STATEMENT

This work presents a new algorithm for training machine learning models. We do not foresee any ethical concerns. All datasets used in this work are from the public domain and are commonly used benchmarks in ML papers.

## REPRODUCIBILITY STATEMENT

We uploaded all the codes used to make all the experiments presented in this paper. We have used random seeds to ensure that one can start optimizing the ML models from the same initial starting point as was used in the experiments. We have used only datasets that are in the public domain, and one can download them from the following website `https://www.csie.ntu.edu.tw/~cjlin/libsvmtools/datasets/`. After acceptance, we will include a link to the GitHub repository where we will host the source codes.

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

# A  THEORETICAL RESULTS AND PROOFS

## A.1  ASSUMPTIONS

**Assumption 4.1.** *(Convex). The function $F$ is convex, i.e., $\forall w, w' \in \mathbb{R}^d$,*

$$F(w) \geq F(w') + \langle \nabla F(w'), w - w' \rangle. \tag{14}$$

**Assumption 4.2.** *($L-$smooth). The gradients of $F$ are $L-$Lipschitz continuous for all $w \in \mathbb{R}^d$, i.e., there exists a constant $L > 0$ such that $\forall w, w' \in \mathbb{R}^d$,*

$$F(w) \leq F(w') + \langle \nabla F(w'), w - w' \rangle + \frac{L}{2}\|w - w'\|^2. \tag{15}$$

**Assumption 4.3.** *The function $F$ is twice continuously differentiable.*

**Assumption 4.4.** *($\mu-$strongly convex). The function $F$ is $\mu-$strongly convex, i.e., there exists a constant $\mu > 0$ such that $\forall w, w' \in \mathbb{R}^d$,*

$$F(w) \geq F(w') + \langle \nabla F(w'), w - w' \rangle + \frac{\mu}{2}\|w - w'\|^2. \tag{16}$$

**Assumption 4.12.** *The function $F(.)$ is bounded below by a scalar $\hat{F}$.*

**Assumption 4.14.** *There exist a constant $\gamma$ such that $\mathbb{E}_{\mathcal{I}}[\|\nabla F_{\mathcal{I}}(w) - \nabla F(w)\|^2] \leq \gamma^2$.*

**Assumption 4.16.** *$\nabla F_{\mathcal{I}}(w)$ is an unbiased estimator of the gradient, i.e., $\mathbb{E}_{\mathcal{I}}[\nabla F_{\mathcal{I}}(w)] = \nabla F(w)$, where the samples $\mathcal{I}$ are drawn independently.*

## A.2  PROOF OF LEMMA 4.5

**Lemma 4.5.** *(Bound on change of $D_k$). Suppose that Assumption 4.2 holds, i.e., $\forall w : \nabla^2 f(w) \preceq LI$, then*

1. *$|(v_k)_i| \leq \Gamma \leq \sqrt{d}L$, where $v_k = z_k \odot \nabla^2 F(w_k)z_k$.*

2. *there $\exists \delta \leq 2(1 - \beta_2)\Gamma$ such that*
$$\|D_{k+1} - D_k\|_\infty \leq \delta, \quad \forall k.$$

*Proof.* By Assumption 4.2, we have that $\|\nabla^2 F(w)\|_2 \leq L$ and hence
$$\|v_k\|_\infty \leq \|\nabla^2 F(w)\|_\infty \leq \sqrt{d}\|\nabla^2 F(w)\|_2 \leq \sqrt{d}L,$$
which finishes the proof of case 1.

Now, from (8) we can derive
$$D_{k+1} - D_k \overset{(8)}{=} (\beta_2 - 1)D_k + (1 - \beta_2)\, z_k \odot \nabla^2 F(w_k)z_k$$

and hence
$$\|D_{k+1} - D_k\|_\infty = (1 - \beta_2)\|D_k - z_k \odot \nabla^2 F(w_k)z_k\|_\infty \leq (1 - \beta_2)\|D_k - v_k\|_\infty$$
$$\leq (1 - \beta_2)\left(\|D_k\|_\infty + \|v_k\|_\infty\right) \leq 2(1 - \beta_2)\Gamma. \qquad \square$$

## A.3  LEMMA REGARDING SMOOTHNESS WITH WEIGHTED NORM

In the following, we present a lemma for smoothness with weighted norm. Theorem 2.1.5 in (Nesterov et al., 2018) provides the same analysis for any norm, and the following lemma can be seen as a special case of Theorem 2.1.5 (Nesterov et al., 2018) with respect to the weighted Euclidean norm. In the following, we provide the proof for completeness.

**Lemma A.1.** *(Smoothness with norm $D$) Suppose that Assumptions 4.1 and 4.2 hold, and $D \succ 0$, then we have:*

$$\|\nabla F(x) - \nabla F(y)\|_D^* \leq \tilde{L}\|x - y\|_D, \tag{17}$$

*where $\tilde{L} = \dfrac{L}{\lambda_{\min}(D)}$.*

*Proof.* By the equality $\tilde{L} = \dfrac{L}{\lambda_{\min}(D)}$, we conclude that $\tilde{L}D \succeq LI$ which results in:

$$F(y) \le F(x) + \langle \nabla F(x), y - x \rangle + \frac{\tilde{L}}{2}\|x - y\|_D^2. \tag{18}$$

Extending proof of Theorem 2.1.5. in (Nesterov, 2013) we define $\phi(y) = F(y) - \langle \nabla F(x), y \rangle$. Then, clearly, $x \in \arg\min \phi(y)$ and (18) is still valid for $\phi(y)$.

Therefore

$$\phi(x) \le \phi(y - \frac{D^{-1}}{\tilde{L}}\nabla\phi(y)),$$

$$\phi(x) \overset{(18)}{\le} \phi(y) + \langle \nabla\phi(y), -\frac{D^{-1}}{\tilde{L}}\nabla\phi(y)\rangle + \frac{\tilde{L}}{2}\| - \frac{D^{-1}}{\tilde{L}}\nabla\phi(y)\|_D^2,$$

$$F(x) - \langle \nabla F(x), x \rangle \le F(y) - \langle \nabla F(x), y \rangle - \frac{1}{\tilde{L}}\langle \nabla\phi(y), D^{-1}\nabla\phi(y)\rangle + \frac{1}{2\tilde{L}}\|D^{-1}\nabla\phi(y)\|_D^2,$$

$$F(x) \le F(y) + \langle \nabla F(x), x - y \rangle - \frac{1}{2\tilde{L}}(\|\nabla\phi(y)\|_D^*)^2,$$

$$F(x) \le F(y) + \langle \nabla F(x), x - y \rangle - \frac{1}{2\tilde{L}}(\|\nabla F(y) - \nabla F(x)\|_D^*)^2.$$

Thus

$$F(x) + \langle \nabla F(x), y - x \rangle + \frac{1}{2\tilde{L}}(\|\nabla F(y) - \nabla F(x)\|_D^*)^2 \le F(y). \tag{19}$$

Adding (19) with itself with $x$ swapped with $y$ we obtain

$$\frac{1}{\tilde{L}}(\|\nabla F(y) - \nabla F(x)\|_D^*)^2 \le \langle \nabla F(y) - \nabla F(x), y - x \rangle$$

$$= \langle D^{-1}(\nabla F(y) - \nabla F(x)), D(y - x)\rangle$$

$$\le \|\nabla F(y) - \nabla F(x)\|_D^*\|y - x\|_D,$$

which implies that

$$\|\nabla F(x) - \nabla F(y)\|_D^* \le \tilde{L}\|x - y\|_D. \tag{20}$$

$\square$

### A.4 PROOF OF THEOREM 4.6

**Lemma A.2.** *Let $f : \mathbb{R}^d \to \mathbb{R}$ be a convex function, and $x^*$ is one of the optimal solutions for (1). Then, for the sequence of $\{w_k\}$ generated by Algorithm 1 we have:*

$$\|w_{k+1} - w^*\|_{\hat{D}_k}^2 + \frac{1}{2}\|w_k - w_{k+1}\|_{\hat{D}_k}^2 + 2\eta_k(1 + \theta_k)(F(w_k) - F(w^*))$$

$$\le \|w_k - w^*\|_{\hat{D}_{k-1}}^2 + \frac{1}{2}\|w_k - w_{k-1}\|_{\hat{D}_{k-1}}^2 + 2\eta_k\theta_k(F(w_{k-1}) - F(w^*)) +$$

$$2(1 - \beta_2)\Gamma\Big(\big(\frac{\eta_k\theta_k}{\alpha} + \frac{1}{2}\big)\|w_{k-1} - w_k\|^2 + \big(\frac{L^2\eta_k\theta_k}{\alpha} + 1\big)\|w_k - w^*\|^2\Big). \tag{21}$$

*Proof.* We extend the proof in (Malitsky & Mishchenko, 2020). We have

$$\|w_{k+1} - w^*\|_{\hat{D}_k}^2 = \|w_{k+1} - w_k + w_k - w^*\|_{\hat{D}_k}^2$$

$$= \|w_{k+1} - w_k\|_{\hat{D}_k}^2 + \|w_k - w^*\|_{\hat{D}_k}^2 + 2\langle w_{k+1} - w_k, \hat{D}_k(w_k - w^*)\rangle$$

$$= \|w_{k+1} - w_k\|_{\hat{D}_k}^2 + \|w_k - w^*\|_{\hat{D}_k}^2 + 2\eta_k\langle \nabla F(w_k), w^* - w_k\rangle$$

$$\le \|w_{k+1} - w_k\|_{\hat{D}_k}^2 + \|w_k - w^*\|_{\hat{D}_k}^2 + 2\eta_k(F(w^*) - F(w_k)) \tag{22}$$

$$= \|w_{k+1} - w_k\|_{\hat{D}_k}^2 + \|w_k - w^*\|_{\hat{D}_k}^2 - 2\eta_k(F(w_k) - F(w^*)),$$

where the third equality comes from the `OASIS`'s step and the inequality follows from convexity of $F(w)$. Now, let's focus on $\|w_{k+1} - w_k\|^2_{\hat{D}_k}$. We have

$$
\begin{aligned}
\|w_{k+1} - w_k\|^2_{\hat{D}_k} &= 2\|w_{k+1} - w_k\|^2_{\hat{D}_k} - \|w_{k+1} - w_k\|^2_{\hat{D}_k} \\
&= 2\langle -\eta_k \hat{D}_k^{-1} \nabla F(w_k), \hat{D}_k(w_{k+1} - w_k)\rangle - \|w_{k+1} - w_k\|^2_{\hat{D}_k} \\
&= -2\eta_k \langle \nabla F(w_k), w_{k+1} - w_k \rangle - \|w_{k+1} - w_k\|^2_{\hat{D}_k} \\
&= -2\eta_k \langle \nabla F(w_k) - \nabla F(w_{k-1}), w_{k+1} - w_k \rangle - 2\eta_k \langle \nabla F(w_{k-1}), w_{k+1} - w_k \rangle - \\
&\quad \|w_{k+1} - w_k\|^2_{\hat{D}_k} \\
&= 2\eta_k \langle \nabla F(w_k) - \nabla F(w_{k-1}), w_k - w_{k+1}\rangle + 2\eta_k \langle \nabla F(w_{k-1}), w_k - w_{k+1}\rangle - \\
&\quad \|w_{k+1} - w_k\|^2_{\hat{D}_k}.
\end{aligned}
$$

Now,

$$
\begin{aligned}
2\eta_k \langle \nabla F(w_k) - \nabla F(w_{k-1}), w_k - w_{k+1}\rangle &\le 2\eta_k \|\nabla F(w_k) - \nabla F(w_{k-1})\|^*_{\hat{D}_k} \|w_k - w_{k+1}\|_{\hat{D}_k} \\
&\stackrel{(10)}{\le} \|w_k - w_{k-1}\|_{\hat{D}_k} \|w_k - w_{k+1}\|_{\hat{D}_k} \\
&\le \frac{1}{2}\|w_k - w_{k-1}\|^2_{\hat{D}_k} + \frac{1}{2}\|w_k - w_{k+1}\|^2_{\hat{D}_k},
\end{aligned}
$$

where the first inequality comes from Cauchy-Schwarz and the third one follows Young's inequality. Further,

$$
\begin{aligned}
\langle \nabla F(w_{k-1}), w_k - w_{k+1}\rangle &= \frac{1}{\eta_{k-1}} \langle \hat{D}_{k-1}(w_{k-1} - w_k), w_k - w_{k+1}\rangle \\
&= \frac{1}{\eta_{k-1}} \langle \hat{D}_{k-1}(w_{k-1} - w_k), \eta_k \hat{D}_k^{-1} \nabla F(w_k)\rangle \\
&= \frac{\eta_k}{\eta_{k-1}} (w_{k-1} - w_k)^T \hat{D}_{k-1} \hat{D}_k^{-1} \nabla F(w_k) \\
&= \frac{\eta_k}{\eta_{k-1}} (w_{k-1} - w_k)^T \nabla F(w_k) + \\
&\quad \frac{\eta_k}{\eta_{k-1}} (w_{k-1} - w_k)^T \left(\hat{D}_{k-1} \hat{D}_k^{-1} - I\right) \nabla F(w_k), \qquad (23)
\end{aligned}
$$

where the first two qualities are due the `OASIS`'s update step. The second term in the above equality can be upperbounded as follows:

$$
(w_{k-1} - w_k)^T \left(\hat{D}_{k-1} \hat{D}_k^{-1} - I\right) \nabla F(w_k) \le (1 - \beta_2)\frac{2\Gamma}{\alpha} \cdot \|w_{k-1} - w_k\| \cdot \|\nabla F(w_k)\|, \quad (24)
$$

where multiplier on the left is obtained via the bound on $\|\cdot\|_\infty$ norm of the diagonal matrix $\hat{D}_{k-1} \hat{D}_k^{-1} - I$:

$$
\|\hat{D}_{k-1} \hat{D}_k^{-1} - I\|_\infty = \max_i \left|\left(\hat{D}_{k-1} \hat{D}_k^{-1} - I\right)_i\right| = \max_i \left|(\hat{D}_{k-1} - \hat{D}_k)_i\right| \left|(\hat{D}_k^{-1})_i\right| \le (1 - \beta_2)\frac{2\Gamma}{\alpha},
$$

which also represents a bound on the operator norm of the same matrix difference. Next we can use the Young's inequality and $\nabla F(w^*) = 0$ to get

$$
(1 - \beta_2)\frac{2\Gamma}{\alpha} \cdot \|w_{k-1} - w_k\| \cdot \|\nabla F(w_k)\| \le (1 - \beta_2)\frac{\Gamma}{\alpha}\left(\|w_{k-1} - w_k\|^2 + L^2\|w_k - w^*\|^2\right). \quad (25)
$$

Therefore, we have

$$
\begin{aligned}
\langle \nabla F(w_{k-1}), w_k - w_{k+1}\rangle &\le \frac{\eta_k}{\eta_{k-1}} (w_{k-1} - w_k)^T \nabla F(w_k) + \\
&\quad (1 - \beta_2)\frac{\Gamma}{\alpha}\left(\|w_{k-1} - w_k\|^2 + L^2\|w_k - w^*\|^2\right).
\end{aligned}
$$

Also,

$$\|w_{k+1} - w_k\|_{\hat{D}_k}^2 \le \frac{1}{2}\|w_k - w_{k-1}\|_{\hat{D}_k}^2 + \frac{1}{2}\|w_k - w_{k+1}\|_{\hat{D}_k}^2 + 2\eta_k\theta_k(w_{k-1} - w_k)^T\nabla F(w_k) +$$

$$2\eta_k\theta_k(1 - \beta_2)\frac{\Gamma}{\alpha}\big(\|w_{k-1} - w_k\|^2 + L^2\|w_k - w^*\|^2\big) - \|w_{k+1} - w_k\|_{\hat{D}_k}^2$$

$$\le \frac{1}{2}\|w_k - w_{k-1}\|_{\hat{D}_k}^2 + \frac{1}{2}\|w_k - w_{k+1}\|_{\hat{D}_k}^2 + 2\eta_k\theta_k(F(w_{k-1}) - F(w_k)) +$$

$$2\eta_k\theta_k(1 - \beta_2)\frac{\Gamma}{\alpha}\big(\|w_{k-1} - w_k\|^2 + L^2\|w_k - w^*\|^2\big) - \|w_{k+1} - w_k\|_{\hat{D}_k}^2.$$

Finally, we have

$$\|w_{k+1} - w^*\|_{\hat{D}_k}^2 \le \frac{1}{2}\|w_k - w_{k-1}\|_{\hat{D}_k}^2 + \frac{1}{2}\|w_k - w_{k+1}\|_{\hat{D}_k}^2 + 2\eta_k\theta_k(F(w_{k-1}) - F(w_k)) +$$

$$2\eta_k\theta_k(1 - \beta_2)\frac{\Gamma}{\alpha}\big(\|w_{k-1} - w_k\|^2 + L^2\|w_k - w^*\|^2\big) - \|w_{k+1} - w_k\|_{\hat{D}_k}^2$$

$$+ \|w_k - w^*\|_{\hat{D}_k}^2 - 2\eta_k(F(w_k) - F(w^*)).$$

By simplifying the above inequality, we have:

$$\|w_{k+1} - w^*\|_{\hat{D}_k}^2 + \frac{1}{2}\|w_k - w_{k+1}\|_{\hat{D}_k}^2 + 2\eta_k(1 + \theta_k)(F(w_k) - F(w^*))$$

$$\le \|w_k - w^*\|_{\hat{D}_k}^2 + \frac{1}{2}\|w_k - w_{k-1}\|_{\hat{D}_k}^2 + 2\eta_k\theta_k(F(w_{k-1}) - F(w^*)) +$$

$$2\eta_k\theta_k(1 - \beta_2)\frac{\Gamma}{\alpha}\big(\|w_{k-1} - w_k\|^2 + L^2\|w_k - w^*\|^2\big)$$

$$= \|w_k - w^*\|_{\hat{D}_{k-1}}^2 + \frac{1}{2}\|w_k - w_{k-1}\|_{\hat{D}_{k-1}}^2 + 2\eta_k\theta_k(F(w_{k-1}) - F(w^*)) +$$

$$2\eta_k\theta_k(1 - \beta_2)\frac{\Gamma}{\alpha}\big(\|w_{k-1} - w_k\|^2 + L^2\|w_k - w^*\|^2\big) +$$

$$\|w_k - w^*\|_{\hat{D}_k - \hat{D}_{k-1}}^2 + \frac{1}{2}\|w_k - w_{k-1}\|_{\hat{D}_k - \hat{D}_{k-1}}^2$$

$$\le \|w_k - w^*\|_{\hat{D}_{k-1}}^2 + \frac{1}{2}\|w_k - w_{k-1}\|_{\hat{D}_{k-1}}^2 + 2\eta_k\theta_k(F(w_{k-1}) - F(w^*)) +$$

$$2\eta_k\theta_k(1 - \beta_2)\frac{\Gamma}{\alpha}\big(\|w_{k-1} - w_k\|^2 + L^2\|w_k - w^*\|^2\big) +$$

$$2(1 - \beta_2)\Gamma(\|w_k - w^*\|^2 + \frac{1}{2}\|w_k - w_{k-1}\|^2)$$

$$= \|w_k - w^*\|_{\hat{D}_{k-1}}^2 + \frac{1}{2}\|w_k - w_{k-1}\|_{\hat{D}_{k-1}}^2 + 2\eta_k\theta_k(F(w_{k-1}) - F(w^*)) +$$

$$2(1 - \beta_2)\Gamma\Big(\frac{\eta_k\theta_k}{\alpha}\|w_{k-1} - w_k\|^2 + \frac{L^2\eta_k\theta_k}{\alpha}\|w_k - w^*\|^2 + \|w_k - w^*\|^2 +$$

$$\frac{1}{2}\|w_k - w_{k-1}\|^2\Big).$$

$$\square$$

**Theorem 4.6.** *Suppose that Assumptions 4.1, 4.2 and 4.3 hold. Let $\{w_k\}$ be the iterates generated by Algorithm 1, then we have:*

$$F(\hat{w}_k) - F^* \le \frac{LC}{k} + 2L(1 - \beta_2)\Gamma\frac{Q_k}{k},$$

*where*

$$C = \|w_1 - w^*\|_{\hat{D}_0}^2 + \frac{1}{2}\|w_1 - w_0\|_{\hat{D}_0}^2 + 2\eta_1\theta_1(F(w_0) - F(w^*))$$

$$Q_k = \sum_{i=1}^{k}\Big((\frac{\eta_i\theta_i}{\alpha} + \frac{1}{2})\|w_{i-1} - w_i\|^2 + (\frac{L^2\eta_i\theta_i}{\alpha} + 1)\|w_i - w^*\|^2\Big).$$

*Proof.* By telescoping inequality (21) in Lemma A.2 we have:

$$\|w_{k+1} - w^*\|_{\hat{D}_k}^2 + \frac{1}{2}\|w_k - w_{k+1}\|_{\hat{D}_k}^2 + 2\eta_k(1+\theta_k)(F(w_k) - F(w^*))$$

$$+ 2\sum_{i=1}^{k-1}[\eta_i(1+\theta_i) - \eta_{i+1}\theta_{i+1}](F(w_k) - F(w^*))$$

$$\leq \underbrace{\|w_1 - w^*\|_{\hat{D}_0}^2 + \frac{1}{2}\|w_1 - w_0\|_{\hat{D}_0}^2 + 2\eta_1\theta_1(F(w_0) - F(w^*)) +}_{C}$$

$$2(1-\beta_2)\Gamma\underbrace{\sum_{i=1}^{k}\left((\frac{\eta_i\theta_i}{\alpha} + \frac{1}{2})\|w_{i-1} - w_i\|^2 + (\frac{L^2\eta_i\theta_i}{\alpha} + 1)\|w_i - w^*\|^2\right)}_{Q_k}.$$

Moreover, by the rule for adaptive learning rule we know $\eta_i(1+\theta_i) - \eta_{i+1}\theta_{i+1} \geq 0, \ \forall i$. Therefore, we have

$$2\eta_k(1+\theta_k)(F(w_k) - F(w^*)) + 2\sum_{i=1}^{k-1}[\eta_i(1+\theta_i) - \eta_{i+1}\theta_{i+1}](F(w_k) - F(w^*))$$

$$\leq C + 2(1-\beta_2)\Gamma Q_k.$$

By setting $\hat{w} = \dfrac{\eta_k(1+\theta_k)w_k + \sum_{i=1}^{k-1}(\eta_i(1+\theta_i) - \eta_{i+1}\theta_{i+1})w_i}{S_k}$, where $S_k := \eta_k(1+\theta_k) + \sum_{i=1}^{k-1}(\eta_i(1+\theta_i) - \eta_{i+1}\theta_{i+1})$, and by using Jensens's inequality, we have:

$$F(\hat{w}_k) - F^* \leq \frac{C}{2S_k} + (1-\beta_2)\Gamma\frac{Q_k}{S_k}.$$

By the fact that $\eta_k \geq \dfrac{1}{2L}$ thus $\dfrac{1}{S_k} \leq \dfrac{2L}{k}$, we have

$$F(\hat{w}_k) - F^* \leq \frac{LC}{k} + 2L(1-\beta_2)\Gamma\frac{Q_k}{k}.$$

$\square$

## A.5 PROOF OF LEMMA 4.8

**Lemma 4.8.** *Suppose that Assumptions 4.2, 4.3 and 4.4 hold, then $\eta_k \in \left[\dfrac{\alpha}{2L}, \dfrac{\Gamma}{2\mu}\right]$.*

*Proof.* By (17) and the point that $\tilde{L} = \dfrac{L}{\lambda_{\min}(\hat{D}_k)}$, we conclude that:

$$\frac{\|w_k - w_{k-1}\|_{\hat{D}_k}}{2\|\nabla F(w_k) - \nabla F(w_{k-1})\|_{\hat{D}_k}^*} \geq \frac{1}{2\tilde{L}} = \frac{\lambda_{\min}(\hat{D}_k)}{2L} \geq \frac{\alpha}{2L}.$$

Now, in order to find the upperbound for $\frac{\|w_k - w_{k-1}\|_{\hat{D}_k}}{2\|\nabla F(w_k) - \nabla F(w_{k-1})\|_{\hat{D}_k}^*}$, we use the following inequality which comes from Assumption 4.4:

$$F(w_k) \geq F(w_{k-1}) + \langle\nabla F(w_{k-1}), (w_k - w_{k-1})\rangle + \frac{\mu}{2}\|w - w_{k-1}\|^2. \tag{26}$$

By setting $\tilde{\mu} = \dfrac{\mu}{\lambda_{\max}(\hat{D}_k)}$, we conclude that $\tilde{\mu}\hat{D}_k \preceq \mu I$ which results in:

$$F(w_k) \geq F(w_{k-1}) + \langle\nabla F(w_{k-1}), (w_k - w_{k-1})\rangle + \frac{\tilde{\mu}}{2}\|w - w_{k-1}\|_{\hat{D}_k}^2. \tag{27}$$

The above inequality results in:

$$\tilde{\mu}\|w - w_{k-1}\|^2_{\hat{D}_k} \leq \langle \nabla F(w_k) - \nabla F(w_{k-1}), (w_k - w_{k-1}) \rangle$$

$$= \langle \hat{D}_k^{-1}(\nabla F(w_k) - \nabla F(w_{k-1})), \hat{D}_k(w_k - w_{k-1}) \rangle$$

$$\leq \|\nabla F(w_k) - \nabla F(w_{k-1}))\|^*_{\hat{D}_k} \|w_k - w_{k-1}\|_{\hat{D}_k}. \tag{28}$$

Therefore, we obtain that

$$\frac{\|w_k - w_{k-1}\|_{\hat{D}_k}}{2\|\nabla F(w_k) - \nabla F(w_{k-1})\|^*_{\hat{D}_k}} \overset{(28)}{\leq} \frac{1}{2\tilde{\mu}} = \frac{\lambda_{\max}(\hat{D}_k)}{2\mu} \leq \frac{\Gamma}{2\mu},$$

and therefore, by the update rule for $\eta_k$, we conclude that $\eta_k \in \left[\dfrac{\alpha}{2L}, \dfrac{\Gamma}{2\mu}\right]$. □

## A.6 PROOF OF THEOREM 4.9

**Lemma A.3.** . *Suppose Assumptions 4.2, 4.3 and 4.4 hold and let $w^*$ be the unique solution of (1). Then for $(w_k)$ generated by Algorithm 1 we have:*

$$\|w_{k+1} - w^*\|^2_{\hat{D}_k} + \frac{1}{2}\|w_{k+1} - w_k\|^2_{\hat{D}_k} + 2\eta_k(1 + \theta_k)(F(w_k) - F(w^*))$$

$$\leq \|w_k - w^*\|^2_{\hat{D}_{k-1}} + \frac{1}{2}\|w_k - w_{k-1}\|^2_{\hat{D}_{k-1}} + 2\eta_k\theta_k(F(w_{k-1}) - F(w^*))$$

$$+ \left( (1 - \beta_2)\Gamma\left(1 + \frac{2\theta_k\eta_k}{\alpha}\right) - \mu\eta_k\theta_k \right) \|w_k - w_{k-1}\|^2$$

$$+ \left( (1 - \beta_2)\Gamma\left(2 + \frac{2L^2\theta_k\eta_k}{\alpha}\right) - \mu\eta_k \right) \|w_k - w^*\|^2.$$

*Proof.* By the update rule in Algorithm 1 we have:

$$\|w_{k+1} - w^*\|^2_{\hat{D}_k} = \|w_{k+1} - w_k + w_k - w^*\|^2_{\hat{D}_k}$$

$$= \|w_{k+1} - w_k\|^2_{\hat{D}_k} + \|w_k - w^*\|^2_{\hat{D}_k} + 2\langle w_{k+1} - w_k, \hat{D}_k(w_k - w^*)\rangle$$

$$= \|w_{k+1} - w_k\|^2_{\hat{D}_k} + \|w_k - w^*\|^2_{\hat{D}_k} + 2\eta_k\langle \nabla F(w_k), w^* - w_k\rangle$$

$$\leq \|w_{k+1} - w_k\|^2_{\hat{D}_k} + \|w_k - w^*\|^2_{\hat{D}_k} + 2\eta_k(F(w^*) - F(w_k)) \tag{29}$$

$$= \|w_{k+1} - w_k\|^2_{\hat{D}_k} + \|w_k - w^*\|^2_{\hat{D}_k} - 2\eta_k(F(w_k) - F(w^*)),$$

where the inequality follows from convexity of $F(w)$. By strong convexity of $F(.)$ we have:

$$F(w^*) \geq F(w_k) + \langle \nabla F(w_k), w^* - w_k \rangle + \frac{\mu}{2}\|w_k - w^*\|^2. \tag{30}$$

In the lights of the strongly convex inequality we can change (29) as follows

$$\|w_{k+1} - w^*\|^2_{\hat{D}_k} \overset{(29),(30)}{\leq} \|w_{k+1} - w_k\|^2_{\hat{D}_k} + \|w_k - w^*\|^2_{\hat{D}_k} + 2\eta_k(F(w^*) - F(w_k) - \frac{\mu}{2}\|w_k - w^*\|^2)$$

$$= \|w_{k+1} - w_k\|^2_{\hat{D}_k} + \|w_k - w^*\|^2_{\hat{D}_k} - 2\eta_k(F(w_k) - F(w^*)) - \mu\eta_k\|w_k - w^*\|^2. \tag{31}$$

Now, let's focus on $\|w_{k+1} - w_k\|_{\hat{D}_k}^2$. We have

$$
\begin{aligned}
\|w_{k+1} - w_k\|_{\hat{D}_k}^2 =& 2\|w_{k+1} - w_k\|_{\hat{D}_k}^2 - \|w_{k+1} - w_k\|_{\hat{D}_k}^2 \\
=& 2\langle -\eta_k \hat{D}_k^{-1} \nabla F(w_k), \hat{D}_k(w_{k+1} - w_k)\rangle - \|w_{k+1} - w_k\|_{\hat{D}_k}^2 \\
=& -2\eta_k\langle \nabla F(w_k), w_{k+1} - w_k\rangle - \|w_{k+1} - w_k\|_{\hat{D}_k}^2 \\
=& -2\eta_k\langle \nabla F(w_k) - \nabla F(w_{k-1}), w_{k+1} - w_k\rangle - 2\eta_k\langle \nabla F(w_{k-1}), w_{k+1} - w_k\rangle - \\
& \|w_{k+1} - w_k\|_{\hat{D}_k}^2 \\
=& 2\eta_k\langle \nabla F(w_k) - \nabla F(w_{k-1}), w_k - w_{k+1}\rangle + 2\eta_k\langle \nabla F(w_{k-1}), w_k - w_{k+1}\rangle - \\
& \|w_{k+1} - w_k\|_{\hat{D}_k}^2.
\end{aligned}
\tag{32}
$$

Now,

$$
\begin{aligned}
2\eta_k\langle \nabla F(w_k) - \nabla F(w_{k-1}), w_k - w_{k+1}\rangle \leq & 2\eta_k\|\nabla F(w_k) - \nabla F(w_{k-1})\|_{\hat{D}_k}^* \|w_k - w_{k+1}\|_{\hat{D}_k} \\
\leq & \|w_k - w_{k-1}\|_{\hat{D}_k} \|w_k - w_{k+1}\|_{\hat{D}_k} \\
\leq & \frac{1}{2}\|w_k - w_{k-1}\|_{\hat{D}_k}^2 + \frac{1}{2}\|w_k - w_{k+1}\|_{\hat{D}_k}^2,
\end{aligned}
\tag{33}
$$

where the first inequality is due to Cauchy–Schwarz inequality, and the second inequality comes from the choice of learning rate such that $\eta_k \leq \frac{\|w_k - w_{k-1}\|_{\hat{D}_k}}{2\|\nabla F(w_k) - \nabla F(w_{k-1})\|_{\hat{D}_k}^*}$. By plugging (33) into (32) we obtain:

$$
\|w_{k+1} - w_k\|_{\hat{D}_k}^2 \overset{(32)}{\leq} \frac{1}{2}\|w_k - w_{k-1}\|_{\hat{D}_k}^2 + \frac{1}{2}\|w_k - w_{k+1}\|_{\hat{D}_k}^2 + 2\eta_k\langle \nabla F(w_{k-1}), w_k - w_{k+1}\rangle - \\
\|w_{k+1} - w_k\|_{\hat{D}_k}^2.
\tag{34}
$$

Now, we can summarize that

$$
\begin{aligned}
\|w_{k+1} - w^*\|_{\hat{D}_k}^2 \overset{(31)}{\leq}& \|w_{k+1} - w_k\|_{\hat{D}_k}^2 + \|w_k - w^*\|_{\hat{D}_k}^2 - 2\eta_k(F(w_k) - F(w^*)) - \mu\eta_k\|w_k - w^*\|^2 \\
\overset{(34)}{\leq}& \frac{1}{2}\|w_k - w_{k-1}\|_{\hat{D}_k}^2 + \frac{1}{2}\|w_k - w_{k+1}\|_{\hat{D}_k}^2 + 2\eta_k\langle \nabla F(w_{k-1}), w_k - w_{k+1}\rangle - \\
& \|w_{k+1} - w_k\|_{\hat{D}_k}^2 + \|w_k - w^*\|_{\hat{D}_k}^2 - 2\eta_k(F(w_k) - F(w^*)) - \mu\eta_k\|w_k - w^*\|^2 \\
=& \frac{1}{2}\|w_k - w_{k-1}\|_{\hat{D}_k}^2 - \frac{1}{2}\|w_k - w_{k+1}\|_{\hat{D}_k}^2 + 2\eta_k\langle \nabla F(w_{k-1}), w_k - w_{k+1}\rangle + \\
& \|w_k - w^*\|_{\hat{D}_k}^2 - 2\eta_k(F(w_k) - F(w^*)) - \mu\eta_k\|w_k - w^*\|^2 \\
=& \frac{1}{2}\|w_k - w_{k-1}\|_{\hat{D}_{k-1}}^2 + \|w_k - w^*\|_{\hat{D}_{k-1}}^2 - \frac{1}{2}\|w_k - w_{k+1}\|_{\hat{D}_k}^2 + \\
& 2\eta_k\langle \nabla F(w_{k-1}), w_k - w_{k+1}\rangle - 2\eta_k(F(w_k) - F(w^*)) + \\
& \frac{1}{2}\|w_k - w_{k-1}\|_{\hat{D}_k - \hat{D}_{k-1}}^2 + \|w_k - w^*\|_{\hat{D}_k - \hat{D}_{k-1}}^2 - \mu\eta_k\|w_k - w^*\|^2.
\end{aligned}
\tag{35}
$$

Next, let us bound $\langle \nabla F(w_{k-1}), w_k - w_{k+1}\rangle$.

By the `OASIS`'s update rule, $w_{k+1} = w_k - \eta_k \hat{D}_k^{-1} \nabla F(w_k)$, we have,

$$
\begin{aligned}
\langle \nabla F(w_{k-1}), w_k - w_{k+1}\rangle =& \frac{1}{\eta_{k-1}}\langle \hat{D}_{k-1}(w_{k-1} - w_k), w_k - w_{k+1}\rangle \\
=& \frac{1}{\eta_{k-1}}\langle \hat{D}_{k-1}(w_{k-1} - w_k), \eta_k \hat{D}_k^{-1} \nabla F(w_k)\rangle \\
=& \frac{\eta_k}{\eta_{k-1}}(w_{k-1} - w_k)^T \hat{D}_{k-1} \hat{D}_k^{-1} \nabla F(w_k) \\
=& \frac{\eta_k}{\eta_{k-1}}(w_{k-1} - w_k)^T \nabla F(w_k) + \\
& \frac{\eta_k}{\eta_{k-1}}(w_{k-1} - w_k)^T \left( \hat{D}_{k-1} \hat{D}_k^{-1} - I \right) \nabla F(w_k).
\end{aligned}
\tag{36}
$$

The second term in the above equality can be bounded from above as follows:

$$(w_{k-1} - w_k)^T \left( \hat{D}_{k-1} \hat{D}_k^{-1} - I \right) \nabla F(w_k) \le (1 - \beta_2) \frac{2\Gamma}{\alpha} \cdot \|w_{k-1} - w_k\| \cdot \|\nabla F(w_k)\|, \quad (37)$$

where multiplier on the left is obtained via the bound on $\| \cdot \|_\infty$ norm of the diagonal matrix $\hat{D}_{k-1} \hat{D}_k^{-1} - I$:

$$\|\hat{D}_{k-1} \hat{D}_k^{-1} - I\|_\infty = \max_i \left| \left( \hat{D}_{k-1} \hat{D}_k^{-1} - I \right)_i \right| = \max_i \left| (\hat{D}_{k-1} - \hat{D}_k)_i \right| \left| (\hat{D}_k^{-1})_i \right| \le (1 - \beta_2) \frac{2\Gamma}{\alpha},$$

which also represents a bound on the operator norm of the same matrix difference. Next we can use the Young's inequality and $\nabla F(w^*) = 0$ to get

$$(1 - \beta_2) \frac{2\Gamma}{\alpha} \cdot \|w_{k-1} - w_k\| \cdot \|\nabla F(w_k)\| \le (1 - \beta_2) \frac{\Gamma}{\alpha} \left( \|w_{k-1} - w_k\|^2 + L^2 \|w_k - w^*\|^2 \right). \quad (38)$$

By the inequalities (72), (36), (37) and (38), and the definition $\theta_k = \dfrac{\eta_k}{\eta_{k-1}}$ we have:

$$\|w_{k+1} - w^*\|_{\hat{D}_k}^2 + \frac{1}{2} \|w_{k+1} - w_k\|_{\hat{D}_k}^2 + 2\eta_k (1 + \theta_k)(F(w_k) - F(w^*))$$

$$\le \|w_k - w^*\|_{\hat{D}_{k-1}}^2 + \frac{1}{2} \|w_k - w_{k-1}\|_{\hat{D}_{k-1}}^2 + 2\eta_k \theta_k (F(w_{k-1}) - F(w^*))$$

$$+ \left( \frac{1}{2} \|w_k - w_{k-1}\|_{\hat{D}_k - \hat{D}_{k-1}}^2 + (1 - \beta_2) \frac{2\Gamma \theta_k \eta_k}{\alpha} \|w_k - w_{k-1}\|^2 - \mu \eta_k \theta_k \|w_k - w_{k-1}\|^2 \right)$$

$$+ \left( \|w_k - w^*\|_{\hat{D}_k - \hat{D}_{k-1}}^2 + (1 - \beta_2) \frac{2\Gamma L^2 \theta_k \eta_k}{\alpha} \|w_k - w^*\|^2 - \mu \eta_k \|w_k - w^*\|^2 \right)$$

$$\le \|w_k - w^*\|_{\hat{D}_{k-1}}^2 + \frac{1}{2} \|w_k - w_{k-1}\|_{\hat{D}_{k-1}}^2 + 2\eta_k \theta_k (F(w_{k-1}) - F(w^*))$$

$$+ \left( (1 - \beta_2)\Gamma \left( 1 + \frac{2\theta_k \eta_k}{\alpha} \right) - \mu \eta_k \theta_k \right) \|w_k - w_{k-1}\|^2$$

$$+ \left( (1 - \beta_2)\Gamma \left( 2 + \frac{2L^2 \theta_k \eta_k}{\alpha} \right) - \mu \eta_k \right) \|w_k - w^*\|^2.$$

$\square$

**Theorem 4.9.** *Suppose that Assumptions 4.2, 4.4, and 4.3 hold and let $w^*$ be the unique solution for (1). Let $\{w_k\}$ be the iterates generated by Algorithm 1. Then, for all $k \ge 0$ we have: If*

$$\beta_2 \ge \max\{1 - \frac{\alpha^4 \mu^4}{4L^2 \Gamma^2 (\alpha^2 \mu^2 + L\Gamma^2)}, 1 - \frac{\alpha^3 \mu^3}{4L\Gamma(2\alpha^2 \mu^2 + L^3 \Gamma^2)}\}$$

$$\Psi^{k+1} \le (1 - \frac{\alpha^2}{2\Gamma^2 \kappa^2}) \Psi^k, \quad (39)$$

*where*

$$\Psi^{k+1} = \|w_{k+1} - w^*\|_{\hat{D}_k}^2 + \frac{1}{2} \|w_{k+1} - w_k\|_{\hat{D}_k}^2 + 2\eta_k (1 + \theta_k)(F(w_k) - F(w^*)).$$

*Proof.* By Lemma A.3, we have:

$$\|w_{k+1} - w^*\|_{\hat{D}_k}^2 + \frac{1}{2}\|w_{k+1} - w_k\|_{\hat{D}_k}^2 + 2\eta_k(1+\theta_k)(F(w_k) - F(w^*))$$

$$\leq \|w_k - w^*\|_{\hat{D}_{k-1}}^2 + \frac{1}{2}\|w_k - w_{k-1}\|_{\hat{D}_{k-1}}^2 + 2\eta_k\theta_k(F(w_{k-1}) - F(w^*))$$

$$+ \left((1-\beta_2)\Gamma\left(1 + \frac{2\theta_k\eta_k}{\alpha}\right) - \mu\eta_k\theta_k\right)\|w_k - w_{k-1}\|^2$$

$$+ \left((1-\beta_2)\Gamma\left(2 + \frac{2L^2\theta_k\eta_k}{\alpha}\right) - \mu\eta_k\right)\|w_k - w^*\|^2.$$

Lemma 4.8 gives us $\eta_k \in \left[\frac{\alpha}{2L}, \frac{\Gamma}{2\mu}\right]$, so we can choose large enough $\beta_2 \in (0,1)$ such that

$$\left((1-\beta_2)\Gamma\left(1 + \frac{2\theta_k\eta_k}{\alpha}\right) - \mu\eta_k\theta_k\right) \leq 0,$$

$$\left((1-\beta_2)\Gamma\left(2 + \frac{2L^2\theta_k\eta_k}{\alpha}\right) - \mu\eta_k\right) \leq 0.$$

In other words, we have the following bound for $\beta_2$:

$$\beta_2 \geq \max\{1 - \frac{\alpha^4\mu^4}{2L^2\Gamma^2(\alpha^2\mu^2 + L\Gamma^2)}, 1 - \frac{\alpha^3\mu^3}{2L\Gamma(2\alpha^2\mu^2 + L^3\Gamma^2)}\}. \tag{40}$$

However, we can push it one step further, by requiring a stricter inequality to hold, to get a recursion which can allows us to derive a linear convergence rate as follows

$$\left((1-\beta_2)\Gamma\left(1 + \frac{2\theta_k\eta_k}{\alpha}\right) - \mu\eta_k\theta_k\right) \leq -\frac{1}{2}\mu\eta_k\theta_k,$$

$$\left((1-\beta_2)\Gamma\left(2 + \frac{2L^2\theta_k\eta_k}{\alpha}\right) - \mu\eta_k\right) \leq -\frac{1}{2}\mu\eta_k,$$

which requires a correspondingly stronger bound on $\beta_2$:

$$\beta_2 \geq \max\{1 - \frac{\alpha^4\mu^4}{4L^2\Gamma^2(\alpha^2\mu^2 + L\Gamma^2)}, 1 - \frac{\alpha^3\mu^3}{4L\Gamma(2\alpha^2\mu^2 + L^3\Gamma^2)}\}. \tag{41}$$

Combining this with the condition for $\eta_k$ and definition for $\theta_k$, we obtain

$$\left((1-\beta_2)\Gamma\left(1 + \frac{2\theta_k\eta_k}{\alpha}\right) - \mu\eta_k\theta_k\right)\|w_k - w_{k-1}\|^2 \leq -\frac{1}{2}\mu\eta_k\theta_k\|w_k - w_{k-1}\|^2$$

$$\leq -\frac{\alpha^2}{4\Gamma^2\kappa^2}\|w_k - w_{k-1}\|_{\hat{D}_{k-1}}^2,$$

and

$$\left((1-\beta_2)\Gamma\left(2 + \frac{2L^2\theta_k\eta_k}{\alpha}\right) - \mu\eta_k\right)\|w_k - w^*\|^2 \leq -\frac{1}{2}\mu\eta_k\|w_k - w^*\|^2$$

$$\leq -\frac{\alpha}{4\Gamma\kappa}\|w_k - w^*\|_{\hat{D}_{k-1}}^2.$$

where $\kappa = \frac{L}{\mu}$. Using it to supplement the main statement of the lemma, we get

$$\|w_{k+1} - w^*\|_{\hat{D}_k}^2 + \frac{1}{2}\|w_{k+1} - w_k\|_{\hat{D}_k}^2 + 2\eta_k(1+\theta_k)(F(w_k) - F(w^*))$$

$$\leq (1 - \frac{\alpha}{4\Gamma\kappa})\|w_k - w^*\|_{\hat{D}_{k-1}}^2 + \frac{1}{2}(1 - \frac{\alpha^2}{2\Gamma^2\kappa^2})\|w_k - w_{k-1}\|_{\hat{D}_{k-1}}^2 + 2\eta_k\theta_k(F(w_{k-1}) - F(w^*)).$$

Mirroring the result in (Malitsky & Mishchenko, 2020) we get a contraction in all terms, since for function value differences we also can show

$$\frac{\eta_k \theta_k}{\eta_k(1 + \theta_k)} = 1 - \frac{\eta_k}{\eta_k(1 + \theta_k)} \leq 1 - \frac{\alpha}{2\Gamma\kappa}.$$

$\square$

## A.7 PROOF OF THEOREM 4.11

**Theorem 4.11.** *Suppose that Assumptions 4.2, 4.4, and 4.3 hold, and let $F^* = F(w^*)$ where $w^*$ is the unique minimizer. Let $\{w_k\}$ be the iterates generated by Algorithm 1, where $0 < \eta_k = \eta \leq \frac{\alpha^2}{L\Gamma}$, and $w_0$ is a starting point. Then, for all $k \geq 0$ we have:*

$$F(w_k) - F^* \leq (1 - \frac{\eta\mu}{\Gamma})^k [F(w_0) - F^*]. \tag{42}$$

*Proof.* By smoothness of $F(.)$ we have:

$$\begin{aligned}
F(w_{k+1}) &= F(w_k - \eta \hat{D}_k^{-1} \nabla f(w_k)) \\
&\leq F(w_k) + \nabla F(w_k)^T (-\eta \hat{D}_k^{-1} \nabla f(w_k)) + \frac{L}{2} \|\eta \hat{D}_k^{-1} \nabla f(w_k)\|^2 \\
&\leq F(w_k) - \frac{\eta}{\Gamma} \|\nabla F(w_k)\|^2 + \frac{\eta^2 L}{2\alpha^2} \|\nabla F(w_k)\|^2 \\
&= F(w_k) - \eta(\frac{1}{\Gamma} - \frac{\eta L}{2\alpha^2}) \|\nabla F(w_k)\|^2 \tag{43} \\
&\leq F(w_k) - \eta \frac{1}{2\Gamma} \|\nabla F(w_k)\|^2, \tag{44}
\end{aligned}$$

where the first inequality comes from Assumption 4.2, and the second inequality is due to Remark 4.10, and finally, the last inequality is by the choice of $\eta$. Since $F(.)$ is strongly convex, we have $2\mu(F(w_k) - F^*) \leq \|\nabla F(w_k)\|^2$, and therefore,

$$F(w_{k+1}) \leq F(w_k) - \frac{\eta\mu}{\Gamma}(F(w_k) - F^*).$$

Also, we have:

$$F(w_{k+1}) - F^* \leq (1 - \frac{\eta\mu}{\Gamma})(F(w_k) - F^*).$$

$\square$

## A.8 PROOF OF THEOREM 4.13

**Theorem 4.13.** *Suppose that Assumptions 4.3, 4.12 and 4.2 hold. Let $\{w_k\}$ be the iterates generated by Algorithm 1, where $0 < \eta_k = \eta \leq \frac{\alpha^2}{L\Gamma}$, and $w_0$ is a starting point. Then, for all $T > 1$ we have:*

$$\frac{1}{T} \sum_{k=1}^{T} \|\nabla F(w_k)\|^2 \leq \frac{2\Gamma[F(w_0) - \hat{F}]}{\eta T} \xrightarrow{T \to \infty} 0. \tag{45}$$

*Proof.* By starting from (44), we have:

$$F(w_{k+1} \leq F(w_k) - \frac{\eta}{2\Gamma} \|\nabla F(w_k)\|^2. \tag{46}$$

By summing both sides of the above inequality from $k = 0$ to $T - 1$ we have:

$$\sum_{k=0}^{T-1} (F(w_{k+1}) - F(w_k)) \leq - \sum_{k=0}^{T-1} \frac{\eta}{2\Gamma} \|\nabla F(w_k)\|^2.$$

By simplifying the left-hand-side of the above inequality, we have

$$\sum_{k=0}^{T-1} [F(w_{k+1}) - F(w_k)] = F(w_T) - F(w_0) \geq \widehat{F} - F(w_0),$$

where the inequality is due to $\hat{F} \leq F(w_T)$ (Assumption 4.12). Using the above, we have

$$\sum_{k=0}^{T-1} \|\nabla F(w_k)\|^2 \leq \frac{2\Gamma[F(w_0) - \widehat{F}]}{\eta}. \tag{47}$$

$\square$

### A.9 PROOF OF THEOREM 4.17

**Lemma A.4** (see Lemma 1 (Nguyen et al., 2018) or Lemma 2.4 (Gower et al., 2019)). *Assume that $\forall i$ the function $F_i(w)$ is convex and L-smooth. Then, $\forall w \in R^d$ the following hold:*

$$\mathbb{E}_{\mathcal{I}}[\|\nabla F_{\mathcal{I}}(w)\|^2] \leq 4L(F(w) - F(w^*)) + 2\sigma^2. \tag{48}$$

**Theorem A.5.** *Suppose that Assumptions 4.2, 4.3, 4.4, 4.15 and 4.16 hold. Let parameters $\alpha > 0$, $\eta > 0$, $\beta_2 \in (0, 1]$ are chosen such that $\alpha < \frac{2\Gamma L}{\mu}$, $\eta \leq \frac{\alpha}{2L}$, $\beta_2 > 1 - \frac{\eta\mu\alpha}{2\Gamma(\Gamma - \eta\mu)} > 0$. Let $\{w_k\}$ be the iterates generated by Algorithm 1, then, for all $k \geq 0$,*

$$\mathbb{E}[\|w_k - w^*\|_{\hat{D}_k}^2] \leq (1-c)^k \|r_0\|_{\hat{D}_0}^2 + (1 + \frac{2(1-\beta_2)\Gamma}{\alpha})\frac{2\sigma^2\eta^2}{\alpha c}, \tag{49}$$

*where $c = \frac{\eta\mu\alpha - (\Gamma - \eta\mu)2(1-\beta_2)\Gamma}{\Gamma\alpha} \in (0, 1)$.*

**Remark A.6.** *If we choose $\beta_2 := 1 - \frac{\eta\mu\alpha}{4\Gamma(\Gamma - \eta\mu)} > 0$ then*

$$\mathbb{E}[\|w_k - w^*\|_{\hat{D}_k}^2] \leq \left(1 - \frac{\eta\mu}{4\Gamma}\right)^k \|w_0 - w^*\|_{\hat{D}_0}^2 + 4\frac{2\Gamma - \eta\mu}{(\Gamma - \eta\mu)\alpha}\frac{\Gamma}{\mu}\sigma^2\eta.$$

*Proof.* In order to bound $\|r_{k+1}\|_{\hat{D}_{k+1}}^2$ by $\|r_{k+1}\|_{\hat{D}_k}^2$ we will use that

$$0 \prec \hat{D}_{k+1} = \hat{D}_k + \hat{D}_{k+1} - \hat{D}_k \preceq \hat{D}_k + \|\hat{D}_{k+1} - \hat{D}_k\|_\infty I \preceq \hat{D}_k + \|\hat{D}_{k+1} - \hat{D}_k\|_\infty I$$

$$\preceq \hat{D}_k + \|D_{k+1} - D_k\|_\infty I \preceq \hat{D}_k + 2(1-\beta_2)\Gamma I$$

$$\preceq \hat{D}_k + 2(1-\beta_2)\Gamma\frac{1}{\alpha}\hat{D}_k \preceq (1 + \frac{2(1-\beta_2)\Gamma}{\alpha})\hat{D}_k. \tag{50}$$

Then

$$\|r_{k+1}\|_{\hat{D}_{k+1}}^2 \overset{(50)}{\leq} (1 + \frac{2(1-\beta_2)\Gamma}{\alpha})\|r_{k+1}\|_{\hat{D}_k}^2 = (1 + \frac{2(1-\beta_2)\Gamma}{\alpha})\|r_k - \eta\hat{D}_k^{-1}\nabla F_{\mathcal{I}_k}(w_k)\|_{\hat{D}_k}^2$$

$$= (1 + \frac{2(1-\beta_2)\Gamma}{\alpha})\|r_k\|_{\hat{D}_k}^2 - 2\eta(1 + \frac{2(1-\beta_2)\Gamma}{\alpha})\langle r_k, \nabla F_{\mathcal{I}_k}(w_k)\rangle$$

$$+ (1 + \frac{2(1-\beta_2)\Gamma}{\alpha})\eta^2 \left(\|\nabla F_{\mathcal{I}_k}(w_k)\|_{\hat{D}_k}^*\right)^2$$

$$\leq (1 + \frac{2(1-\beta_2)\Gamma}{\alpha})\|r_k\|_{\hat{D}_k}^2 - 2\eta(1 + \frac{2(1-\beta_2)\Gamma}{\alpha})\langle r_k, \nabla F_{\mathcal{I}_k}(w_k)\rangle$$

$$+ (1 + \frac{2(1-\beta_2)\Gamma}{\alpha})\frac{\eta^2}{\alpha}\|\nabla F_{\mathcal{I}_k}(w_k)\|^2. \tag{51}$$

Now, taking an expectation with respect to $\mathcal{I}_k$ conditioned on the past, we obtain

$$\mathbb{E}[\|r_{k+1}\|^2_{\hat{D}_{k+1}}] \overset{(51)}{\leq} (1 + \tfrac{2(1-\beta_2)\Gamma}{\alpha})\|r_k\|^2_{\hat{D}_k} + 2\eta(1 + \tfrac{2(1-\beta_2)\Gamma}{\alpha})\left(F^* - F(w_k) - \tfrac{\mu}{2}\|w_k - w^*\|^2\right)$$

$$+ (1 + \tfrac{2(1-\beta_2)\Gamma}{\alpha})\tfrac{\eta^2}{\alpha}\left(4L(F(w_k) - F^*) + 2\sigma^2\right)$$

$$\leq (1 + \tfrac{2(1-\beta_2)\Gamma}{\alpha})\|r_k\|^2_{\hat{D}_k} + 2\eta(1 + \tfrac{2(1-\beta_2)\Gamma}{\alpha})\left(F^* - F(w_k) - \tfrac{\mu}{2\Gamma}\|r_k\|^2_{\hat{D}_k}\right)$$

$$+ (1 + \tfrac{2(1-\beta_2)\Gamma}{\alpha})\tfrac{\eta^2}{\alpha}\left(4L(F(w_k) - F^*) + 2\sigma^2\right)$$

$$= (1 + \tfrac{2(1-\beta_2)\Gamma}{\alpha})\left(1 - 2\eta\tfrac{\mu}{2\Gamma}\right)\|r_k\|^2_{\hat{D}_k}$$

$$+ (1 + \tfrac{2(1-\beta_2)\Gamma}{\alpha})\left(4L\tfrac{\eta^2}{\alpha} - 2\eta\right)((F(w_k) - F^*)) + (1 + \tfrac{2(1-\beta_2)\Gamma}{\alpha})\tfrac{\eta^2}{\alpha}2\sigma^2$$

$$\leq (1 + \tfrac{2(1-\beta_2)\Gamma}{\alpha})\left(1 - \eta\tfrac{\mu}{\Gamma}\right)\|r_k\|^2_{\hat{D}_k} + (1 + \tfrac{2(1-\beta_2)\Gamma}{\alpha})\tfrac{\eta^2}{\alpha}2\sigma^2, \tag{52}$$

where we used the fact that $2L\tfrac{\eta}{\alpha} - 1 \leq 0 \to \eta \leq \tfrac{\alpha}{2L}$. In order to achieve a convergence we need to have

$$(1 + \tfrac{2(1-\beta_2)\Gamma}{\alpha})\left(1 - \tfrac{\mu\eta}{\Gamma}\right) =: 1 - c < 1.$$

We have

$$(1 + \tfrac{2(1-\beta_2)\Gamma}{\alpha})\left(1 - \tfrac{\eta\mu}{\Gamma}\right) = 1 - \underbrace{\tfrac{\eta\mu\alpha - (\Gamma - \eta\mu)2(1-\beta_2)\Gamma}{\Gamma\alpha}}_{c}$$

Now, we need to choose $\beta_2$ such that

$$1 > \beta_2 > 1 - \tfrac{\eta\mu\alpha}{2\Gamma(\Gamma - \eta\mu)} > 0.$$

With such a choice we can conclude that

$$\mathbb{E}[\|r_k\|^2_{\hat{D}_k}] \overset{(52)}{\leq} (1 - c)\|r_{k-1}\|^2_{\hat{D}_{k-1}} + (1 + \tfrac{2(1-\beta_2)\Gamma}{\alpha})\tfrac{\eta^2}{\alpha}2\sigma^2,$$

$$\leq (1 - c)^k\|r_0\|^2_{\hat{D}_0} + \sum_{i=0}^{k-1}(1 - c)^i(1 + \tfrac{2(1-\beta_2)\Gamma}{\alpha})\tfrac{\eta^2}{\alpha}2\sigma^2,$$

$$\leq (1 - c)^k\|r_0\|^2_{\hat{D}_0} + \sum_{i=0}^{\infty}(1 - c)^i(1 + \tfrac{2(1-\beta_2)\Gamma}{\alpha})\tfrac{\eta^2}{\alpha}2\sigma^2,$$

$$\leq (1 - c)^k\|r_0\|^2_{\hat{D}_0} + (1 + \tfrac{2(1-\beta_2)\Gamma}{\alpha})\tfrac{2\sigma^2\eta^2}{\alpha c}.$$

$\square$

**Theorem 4.17.** *Suppose that Assumptions 4.2, 4.3, 4.4, 4.15 and 4.16 hold. Let $\{w_k\}$ be the iterates generated by Algorithm 1 with $\eta_k = \eta \in (0, \tfrac{\alpha^2\mu}{\Gamma L^2})$, then, for all $k \geq 0$,*

$$\mathbb{E}[F(w_k) - F^*] \leq (1 - c)^k\left(F(w_0) - F^*\right) + \tfrac{\eta^2 L\sigma^2}{c\alpha^2}, \tag{53}$$

*where $c = \tfrac{2\eta\mu}{\Gamma} - \tfrac{2\eta^2 L^2}{\alpha^2} \in (0, 1)$. Moreover, if $\eta_k = \eta \in (0, \tfrac{\alpha^2\mu}{2\Gamma L^2})$ then*

$$\mathbb{E}[F(w_k) - F^*] \leq \left(1 - \tfrac{\eta\mu}{\Gamma}\right)^k\left(F(w_0) - F^*\right) + \tfrac{\eta\Gamma L\sigma^2}{\alpha^2\mu}. \tag{54}$$

*Proof.* First, we will upper-bound the $F(w_{k+1})$ using smoothness of $F(w)$

$$F(w_{k+1}) = F(w_k - \eta\hat{D}_k^{-1}\nabla F_{\mathcal{I}_k}(w_k))$$

$$\leq F(w_k) + \nabla F(w_k)^T(-\eta\hat{D}_k^{-1}\nabla F_{\mathcal{I}_k}(w_k)) + \tfrac{L}{2}\|\eta\hat{D}_k^{-1}\nabla F_{\mathcal{I}_k}(w_k)\|^2$$

$$\leq F(w_k) - \eta\nabla F(w_k)^T\hat{D}_k^{-1}\nabla F_{\mathcal{I}_k}(w_k) + \tfrac{\eta^2 L}{2\alpha^2}\|\nabla F_{\mathcal{I}_k}(w_k)\|^2,$$

where the first inequality is because of Assumptions 4.2 and 4.4, and the second inequality is due to Remark 4.10. By taking the expectation over the sample $\mathcal{I}_k$, we have

$$\mathbb{E}_{\mathcal{I}_k}[F(w_{k+1})] \leq F(w_k) - \eta \mathbb{E}_{\mathcal{I}_k}[\nabla F(w_k)^T \hat{D}_k^{-1} \nabla F_{\mathcal{I}_k}(w_k)] + \frac{\eta^2 L}{2\alpha^2} \mathbb{E}_{\mathcal{I}_k}[\|\nabla F_{\mathcal{I}_k}(w_k)\|^2]$$

$$= F(w_k) - \eta \nabla F(w_k)^T \hat{D}_k^{-1} \nabla F(w_k) + \frac{\eta^2 L}{2\alpha^2} \mathbb{E}_{\mathcal{I}_k}[\|\nabla F_{\mathcal{I}_k}(w_k)\|^2]$$

$$\leq F(w_k) - \frac{\eta}{\Gamma} \|\nabla F(w_k)\|^2 + \frac{\eta^2 L}{2\alpha^2} \left(4L(F(w_k) - F^*) + 2\sigma^2\right)$$

$$\leq F(w_k) + \frac{\eta}{\Gamma} 2\mu \left(F^* - F(w_k)\right) + \frac{\eta^2 L}{2\alpha^2} \left(4L(F(w_k) - F^*) + 2\sigma^2\right) \tag{55}$$

Subtracting $F^*$ from both sides, we obtain

$$\mathbb{E}_{\mathcal{I}_k}[F(w_{k+1}) - F^*] \overset{(55)}{\leq} \left(1 - \frac{\eta}{\Gamma} 2\mu + \frac{\eta^2 L}{2\alpha^2} 4L\right)(F(w_k) - F^*) + \frac{\eta^2 L \sigma^2}{\alpha^2}$$

$$\leq \left(1 - \underbrace{\left(\frac{2\eta\mu}{\Gamma} - \frac{2\eta^2 L^2}{\alpha^2}\right)}_{c}\right)(F(w_k) - F^*) + \frac{\eta^2 L \sigma^2}{\alpha^2}. \tag{56}$$

By taking the total expectation over all batches $\mathcal{I}_0, \mathcal{I}_1, \mathcal{I}_2,...$ and all history starting with $w_0$, we have

$$\mathbb{E}[F(w_k) - F^*] \overset{(56)}{\leq} (1-c)^k (F(w_0) - F^*) + \sum_{i=0}^{k-1} (1-c)^i \frac{\eta^2 L \sigma^2}{\alpha^2}$$

$$\leq (1-c)^k (F(w_0) - F^*) + \sum_{i=0}^{\infty} (1-c)^i \frac{\eta^2 L \sigma^2}{\alpha^2}$$

$$= (1-c)^k (F(w_0) - F^*) + \frac{\eta^2 L \sigma^2}{c\alpha^2}$$

and the (53) is obtained. Now, if $\eta \leq \frac{\alpha^2 \mu}{2\Gamma L^2}$ then

$$c = \frac{2\eta\mu}{\Gamma} - \frac{2\eta^2 L^2}{\alpha^2} \geq 2\eta \left(\frac{\mu}{\Gamma} - \frac{\alpha^2 \mu}{2\Gamma L^2} \frac{L^2}{\alpha^2}\right) = 2\eta \left(\frac{\mu}{\Gamma} - \frac{\mu}{2\Gamma}\right) = \frac{\eta\mu}{\Gamma}$$

and (54) follows. $\qquad\square$

### A.10 PROOF OF THEOREM 4.18

**Theorem 4.18.** *Suppose that Assumptions 4.3, 4.12, 4.2, 4.14 and 4.16 hold. Let $\{w_k\}$ be the iterates generated by Algorithm 1, where $0 < \eta_k = \eta \leq \frac{\eta^2}{L\Gamma}$, and $w_0$ is the starting point. Then, for all $k \geq 0$,*

$$\mathbb{E}\left[\frac{1}{T} \sum_{k=0}^{T-1} \|\nabla F(w_k)\|^2\right] \leq \frac{2\Gamma[F(w_0) - \widehat{F}]}{\eta T} + \frac{\eta \Gamma \gamma^2 L}{\alpha^2} \xrightarrow{T \to \infty} \frac{\eta \Gamma \gamma^2 L}{\alpha^2}.$$

*Proof.* We have that

$$F(w_{k+1}) = F(w_k - \eta \hat{D}_k^{-1} \nabla F_{\mathcal{I}_k}(w_k))$$

$$\leq F(w_k) + \nabla F(w_k)^T (-\eta \hat{D}_k^{-1} \nabla F_{\mathcal{I}_k}(w_k)) + \frac{L}{2} \|\eta \hat{D}_k^{-1} \nabla F_{\mathcal{I}_k}(w_k)\|^2$$

$$\leq F(w_k) - \eta \nabla F(w_k)^T \hat{D}_k^{-1} \nabla F_{\mathcal{I}_k}(w_k) + \frac{\eta^2 L}{2\alpha^2} \|\nabla F_{\mathcal{I}_k}(w_k)\|^2,$$

where the first inequality is because of Assumptions 4.2 and 4.4, and the second inequality is due to Remark 4.10. By taking the expectation over the sample $\mathcal{I}_k$, we have

$$\mathbb{E}_{\mathcal{I}_k}[F(w_{k+1})] \leq F(w_k) - \eta \mathbb{E}_{\mathcal{I}_k}[\nabla F(w_k)^T \hat{D}_k^{-1} \nabla F_{\mathcal{I}_k}(w_k)] + \frac{\eta^2 L}{2\alpha^2} \mathbb{E}_{\mathcal{I}_k}[\|\nabla F_{\mathcal{I}_k}(w_k)\|^2]$$

$$= F(w_k) - \eta \nabla F(w_k)^T \hat{D}_k^{-1} \nabla F(w_k) + \frac{\eta^2 L}{2\alpha^2} \mathbb{E}_{\mathcal{I}_k}[\|\nabla F_{\mathcal{I}_k}(w_k)\|^2]$$

$$\leq F(w_k) - \eta \left( \frac{1}{\Gamma} - \frac{\eta L}{2\alpha^2} \right) \|\nabla F(w_k)\|^2 + \frac{\eta^2 \gamma^2 L}{2\alpha^2} \tag{57}$$

$$\leq F(w_k) - \frac{\eta}{2\Gamma} \|\nabla F(w_k)\|^2 + \frac{\eta^2 \gamma^2 L}{2\alpha^2}, \tag{58}$$

where the second inequality is due to Remark 4.10 and Assumption 4.14, and the third inequality is due to the choice of the step length.

By inequality (58) and taking the total expectation over all batches $\mathcal{I}_0$, $\mathcal{I}_1$, $\mathcal{I}_2$,... and all history starting with $w_0$, we have

$$\mathbb{E}[F(w_{k+1}) - F(w_k)] \leq -\frac{\eta}{2\Gamma} \mathbb{E}[\|\nabla F(w_k)\|^2] + \frac{\eta^2 \gamma^2 L}{2\alpha^2}.$$

By summing both sides of the above inequality from $k = 0$ to $T - 1$,

$$\sum_{k=0}^{T-1} \mathbb{E}[F(w_{k+1}) - F(w_k)] \leq -\frac{\eta}{2\Gamma} \sum_{k=0}^{T-1} \mathbb{E}[\|\nabla F(w_k)\|^2] + \frac{\eta^2 \gamma^2 LT}{2\alpha^2}$$

$$= -\frac{\eta}{2\Gamma} \mathbb{E}\left[ \sum_{k=0}^{T-1} \|\nabla F(w_k)\|^2 \right] + \frac{\eta^2 \gamma^2 LT}{2\alpha^2}.$$

By simplifying the left-hand-side of the above inequality, we have

$$\sum_{k=0}^{T-1} \mathbb{E}\left[ F(w_{k+1}) - F(w_k) \right] = \mathbb{E}[F(w_T)] - F(w_0) \geq \hat{F} - F(w_0),$$

where the inequality is due to $\hat{F} \leq F(w_T)$ (Assumption 4.12). Using the above, we have

$$\mathbb{E}\left[ \sum_{k=0}^{T-1} \|\nabla F(w_k)\|^2 \right] \leq \frac{2\Gamma[F(w_0) - \hat{F}]}{\eta} + \frac{\eta \Gamma \gamma^2 LT}{\alpha^2}.$$

$\square$

### A.11 PROOFS WITH LINE SEARCH

Given the current iterate $w_k$, the steplength is chosen to satisfy the following sufficient decrease condition

$$F(w_k + \eta_k p_k) \leq F(w_k) - c_1 \eta_k \nabla F(w_k)^T \hat{D}_k^{-1} \nabla F(w_k), \tag{59}$$

where $p_k = -\hat{D}_k^{-1} \nabla F(w_k)$ and $c_1 \in (0, 1)$. The mechanism works as follows. Given an initial steplength (say $\eta_k = 1$), the function is evaluated at the trial point $w_k + \alpha_k p_k$ and condition (59) is checked. If the trial point satisfies (59), then the step is accepted. If the trial point does not satisfy (59), the steplength is reduced (e.g., $\eta_k = \tau \eta_k$ for $\tau \in (0, 1)$). This process is repeated until a steplength that satisfies (59) is found.

---

**Algorithm 2** Backtracking Armijo Linesearch (Nocedal & Wright, 2006)

---

**Input:** $w_k$, $p_k$

1: Select $\eta_{\text{initial}}$, $c_1 \in (0, 1)$ and $\tau \in (0, 1)$
2: $\eta^0 = \eta_{\text{initial}}$, $j = 0$
3: **while** $F(w_k + \eta_k p_k) > F(w_k) + c_1 \eta_k \nabla F(w_k)^T p_k$ **do**
4:     Set $\eta^{j+1} = \tau \eta^j$
5:     Set $j = j + 1$
6: **end while**

**Output:** $\eta_k = \eta^j$

---

By following from the study (Berahas et al., 2019), we have the following theorems.

### A.11.1 DETERMINITIC REGIME - STRONGLY CONVEX

**Theorem A.7.** *Suppose that Assumptions 4.3 and 4.2 and 4.4 hold. Let $\{w_k\}$ be the iterates generated by Algorithm 1, where $\eta_k$ is the maximum value in $\{\tau^{-j} : j = 0, 1, \dots\}$ satisfying (59) with $0 < c_1 < 1$, and $w_0$ is the starting point. Then for all $k \geq 0$,*

$$F(w_k) - F^\star \leq \left(1 - \frac{4\mu\alpha^2 c_1(1 - c_1)\tau}{\Gamma^2 L}\right)^k [F(w_0) - F^\star].$$

*Proof.* Starting with (43) we have

$$F(w_k - \eta_k \hat{D}_k^{-1} \nabla F(w_k)) \leq F(w_k) - \eta_k \left(\frac{1}{\Gamma} - \eta_k \frac{L}{2\alpha^2}\right) \|\nabla F(w_k)\|^2. \tag{60}$$

From the Armijo backtracking condition (59), we have

$$F(w_k - \eta_k \hat{D}_k^{-1} \nabla F(w_k)) \leq F(w_k) - c_1 \eta_k \nabla F(w_k)^T \hat{D}_k^{-1} \nabla F(w_k)$$

$$\leq F(w_k) - \frac{c_1 \eta_k}{\Gamma} \|\nabla F(w_k)\|^2. \tag{61}$$

Looking at (60) and (61), it is clear that the Armijo condition is satisfied when

$$\eta_k \leq \frac{2\alpha^2(1 - c_1)}{\Gamma L}. \tag{62}$$

Thus, any $\eta_k$ that satisfies (62) is guaranteed to satisfy the Armijo condition (59). Since we find $\eta_k$ using a constant backtracking factor of $\tau < 1$, we have that

$$\eta_k \geq \frac{2\alpha^2(1 - c_1)\tau}{\Gamma L}. \tag{63}$$

Therefore, from (60) and by (62) and (63) we have

$$F(w_{k+1}) \leq F(w_k) - \eta_k \left(\frac{1}{\Gamma} - \frac{\eta_k L}{2\alpha^2}\right) \|\nabla F(w_k)\|^2$$

$$\leq F(w_k) - \frac{\eta_k c_1}{\Gamma} \|\nabla F(w_k)\|^2$$

$$\leq F(w_k) - \frac{2\alpha^2 c_1(1 - c_1)\tau}{\Gamma^2 L} \|\nabla F(w_k)\|^2. \tag{64}$$

By strong convexity, we have $2\mu(F(w) - F^\star) \leq \|\nabla F(w)\|^2$, and thus

$$F(w_{k+1}) \leq F(w_k) - \frac{4\mu\alpha^2 c_1(1 - c_1)\tau}{\Gamma^2 L}(F(w) - F^\star). \tag{65}$$

Subtracting $F^\star$ from both sides, and applying (65) recursively yields the desired result. $\qquad\square$

### A.11.2 DETERMINISTIC REGIME - NONCONVEX

**Theorem A.8.** *Suppose that Assumptions 4.3, 4.12 and 4.2 hold. Let $\{w_k\}$ be the iterates generated by Algorithm 1, where $\eta_k$ is the maximum value in $\{\tau^{-j} : j = 0, 1, \dots\}$ satisfying (59) with $0 < c_1 < 1$, and where $w_0$ is the starting point. Then,*

$$\lim_{k \to \infty} \|\nabla F(w_k)\| = 0, \tag{66}$$

*and, moreover, for any $T > 1$,*

$$\frac{1}{T} \sum_{k=0}^{T-1} \|\nabla F(w_k)\|^2 \leq \frac{\Gamma^2 L[F(w_0) - \widehat{F}]}{2\alpha^2 c_1(1 - c_1)\tau T} \xrightarrow{\tau \to \infty} 0.$$

*Proof.* We start with (64)

$$F(w_{k+1}) \leq F(w_k) - \frac{2\alpha^2 c_1(1 - c_1)\tau}{\Gamma^2 L} \|\nabla F(w_k)\|^2.$$

Summing both sides of the above inequality from $k = 0$ to $T - 1$,

$$\sum_{k=0}^{T-1} (F(w_{k+1}) - F(w_k)) \leq -\sum_{k=0}^{T-1} \frac{2\alpha^2 c_1(1 - c_1)\tau}{\Gamma^2 L} \|\nabla F(w_k)\|^2.$$

The left-hand-side of the above inequality is a telescopic sum and thus,

$$\sum_{k=0}^{T-1} [F(w_{k+1}) - F(w_k)] = F(w_T) - F(w_0) \geq \widehat{F} - F(w_0),$$

where the inequality is due to $\hat{F} \leq F(w_T)$ (Assumption 4.12). Using the above, we have

$$\sum_{k=0}^{T-1} \|\nabla F(w_k)\|^2 \leq \frac{\Gamma^2 L[F(w_0) - \widehat{F}]}{2\alpha^2 c_1(1 - c_1)\tau}. \tag{67}$$

Taking limits we obtain,

$$\lim_{\tau \to \infty} \sum_{k=0}^{\tau-1} \|\nabla F(w_k)\|^2 < \infty,$$

which implies (66). Dividing (67) by $T$ we conclude

$$\frac{1}{T} \sum_{k=0}^{T-1} \|\nabla F(w_k)\|^2 \leq \frac{\Gamma^2 L[F(w_0) - \widehat{F}]}{2\alpha^2 c_1(1 - c_1)\tau T}.$$

□

## A.12 PROOFS WITH DECAYING STEP-SIZE

In this section, we provide the analysis with decaying stepsize for both strongly convex and nonconvex cases in the stochastic regime. The proofs for the following theorems follows from the proofs in Theorems 4.17 and 4.18, and Theorems 4.7 and 4.9 in Bottou et al. (2018).

### A.12.1 STOCHASTIC REGIME – STRONGLY CONVEX

**Theorem A.9.** *Suppose that Assumptions 4.2, 4.3, 4.4, 4.15 and 4.16 hold., and let $F^\star = F(w^\star)$, where $w^\star$ is the minimizer of $F$. Let $\{w_k\}$ be the iterates generated by Algorithm 1, where $\eta_k$ is a sequence of stepsize such that, for all $k \geq 0$,*

$$\eta_k = \frac{\phi}{\zeta + k} \quad \text{for some } \phi > \frac{\Gamma}{\mu} \text{ and } \zeta > 0 \text{ such that } \eta_0 \leq \frac{\alpha^2 \mu}{2\Gamma L^2} \tag{68}$$

*Then, for all $k \geq 0$, the expected optimality gap satisfies*

$$\mathbb{E}[F(w_k) - F^\star] \leq \frac{\nu}{\zeta + k} \tag{69}$$

*where*

$$\nu := \max\left\{ \frac{\phi^2 L \sigma^2}{(\frac{\phi\mu}{\Gamma} - 1)\alpha^2}, \zeta(F(w_0) - F^\star) \right\} \tag{70}$$

*Proof.* By taking total expectation from (56), and replacing $\eta$ with $\eta_k$ we have:

$$\mathbb{E}[F(w_{k+1}) - F^*] \leq \left( 1 - \underbrace{\left( \frac{2\eta_k \mu}{\Gamma} - \frac{2\eta_k^2 L^2}{\alpha^2} \right)}_{c} \right) \mathbb{E}[F(w_k) - F^*] + \frac{\eta_k^2 L \sigma^2}{\alpha^2}. \tag{71}$$

By the assumption $\eta_0 \leq \frac{\alpha^2 \mu}{2\Gamma L^2}$, and the way that $\eta_k$ is designed, i.e., in a decaying way, we have $\eta_k < \eta_0 \leq \frac{\alpha^2 \mu}{2\Gamma L^2}$, therefore

$$c = \frac{2\eta_k \mu}{\Gamma} - \frac{2\eta_k^2 L^2}{\alpha^2} \geq 2\eta_k \left( \frac{\mu}{\Gamma} - \frac{\alpha^2 \mu}{2\Gamma L^2} \frac{L^2}{\alpha^2} \right) = 2\eta_k \left( \frac{\mu}{\Gamma} - \frac{\mu}{2\Gamma} \right) = \frac{\eta_k \mu}{\Gamma}.$$

Thus, we have:

$$\mathbb{E}[F(w_{k+1}) - F^*] \leq \left( 1 - \frac{\eta_k \mu}{\Gamma} \right) \mathbb{E}[F(w_k) - F^*] + \frac{\eta_k^2 L \sigma^2}{\alpha^2}. \tag{72}$$

We do the rest of the analysis of (69) with induction. First, for $k = 0$, (69) holds by the definition of $\nu$. Next, assuming (69) holds for $k \geq 0$, by (72) we have:

$$
\begin{aligned}
\mathbb{E}[F(w_{k+1}) - F^\star] &\leq \left(1 - \frac{\frac{\phi\mu}{\Gamma}}{\hat{k}}\right) \frac{\nu}{\hat{k}} + \frac{\phi^2 L \sigma^2}{\hat{k}^2 \alpha^2} \\
&= \left(\frac{\hat{k} - \frac{\phi\mu}{\Gamma}}{\hat{k}^2}\right) \nu + \frac{\phi^2 L \sigma^2}{\hat{k}^2 \alpha^2} \\
&= \left(\frac{\hat{k} - 1}{\hat{k}^2}\right) \nu \underbrace{- \left(\frac{\frac{\phi\mu}{\Gamma} - 1}{\hat{k}^2}\right) \nu + \frac{\phi^2 L \sigma^2}{\hat{k}^2 \alpha^2}}_{\text{nonpositive by definition of } \nu} \\
&\leq \frac{\nu}{\hat{k} + 1}
\end{aligned}
\tag{73}
$$

$\square$

### A.12.2 STOCHASTIC REGIME – NONCONVEX

**Theorem A.10.** *Suppose that Assumptions 4.3, 4.12, 4.2, 4.14 and 4.16 hold. Let $\{w_k\}$ be the iterates generated by Algorithm 1, where $\eta_k$ is a steplength sequence satisfying*

$$
\sum_{k=0}^{\infty} \eta_k = \infty \quad \text{and} \quad \sum_{k=0}^{\infty} \eta_k^2 < \infty.
\tag{74}
$$

*and $w_0$ is the starting point. Then, with $A_T := \sum_{k=0}^{T-1} \eta_k$,*

$$
\lim_{T \to \infty} \mathbb{E}[\sum_{k=0}^{T-1} \eta_k \|\nabla F(w_k)\|^2] < \infty,
\tag{75}
$$

*and therefore* $\mathbb{E}[\frac{1}{A_T} \sum_{k=0}^{T-1} \eta_k \|\nabla F(w_k)\|^2] \xrightarrow{T \to \infty} 0.$
$\tag{76}$

*Proof.* The steplength sequence $\{\eta_k\}$ goes to zero due to the second condition in (74). meaning that w.l.o.g., we may assume that $\eta_k \leq \frac{2\alpha^2}{2\Gamma L}$ for all $k \geq 0$. By starting from and taking total expectation from (57), we have

$$
\begin{aligned}
\mathbb{E}[F(w_{k+1})] - \mathbb{E}[F(w_k)] &\leq -\eta_k \left(\frac{1}{\Gamma} - \frac{\eta_k L}{2\alpha^2}\right) \mathbb{E}[\|\nabla F(w_k)\|^2] + \frac{\eta_k^2 \gamma^2 L}{2\alpha^2} \\
&\leq -\frac{\eta_k}{2\Gamma} \mathbb{E}[\|\nabla F(w_k)\|^2] + \frac{\eta_k^2 \gamma^2 L}{2\alpha^2}.
\end{aligned}
$$

Summing both sides of this inequality for $k \in \{0, \ldots, T-1\}$ gives:

$$
\widehat{F} - \mathbb{E}[F(w_0)] \leq \mathbb{E}[F(w_T)] - \mathbb{E}[F(w_0)] \leq -\frac{1}{2\Gamma} \sum_{k=0}^{T-1} \eta_k \mathbb{E}[\|\nabla F(w_k)\|^2] + \frac{\gamma^2 L}{2\alpha^2} \sum_{k=0}^{T-1} \eta_k^2.
\tag{77}
$$

Multiplying by $2\Gamma$, and rearranging we get

$$
\sum_{k=0}^{T-1} \eta_k \mathbb{E}[\|\nabla F(w_k)\|^2] \leq 2\Gamma(\mathbb{E}[F(w_0)] - \widehat{F}) + \frac{\Gamma \gamma^2 L}{\alpha^2} \sum_{k=0}^{T-1} \eta_k^2.
\tag{78}
$$

The second condition in (74) guarantees that the right-hand side of the above inequality converges to a finite limit when $T$ increases, therefore, it implies (75). By the first condition in (74), we conclude that $A_T \to \infty$ as $T \to \infty$, which results in (76). $\square$

### A.13 CONVERGENCE BOUNDS INTERPRETATION

In this section, we provide the interpretation behind the convergence bounds in our theoretical results.

- In Theorem 4.6 there is the term $(1 - \beta_2)\Gamma\dfrac{Q_k}{k}$, which essentially highlights that the parameter $\beta_2$ controls the size of the neighborhood of optimal solution(s). In other words, if $\beta_2$ is very close to 1 (which is the case for our algorithm), the convergence neighborhood would be very small. Meaning that, with the assumption regarding bounded iterates, our algorithm converges to a very small neighborhood of optimal solution as $\beta_2$ is very close to 1 (while in SGD the step-size controls the size of this neighborhood and accordingly, the step-size needs to go zero in order to converge to the small neighborhood of the optimal solution(s)). Note this convergence guarantee is for smooth and convex case with **adaptive learning rate**.

- Theorem 4.9 provides a linear convergence rate for the smooth and strongly convex case with **adaptive learning rate**. The analysis behind this theorem is based on the Lyapunov energy that decreases linearly.

- Remark 4.10 guarantees that the OASIS preconditioning matrix is positive definite and its eigenvalues are uniformly bounded above and below.

- Theorems 4.11 and 4.13 provides linear and sublinear rate to the stationary point(s) for deterministinc strongly convex and nonconvex cases, respectively. The learning rate is fixed in both mentioned cases, and the convergence rates are similar to those of limited-memory quasi-Newton approaches (e.g., L-BFGS).

- Theorems 4.17 and 4.18 provides linear and sublinear rate to the neighborhood of stationary point(s) (in expectation) for deterministinc strongly convex and nonconvex cases , respectively. Please note that we provided another convergence guarantee for the stochastic strongly convex case (please see Theorem A.5). Again, as the deterministic cases, the provided convergence guarantees completely match with those of the classical quasi-Newton methods (such as L-BFGS); the learning rate is fixed in the latter cases. Please note that these methods (such as OASIS and classical limited-memory quasi-Newton methods) in theory are not better than GD-type methods. In practice, however, they have shown their strength.

- Theorems A.9 and A.10 provides convergence guarantees for the stochastic setting with decaying learning rate. The mentioned theorems provides the convergence guarantees to the stationary point(s) (in expectation) for the stochastic strongly convex and nonconvex cases.

- Theorems A.7 and A.8 provides linear and sublinear convergence rates to the stationary point(s) for the deterministic strongly convex and nonconvex cases with utilizing linesearch.

## B ADDITIONAL ALGORITHM DETAILS

### B.1 RELATED WORK

As was mentioned in Section 2, we follow the generic iterate update:

$$w_{k+1} = w_k - \eta_k \hat{D}_k^{-1} m_k.$$

Table 3 summarizes the methodologies discussed in Section 2.

### B.2 STOCHASTIC OASIS

Here we describe a stochastic variant of OASIS in more detail. As mentioned in Section 3, to estimate gradient and Hessian diagonal, the choices of sets $\mathcal{I}_k$, $\mathcal{J}_k \subset [n]$, are independent and correspond to only a fraction of data. This change results in Algorithm 3.

Table 3: Summary of Algorithms Discussed in Section 2

| Algorithm | $m_k$ | $\hat{D}_k$ |
|---|---|---|
| SGD (Robbins & Monro, 1951) | $\beta_1 m_{t-1} + (1-\beta_1)g_k$ | $1$ |
| Adagrad (Duchi et al., 2011) | $g_k$ | $\sqrt{\sum_{i=1}^k \texttt{diag}(g_i \odot g_i)}$ |
| RMSProp (Tieleman & Hinton, 2012) | $g_k$ | $\sqrt{\beta_2 \hat{D}_{k-1}^2 + (1-\beta_2)\texttt{diag}(g_k \odot g_k)}$ |
| Adam (Kingma & Ba, 2014) | $\dfrac{(1-\beta_1)\sum_{i=1}^k \beta_1^{k-i}g_i}{1-\beta_1^k}$ | $\sqrt{\dfrac{(1-\beta_2)\sum_{i=1}^k \beta_2^{k-i}\texttt{diag}(g_i \odot g_i)}{1-\beta_2^k}}$ |
| AdaHessian (Yao et al., 2020) | $\dfrac{(1-\beta_1)\sum_{i=1}^k \beta_1^{k-i}g_i}{1-\beta_1^k}$ | $\sqrt{\dfrac{(1-\beta_2)\sum_{i=1}^k \beta_2^{k-i}v_i^2}{1-\beta_2^k}}$ |
| OASIS | $g_k$ | $|\beta_2 D_{k-1} + (1-\beta_2)v_k|_\alpha$ |

    * $v_i = \texttt{diag}(z_i \odot \nabla^2 F(w_i)z_i)$ and $z_i \sim$ Rademacher$(0.5)$ $\forall i \geq 1$,

    ** $(|A|_\alpha)_{ii} = \max\{|A|_{ii}, \alpha\}$

    *** $D_k = \beta_2 D_{k-1} + (1-\beta_2)v_k$

---

**Algorithm 3** Stochastic OASIS

**Input:** $w_0, \eta_0, \mathcal{I}_k, D_0, \theta_0 = +\infty, \beta_2, \alpha$

  1: $w_1 = w_0 - \eta_0 \hat{D}_0^{-1}\nabla F(w_0)_{\mathcal{I}_k}$
  2: **for** $k = 1, 2, \ldots$ **do**
  3:    Calculate $D_k = \beta_2 D_{k-1} + (1-\beta_2)\,\texttt{diag}(z_k \odot \nabla^2 F_{\mathcal{J}_k}(w_k)z_k)$
  4:    Calculate $\hat{D}_k$ by setting $(\hat{D}_k)_{i,i} = \max\{|D_k|_{i,i}, \alpha\}, \;\; \forall i \in [d]$
  5:    Update $\eta_k = \min\{\sqrt{1 + \theta_{k-1}}\eta_{k-1}, \frac{\|w_k - w_{k-1}\|_{\hat{D}_k}}{2\|\nabla F_{\mathcal{I}_k}(w_k) - \nabla F_{\mathcal{I}_k}(w_{k-1})\|_{\hat{D}_k}^*}\}$
  6:    Set $w_{k+1} = w_k - \eta_k \hat{D}_k^{-1}\nabla F_{\mathcal{I}_k}(w_k)$
  7:    Set $\theta_k = \dfrac{\eta_k}{\eta_{k-1}}$
  8: **end for**

---

Additionally, in order to compare the performance of our preconditioner schema independent of the adaptive learning-rate rule, we also consider variants of OASIS with fixed $\eta$. We explore two modifications, denoted as "Fixed LR" and "Momentum" in Section 5. "Fixed LR" is obtained from Algorithm 3 by simply having a fixed scalar $\eta$ for all iterations, which results in Algorithm 4. "Momentum" is obtained from "Fixed LR" by considering a simple form of first-order momentum with a parameter $\beta_1$, and this results in Algorithm 5. In Section C.5, we show that OASIS is **robust** with respect to the different choices of learning rate, obtaining a narrow spectrum of changes.

---

**Algorithm 4** OASIS- Fixed LR

**Input:** $w_0, \eta, \mathcal{I}_k, D_0, \beta_2, \alpha$

  1: $w_1 = w_0 - \eta \hat{D}_0^{-1}\nabla F_{\mathcal{I}_k}(w_0)$
  2: **for** $k = 1, 2, \ldots$ **do**
  3:    Calculate $D_k = \beta_2 D_{k-1} + (1-\beta_2)\,\texttt{diag}(z_k \odot \nabla^2 F_{\mathcal{J}_k}(w_k)z_k)$

  4:    Calculate $\hat{D}_k$ by setting $(\hat{D}_k)_{i,i} = \max\{|D_k|_{i,i}, \alpha\}, \;\; \forall i \in [d]$
  5:    Set $w_{k+1} = w_k - \eta \hat{D}_k^{-1}\nabla F_{\mathcal{I}_k}(w_k)$
  6: **end for**

---

**Algorithm 5** `OASIS`- Momentum

---

**Input:** $w_0, \eta, \mathcal{I}_k, D_0, \beta_1, \beta_2\ \alpha$

1: Set $m_0 = \nabla F_{\mathcal{I}_k}(w_0)$
2: $w_1 = w_0 - \eta \hat{D}_0^{-1} m_0$
3: **for** $k = 1, 2, \dots$ **do**
4:    Calculate $D_k = \beta_2 D_{k-1} + (1-\beta_2)\, \texttt{diag}(z_k \odot \nabla^2 F_{\mathcal{J}_k}(w_k) z_k)$

5:    Calculate $\hat{D}_k$ by setting $(\hat{D}_k)_{i,i} = \max\{|D_k|_{i,i}, \alpha\}, \ \ \forall i \in [d]$
6:    Calculate $m_k = \beta_1 m_{k-1} + (1-\beta_1)\nabla F_{\mathcal{I}_k}(w_k)$
7:    Set $w_{k+1} = w_k - \eta \hat{D}_k^{-1} m_k$
8: **end for**

---

In our experiments, we used a biased version of the algorithm with $\mathcal{I}_k = \mathcal{J}_k$. One of the benefits of this choice is to compute the Hessian-vector product efficiently. By reusing the computed gradient, the overhead of computing gradients with respect to different samples is significantly reduced (see Section B.3).

The final remark is related to a strategy to obtain $D_0$, which is required at the start of `OASIS` Algorithm. One option is to do warmstarting, i.e., spend some time before training in order to sample some number of Hutchinson's estimates. The second option is to use bias corrected rule for $D_k$ similar to the one used in Adam and AdaHessian

$$D_k = \beta_2 D_{k-1} + (1-\beta_2)\texttt{diag}(z_k \odot \nabla^2 F_{\mathcal{J}_k}(w_k)z_k),$$

$$D_k^{cor} = \frac{D_k}{1 - \beta_2^{k+1}},$$

which allows to obtain $D_0$ by defining $D_{-1}$ to be a zero diagonal.

### B.3 EFFICIENT HESSIAN-VECTOR COMPUTATION

In the `OASIS` Algorithm, similar to AdaHessian (Yao et al., 2020) methodology, the calculation of the Hessian-vector product in Hutchinson's method is the main overhead. In this section, we present how this product can be calculated efficiently. First, lets focus on two popular machine learning problems: $(i)$ logistic regression; and $(ii)$ non-linear least squares problems. Then, we show the efficient calculation of this product in deep learning problems.

**Logistic Regression.**    One can note that we can show the $\ell_2$-regularized logistic regression as:

$$F(w) = \frac{1}{n}\left(\mathbb{1}_n * \log\left(1 + e^{-Y \odot X^T w}\right)\right) + \frac{\lambda}{2}\|w\|^2, \tag{79}$$

where $\mathbb{1}_n$ is the vector of ones with size $1 \times n$, $*$ is the standard multiplication operator between two matrices, and $X$ is the feature matrix and $Y$ is the label matrix. Further, the Hessian-vector product for logistic regression problems can be calculated as follows (for any vector $v \in \mathbb{R}^d$):

$$\nabla^2 F(w) * v = \frac{1}{n}\left( \underbrace{X^T * \underbrace{\left[\frac{Y \odot Y \odot e^{-Y \odot X^T w}}{(1 + e^{-Y \odot X^T w})^2}\right] \odot \underbrace{X * v}_{①}}_{②}}_{③} \right) + \lambda v. \tag{80}$$

The above calculation shows the order of computations in order to compute the Hessian-vector product without constructing the Hessian explicitly.

**Non-linear Least Squares.**    The non-linear least squares problems can be written as following:

$$F(w) = \frac{1}{2n}\|Y - \phi(X^T w)\|^2. \tag{81}$$

The Hessian-vector product for the above problem can be efficiently calculated as:

$\nabla^2 F(w) * v =$

$$\frac{1}{n} \left( -X^T * \underbrace{\left[ \underbrace{\phi(X^T w) \odot (1 - \phi(X^T w)) \odot (Y - 2(1+Y) \odot \phi(X^T w) + 3\phi(X^T w) \odot \phi(X^T w))}_{\text{②}} \right] \odot \underbrace{X * v}_{\text{①}}}_{\text{③}} \right). \quad (82)$$

**Deep Learning.** In general, the Hessian-vector product can be efficiently calculated by:

$$\nabla^2 F(w) * v = \frac{\partial^2 F(w)}{\partial w \partial w} * v = \frac{\partial}{\partial w} \left( \frac{\partial F(w)^T}{\partial w} v \right). \quad (83)$$

As is clear from (83), two rounds of back-propagation are needed. In fact, the round regarding the gradient evaluation is already calculated in the corresponding iteration; and thus, one extra round of back-propagation is needed in order to evaluate the Hessian-vector product. This means that the extra cost for the above calculation is almost equivalent to one gradient evaluation.

Table 5: Deep Neural Networks used in the experiments.

| Data | Network | # Train | # Test | # Classes | $d$ |
|------|---------|---------|--------|-----------|-----|
| **MNIST** | **Net DNN** | 60K | 10K | 10 | 21.8K |
| **CIFAR10** | **ResNet20** | 50K | 10K | 10 | 272K |
| | **ResNet32** | 50K | 10K | 10 | 467K |
| **CIFAR100** | **ResNet18** | 50K | 10K | 100 | 11.22M |

# C  DETAILS OF EXPERIMENTS

## C.1  TABLE OF ALGORITHMS

Table 4 summarizes the algorithms implemented in Section 5.

| Algorithm | Description and Reference |
|-----------|---------------------------|
| SGD | Stochastic gradient method (Robbins & Monro, 1951) |
| Adam | Adam method (Kingma & Ba, 2014) |
| AdamW | Adam with decoupled weight decay (Loshchilov & Hutter, 2017) |
| AdGD | Adaptive Gradient Descent (Malitsky & Mishchenko, 2020) |
| AdaHessian | AdaHessian method (Yao et al., 2020) |
| OASIS-Adaptive LR | Our proposed method with adaptive learning rate |
| OASIS-Fixed LR | Our proposed method with fixed learning rate |
| OASIS-Momentum | Our proposed method with momentum |

Table 4: Description of implemented algorithms

To display the optimality gap for logistic regression problems, we used Trust Region (TR) Newton Conjugate Gradient (Newton-CG) method (Nocedal & Wright, 2006) to find a $w$ such that $\|\nabla F(w)\|^2 < 10^{-19}$. Hereafter, we denote $F(w) - F(w^*)$ the optimality gap, where we refer $w^*$ to the solution found by TR Newton-CG.

## C.2  PROBLEM DETAILS

Some metrics for the image classification problems are given in Table 5.

The number of parameters for ResNet architectures is particularly important, showcasing that OASIS is able to operate in very high-dimensional problems, contrary to the widespread belief that methods using second-order information are fundamentally limited by the dimensionality of the parameter space.

## C.3  BINARY CLASSIFICATION

In Section 5 of the main paper, we studied the empirical performance of OASIS on binary classification problems in a deterministic setting, and compared it with AdGD and AdaHessian. Here, we present the extended details on the experiment.

### C.3.1  PROBLEM AND DATA

As a common practice for the empirical research of the optimization algorithms, *LIBSVM* datasets[6] are chosen for the exercise. Specifically, we chose 5 popular binary class datasets, *ijcnn1, rcv1, news20, covtype* and *real-sim*. Table 6 summarizes the basic statistics of the datasets.

---

[6]Datasets are available at `https://www.csie.ntu.edu.tw/~cjlin/libsvmtools/datasets/`.

Table 6: Summary of Datasets.

| Dataset | # feature | $n$ (# Train) | # Test | % Sparsity |
|---|---|---|---|---|
| *ijcnn1*[1] | 22 | 49,990 | 91,701 | 40.91 |
| *rcv1*[1] | 47,236 | 20,242 | 677,399 | 99.85 |
| *news20*[2] | 1,355,191 | 14,997 | 4,999 | 99.97 |
| *covtype*[2] | 54 | 435,759 | 145,253 | 77.88 |
| *real-sim*[2] | 20,958 | 54,231 | 18,078 | 99.76 |

[1] dataset has default training/testing samples.
[2] dataset is randomly split by 75%-training & 25%-testing.

Let $(x_i, y_i)$ be a training sample indexed by $i \in [n] := \{1, 2, ..., n\}$, where $x_i \in \mathbb{R}^d$ is a feature vector and $y_i \in \{-1, +1\}$ is a label. The loss functions are defined in the forms

$$f_i(w) = \log(1 + e^{-y_i x_i^T w}) + \frac{\lambda}{2} \|w\|^2, \tag{84}$$

$$f_i(w) = \left( y_i - \frac{1}{1 + e^{-x_i^T w}} \right)^2, \tag{85}$$

where (84) is a regularized logistic regression of a particular choice of $\lambda > 0$ (and we used $\lambda \in \{0.1/n, 1/n, 10/n\}$ in the experiment), and hence a strongly convex function; and (85) is a non-linear least square loss, which is apparently non-convex.

The problem we aimed to solve is then defined in the form

$$\min_{w \in \mathbb{R}^d} \left\{ F(w) := \frac{1}{n} \sum_{i=1}^{n} f_i(w) \right\}, \tag{86}$$

and we denote $w^*$ the global optimizer of (86) for logistic regression.

### C.3.2 CONFIGURATION OF ALGORITHM

To better evaluate the performance of the algorithms, we configured `OASIS`, AdGD and AdaHessian with different choices of parameters.

- `OASIS` was configured with different values of $\beta_2$ and $\alpha$, where $\beta_2$ can be any values in $\{0.95, 0.99, 0.995, 0.999\}$ and $\alpha$ can be any value in the set $\{10^{-3}, 10^{-5}, 10^{-7}\}$ (see Section C.5 where `OASIS` has a narrow spectrum of changes with respect to different values of $\alpha$ and $\beta_2$). In addition, we adopted a warmstarting approach to evaluate the diagonal of Hessian at the starting point $w_0$.
- AdGD was configured with 12 different values of the initial learning rate, i.e., $\eta_0 \in \{10^{-11}, 10^{-10}, ..., 0.1, 1.0\}$.
- AdaHessian was configured with different values of the fixed learning rate: for logistic regression, we used 12 values between 0.1 and 5; for non-linear least square, we used $0.01, 0.05, 0.1, 0.5, 1.0$ and $2.0$.

To take into account the randomness of the performance, we used 10 distinct random seeds to initialize $w_0$ for each algorithm, dataset and problem.

### C.3.3 EXTENDED EXPERIMENTAL RESULTS

This section presents the extended results on our numerical experiments. For the whole experiments in this paper, we ran each method 10 times starting from different initial points.

For **logistic regression**, we show the evolution of the optimality gap in Figure 6 and the ending gap in Figure 7; the evolution of the testing accuracy and the maximum achieved ones are shown in Figure 8 and Figure 9 respectively.

For **non-linear least square**, the evolution of the objective and its ending values are shown in Figure 10; the evolution of the testing accuracy along with the maximum achived ones are shown in Figure 11.

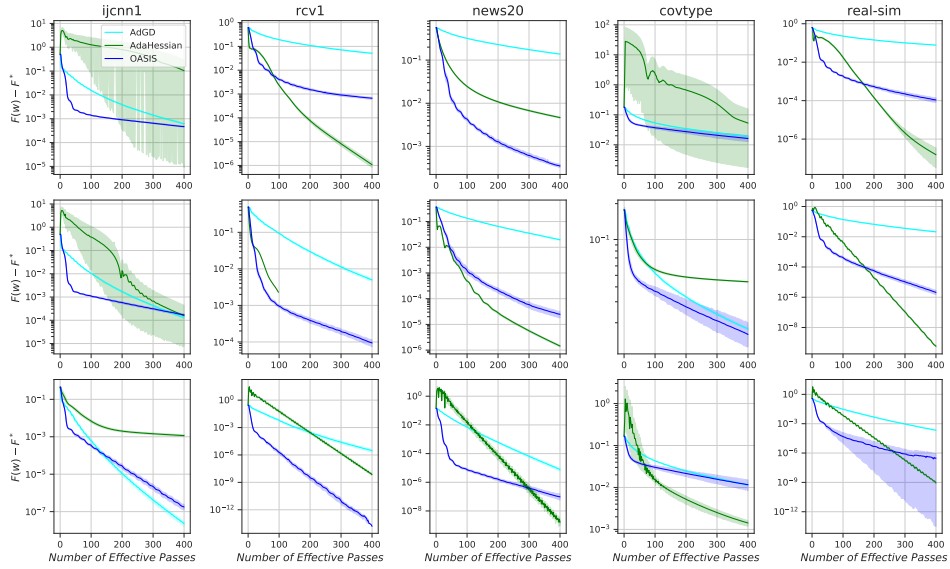

Figure 6: Evolution of the optimality gaps of OASIS, AdGD and AdaHessian for $\ell_2$-regularized Logistic regression: $\lambda = \frac{0.1}{n}$ (top row), $\lambda = \frac{1}{n}$ (middle row), and $\lambda = \frac{10}{n}$ (bottom row). From left to right: *ijcnn1, rcv1, news20, covtype* and *real-sim*.

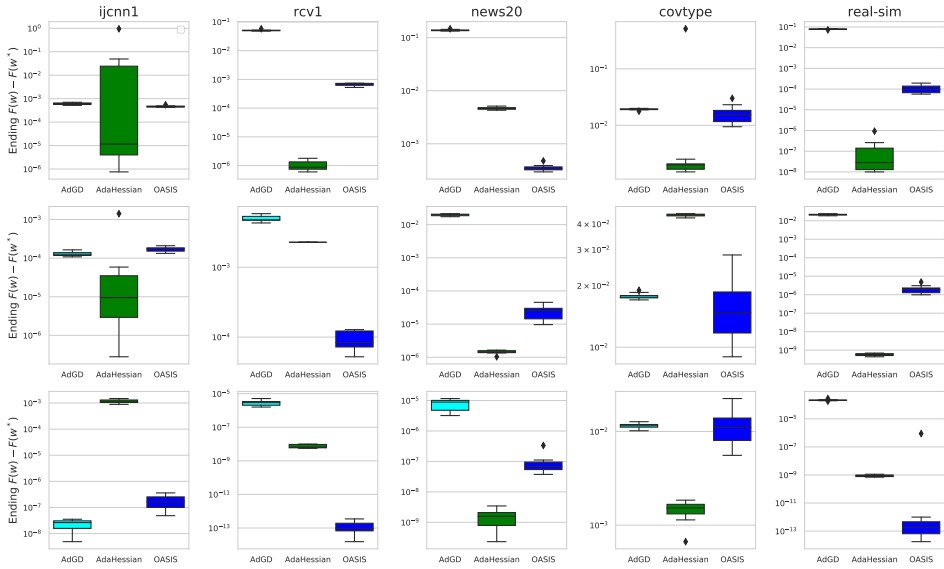

Figure 7: Ending optimality gaps of OASIS, AdGD and AdaHessian for $\ell_2$-regularized Logistic regression: $\lambda = \frac{0.1}{n}$ (top row), $\lambda = \frac{1}{n}$ (middle row), and $\lambda = \frac{10}{n}$ (bottom row). From left to right: *ijcnn1, rcv1, news20, covtype* and *real-sim*.

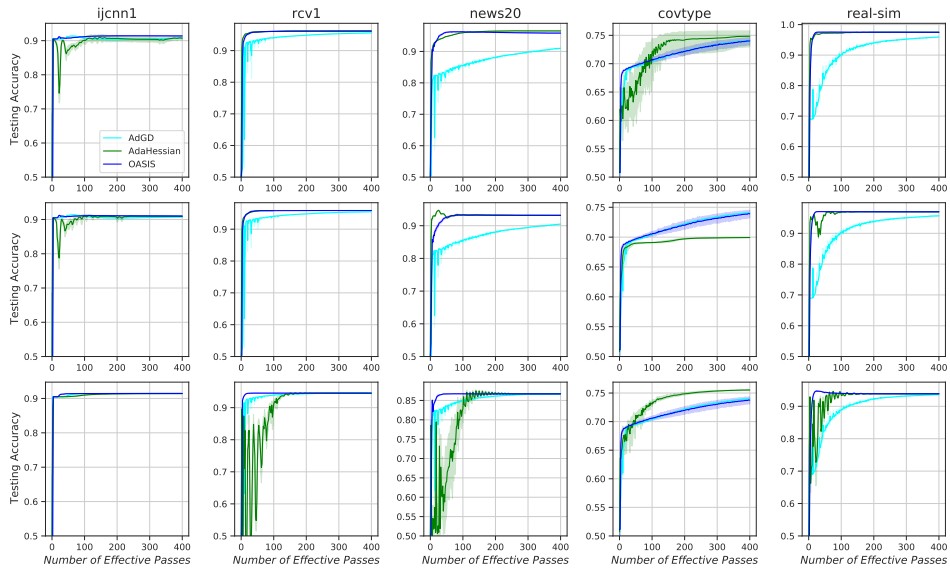

Figure 8: Evolution of the testing accuracy of OASIS, AdGD and AdaHessian for $\ell_2$-regularized Logistic regression: $\lambda = \frac{0.1}{n}$ (top row), $\lambda = \frac{1}{n}$ (middle row), and $\lambda = \frac{10}{n}$ (bottom row). From left to right: *ijcnn1, rcv1, news20, covtype* and *real-sim*.

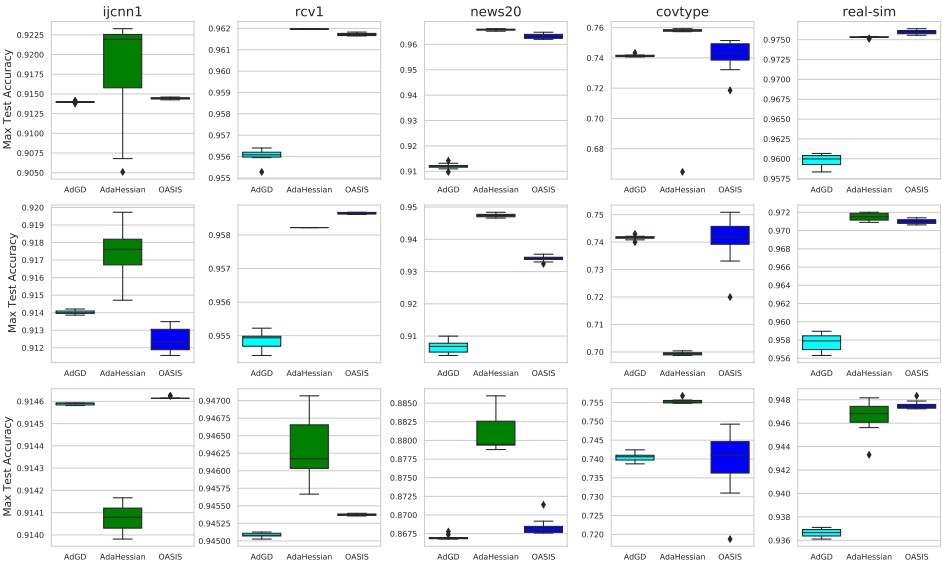

Figure 9: Maximum testing accuracy of OASIS, AdGD and AdaHessian for $\ell_2$-regularized Logistic regression: $\lambda = \frac{0.1}{n}$ (top row), $\lambda = \frac{1}{n}$ (middle row), and $\lambda = \frac{10}{n}$ (bottom row). From left to right: *ijcnn1, rcv1, news20, covtype* and *real-sim*.

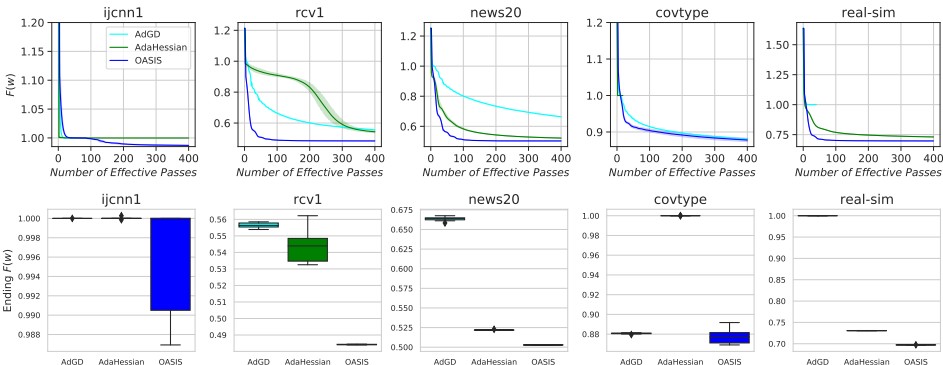

Figure 10: Evolution of the objective $F(w)$ (top row) and the ending $F(w)$ (bottom row) of OASIS, AdGD and AdaHessian for non-linear least square. From left to right: *ijcnn1, rcv1, news20, covtype* and *real-sim*.

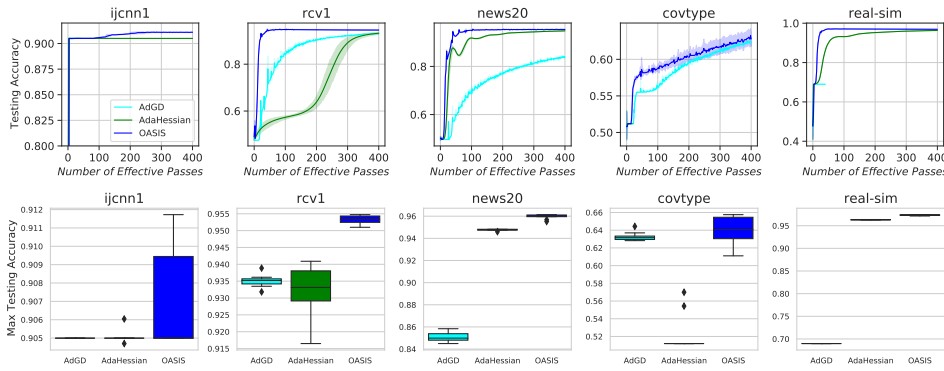

Figure 11: Evolution of the testing accuracy (top row) and the maximum accuracy (bottom row) of OASIS, AdGD and AdaHessian for non-linear least square. From left to right: *ijcnn1, rcv1, news20, covtype* and *real-sim*.

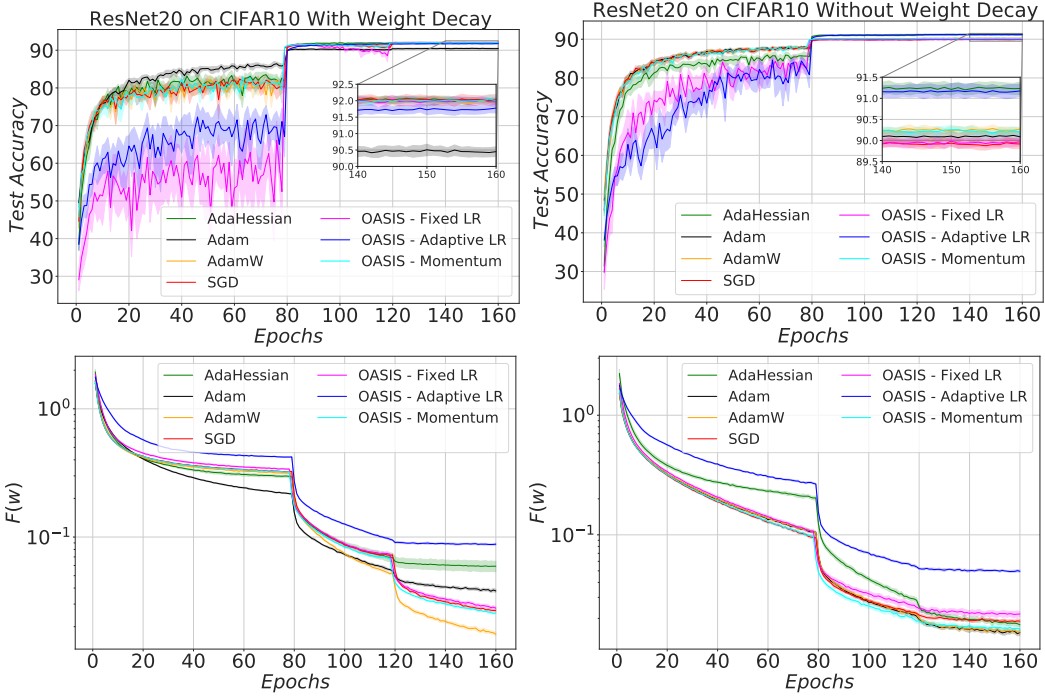

Figure 12: ResNet20 on CIFAR10 with and without weight decay. Final accuracy results can be found in Table 7.

## C.4    IMAGE CLASSIFICATION

In the following sections, we provide the results on standard bench-marking neural network training tasks: `CIFAR10`, `CIFAR100`, and `MNIST`.

### C.4.1    CIFAR10

In our experiments, we compared the performance of the algorithm described in Table 4 in 2 settings - with and without weight decay. The procedure for choosing parameter values differs slightly for these two cases, so they are presented separately. Implementations of ResNet20/32 architectures are taken from AdaHessian repository.[7]

Analogously to (Malitsky & Mishchenko, 2020), we also modify terms in the update formula for $\eta_k$

$$\eta_k = \min\{\sqrt{1 + \theta_{k-1}}\eta_{k-1}, \frac{\|w_k - w_{k-1}\|_{\hat{D}_k}}{2\|\nabla F(w_k) - \nabla F(w_{k-1})\|^*_{\hat{D}_k}}\}.$$

Specifically, we incorporate parameter $\gamma$ into $\sqrt{1 + \gamma\theta_{k-1}}$ and use more optimistic bound of $1/L_k$ instead of $1/2L_k$, which results in a slightly modified rule

$$\eta_k = \min\{\sqrt{1 + \gamma\theta_{k-1}}\eta_{k-1}, \frac{\|w_k - w_{k-1}\|_{\hat{D}_k}}{\|\nabla F(w_k) - \nabla F(w_{k-1})\|^*_{\hat{D}_k}}\}.$$

In order to find the best value for $\gamma$, we include it into hyperparameter tuning procedure, ranging the values in the set $\{1, 0.1, 0.05, 0.02, 0.01\}$ (see Section C.5 where `OASIS` has a narrow spectrum of changes with respect to different values of $\gamma$).

Moreover, we used a learning-rate decaying strategy as considered in (Yao et al., 2020) to have the same and consistent settings. In the aforementioned setting, $\eta_k$ would be decreased by a multiplier on some specific epochs common for every algorithms (the epochs 80 and 120).

---

[7]https://github.com/amirgholami/adahessian

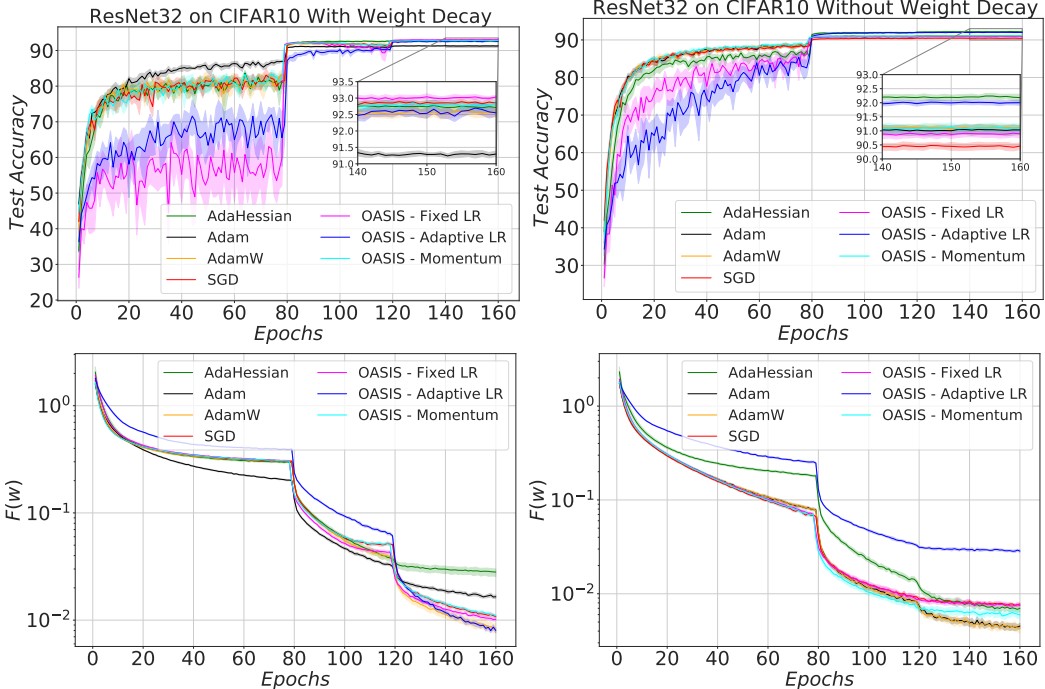

Figure 13: ResNet32 on CIFAR10 with and without weight decay. Final accuracy results can be found in Table 7.

Table 7: Results of ResNet20/32 on CIFAR10 with and without weight decay. Variant of our method with fixed learning rate beats or is on par with others in the weight decay setting; while OASIS with adaptive learning rate produces consistent results showing a close second best performance without any learning rate tuning (for without weight decay setting).

| Setting | ResNet20, WD | ResNet20, no WD | ResNet32, WD | ResNet32, no WD |
|---|---|---|---|---|
| SGD | $92.02 \pm 0.22$ | $89.92 \pm 0.16$ | $92.85 \pm 0.12$ | $90.55 \pm 0.21$ |
| Adam | $90.46 \pm 0.22$ | $90.10 \pm 0.19$ | $91.30 \pm 0.15$ | $91.03 \pm 0.37$ |
| AdamW | $91.99 \pm 0.17$ | $90.25 \pm 0.14$ | $92.58 \pm 0.25$ | $91.10 \pm 0.16$ |
| AdaHessian | $\mathbf{92.03 \pm 0.10}$ | $\mathbf{91.22 \pm 0.24}$ | $92.71 \pm 0.26$ | $\mathbf{92.19 \pm 0.14}$ |
| OASIS-Adaptive LR | $91.20 \pm 0.20$ | $91.19 \pm 0.24$ | $92.61 \pm 0.22$ | $91.97 \pm 0.14$ |
| OASIS-Fixed LR | $91.96 \pm 0.21$ | $89.94 \pm 0.16$ | $\mathbf{93.01 \pm 0.09}$ | $90.88 \pm 0.21$ |
| OASIS-Momentum | $92.01 \pm 0.19$ | $90.23 \pm 0.14$ | $92.77 \pm 0.18$ | $91.11 \pm 0.25$ |

WD := Weight decay

*With weight decay*: It is important to mention that due to variance in stochastic setting, the learning rate needs to decay (regardless of the learning rate rule is adaptive or fixed) to get convergence. One can apply adaptive learning rate rule without decaying by using variance-reduced methods (future work). Experimental setup is generally taken to be similar to the one used in (Yao et al., 2020). Parameter values for SGD, Adam, AdamW and AdaHessian are taken from the same source, meaning that learning rate is set to be $0.1/0.001/0.01/0.15$ and, where they are used, $\beta_1$ and $\beta_2$ are set to be $0.9$ and $0.999$. For `OASIS`, due to a significantly different form of the preconditioner aggregation, we conducted a small scale search for the best value of $\beta_2$, choosing from the set $\{0.999, 0.99, 0.98, 0.97, 0.96, 0.95\}$, which produced values of $0.99/0.95/0.98$ for "Fixed LR," "Momentum" and adaptive variants correspondingly for ResNet20 and $0.99/0.99/0.98$ for "Fixed LR," "Momentum" and adaptive variants correspondingly for ResNet32, while $\beta_1$ for momentum variant is still set to $0.9$. Additionally, in the experiments with fixed learning rate, it is tuned for each architecture, producing values $0.1/0.1$ for ResNet20 and $0.1/0.1$ for ResNet32 for "Fixed LR" and "Momentum" variants of `OASIS` correspondingly. For the adaptive variant, $\gamma$ is set to $0.1/1.0$ for ResNet20/32. $\alpha$ is set to $0.1$ for all `OASIS` variants. We train all methods for 160 epochs and for all methods. With fixed learning rate we employ identical scheduling, reducing step size by a factor of 10 at epochs 80 and 120. For the adaptive variant scheduler analogue is implemented, multiplying step size by $\rho$ at epochs 80 and 120, where $\rho$ is set to be $0.1/0.5$ for ResNet20/32. Weight decay value for all optimizers with fixed learning rate is $0.0005$ and decoupling for `OASIS` is done similarly to AdamW and AdaHessian. For adaptive `OASIS` weight decay is set to $0.001$ without decoupling. Batch size for all optimizers is 256.

*Without weight decay*: In this setting we tune learning rate for SGD, Adam, AdamW and AdaHessian, obtaining the values $0.15/0.005/0.005/0.25$ and $0.125/0.01/0.01/0.25$ for ResNet20/32 accordingly. Where relevant, $\beta_1, \beta_2$ are taken to be $0.9, 0.999$. For `OASIS` we still try to tune $\beta_2$, which produces values of $0.999$ for ResNet20 and adaptive case of ResNet32 and $0.99$ for "Fixed LR," "Momentum" in the case of ResNet32; for the momentum variant $\beta_1 = 0.9$. Learning rates are $0.025/0.05$ for ResNet20 and $0.025/0.1$ for ResNet32 for "Fixed LR" and "Momentum" variants of `OASIS` correspondingly. For adaptive variant $\gamma$ is set to $0.01/0.01$ for ResNet20/32. $\alpha$ is set to $0.1$ for adaptive `OASIS` variant, while hyperoptimization showed, that for "Fixed LR" and "Momentum" a value of $0.01$ can be used. We train all methods for 160 epochs and for all methods with fixed learning rate we employ identical scheduling, reducing step size by a factor of 10 at epochs 80 and 120. For the adaptive variant scheduler analogue is implemented, multiplying step size by $\rho$ at epochs 80 and 120, where $\rho$ is set to be $0.1/0.1$ for ResNet20/32. Batch size for all optimizers is 256.

Finally, all of the results presented are based on 10 runs with different seeds, where parameters are chosen amongst the best runs produced at the preceding tuning phase.

All available results can be seen in Table 7 and Figures 12 and 13. We ran our experiments on an NVIDIA V100 GPU.

### C.4.2   CIFAR100

Analogously to CIFAR10, we do experiments in settings with and without weight decay. In both cases we took best learning rates from the corresponding CIFAR10 experiment with ResNet20 architecture without any additional tuning, which results in $0.1$ for all. $\beta_2$ is set to $0.99$ in the case with weight decay and to $0.999$ in the case without it. ResNet18 architecture implementation is taken from a github repository.[8] We train all methods for 200 epochs. For all optimizers learning rate is decreased (or effectively decreased in case of fully adaptive `OASIS`) by a factor of 5 at epochs 60, 120 and 160.

All of the results presented are based off of 10 runs with different seeds. All available results can be seen in Table 8 and Figure 14. We ran our experiments on an NVIDIA V100 GPU.

### C.4.3   MNIST

Similar to the previous results, we ran the methods with 10 different random seeds. We considered **Net DNN** which has 2 convolutional layers and 2 fully connected layers with ReLu non-linearity, mentioned in the Table 5. In order to tune the hyperparameters for other algorithms, we considered the set of learning rates $\{10^0, 10^{-1}, 10^{-2}, 10^{-3}\}$. For `OASIS`, we used the set of $\{10^{-1}, 10^{-2}\}$ and the

---

[8]https://github.com/uoguelph-mlrg/Cutout

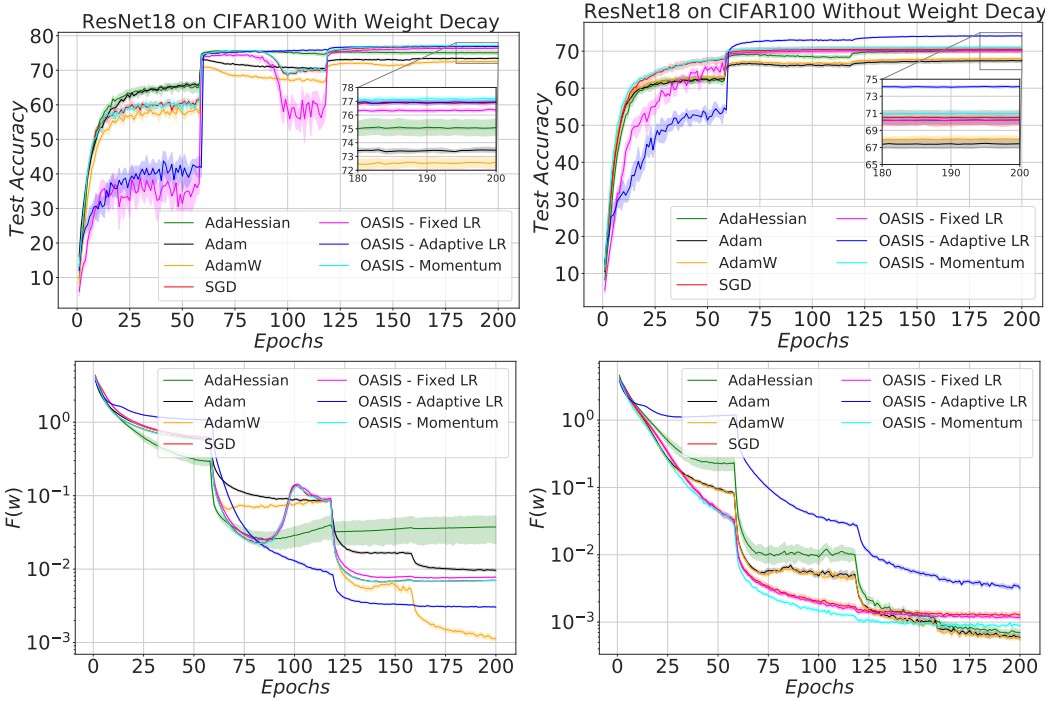

Figure 14: ResNet18 on CIFAR100 with and without weight decay. Final accuracy results can be found in Table 8.

Table 8: Results of ResNet18 on CIFAR100. Simply transferring parameter values from similar task with ResNet20 on CIFAR10 predictably damages performance of optimizers compared to their heavily tuned versions. Notably, in the setting without weight decay performance of SGD and Adam became unstable for different initializations, while adaptive variant of OASIS produces behaviour robust to the choice of the random seed.

| Setting | ResNet18, WD | ResNet18, no WD |
|---|---|---|
| SGD | $76.57 \pm 0.24$ | $70.50 \pm 1.51$ |
| Adam | $73.40 \pm 0.31$ | $67.40 \pm 0.91$ |
| AdamW | $72.51 \pm 0.76$ | $67.96 \pm 0.69$ |
| AdaHessian | $75.71 \pm 0.47$ | $70.16 \pm 0.82$ |
| OASIS-Adaptive LR | $\mathbf{76.93 \pm 0.22}$ | $\mathbf{74.13 \pm 0.20}$ |
| OASIS-Fixed LR | $76.28 \pm 0.21$ | $70.18 \pm 0.76$ |
| OASIS-Momentum | $76.89 \pm 0.34$ | $70.93 \pm 0.77$ |

WD := Weight decay

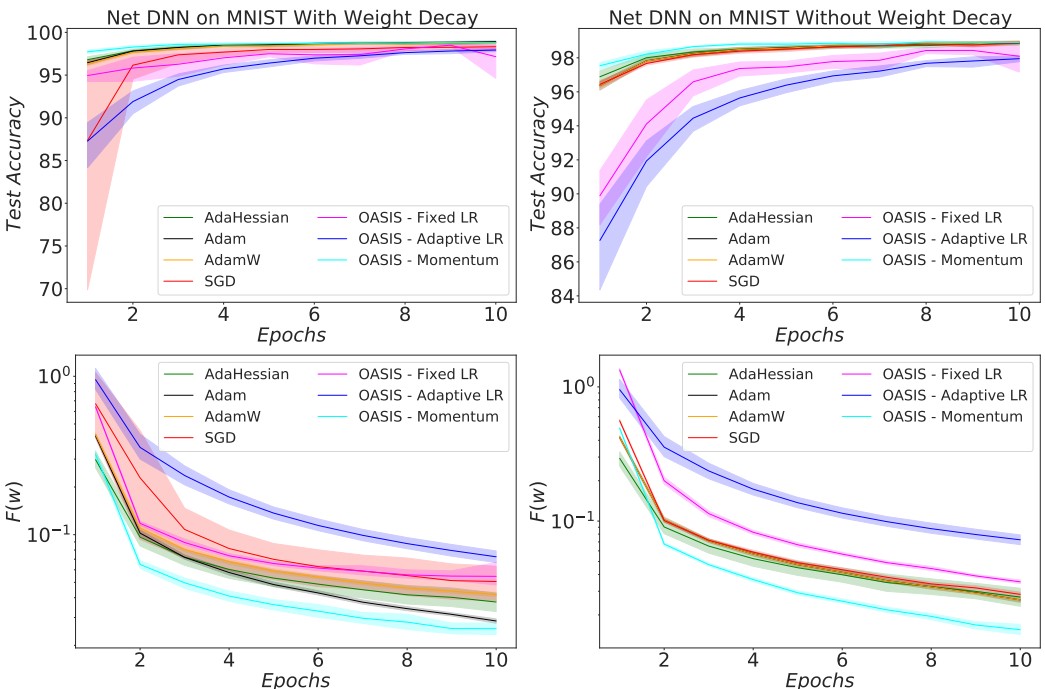

Figure 15: Net DNN on MNIST with and without weight decay. Final accuracy results can be found in Table 9.

Table 9: Results of Net DNN on MNIST. Gradient momentum usage seems improve results on short trajectories for all methods, including `OASIS`.

| Setting | Net DNN, WD | Net DNN, no WD |
|---|---|---|
| SGD | $98.37 \pm 0.45$ | $98.86 \pm 0.18$ |
| Adam | $\mathbf{98.92 \pm 0.14}$ | $98.90 \pm 0.13$ |
| AdamW | $98.76 \pm 0.13$ | $\mathbf{98.92 \pm 0.13}$ |
| AdaHessian | $98.82 \pm 0.16$ | $98.86 \pm 0.16$ |
| `OASIS`-Adaptive LR | $97.93 \pm 0.27$ | $97.95 \pm 0.29$ |
| `OASIS`-Fixed LR | $97.24 \pm 3.97$ | $98.09 \pm 1.36$ |
| `OASIS`-Momentum | $98.78 \pm 0.22$ | $98.89 \pm 0.10$ |

WD := Weight decay

set for truncation parameter $\alpha \in \{10^{-1}, 10^{-2}\}$. As is clear from the results shown in Figure 15 and Table 9, `OASIS` with momentum has the best performance in terms of training (lowest loss function), and and it is comparable with the best test accuracy for the cases with and without weight decay. It is worth mentioning that `OASIS` with adaptive learning rate got satisfactory results with lower number of parameters, which require tuning, which is vividly important, especially in comparison to first-order methods that are sensitive to the choice of learning rate. We ran our experiments on a Tesla K80 GPU.

## C.5 SENSITIVITY ANALYSIS OF `OASIS`

It is worth noting that we designed and analyzed `OASIS` for the deterministic setting with adaptive learning rate. It is safe to state that no sensitive tuning is required in the deterministic setting (the learning rate is updated adaptively, and the performance of `OASIS` is completely robust even if the rest of the hyperparameters are not perfectly hand-tuned); see Figures 16 and 17.

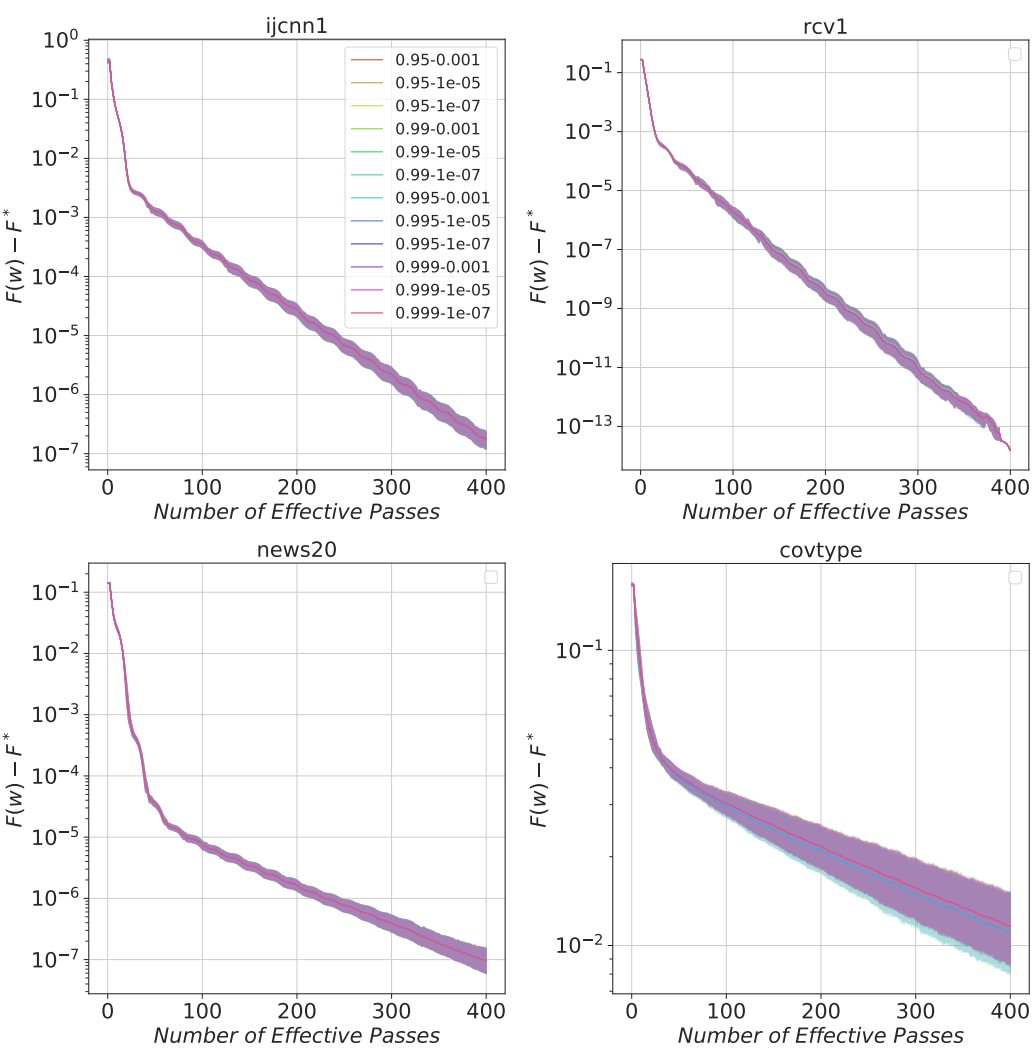

Figure 16: Sensitivity of OASIS w.r.t. $(\beta_2, \alpha)$, Deterministic Logistic regression.

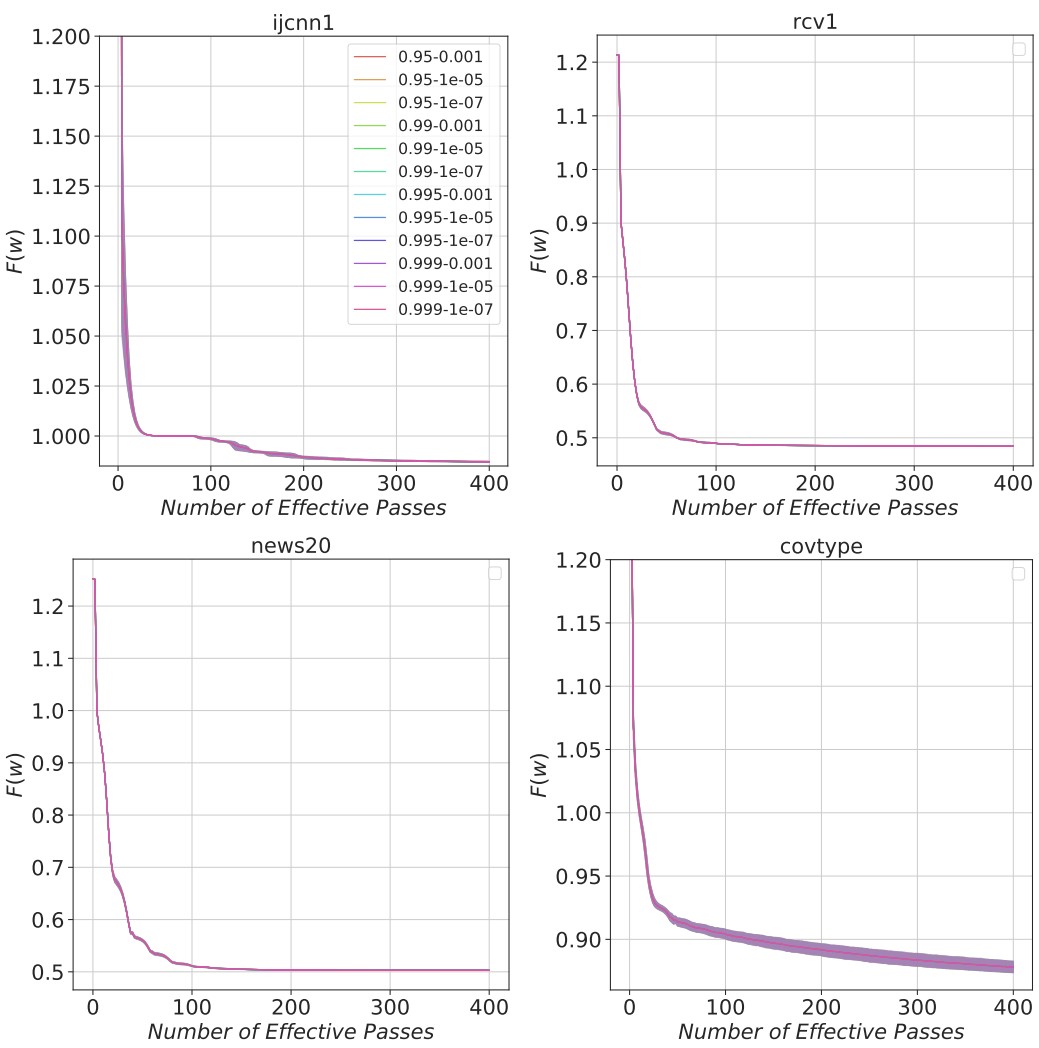

Figure 17: Sensitivity of OASIS w.r.t. $(\beta_2, \alpha)$, Deterministic Non-linear Least Square.

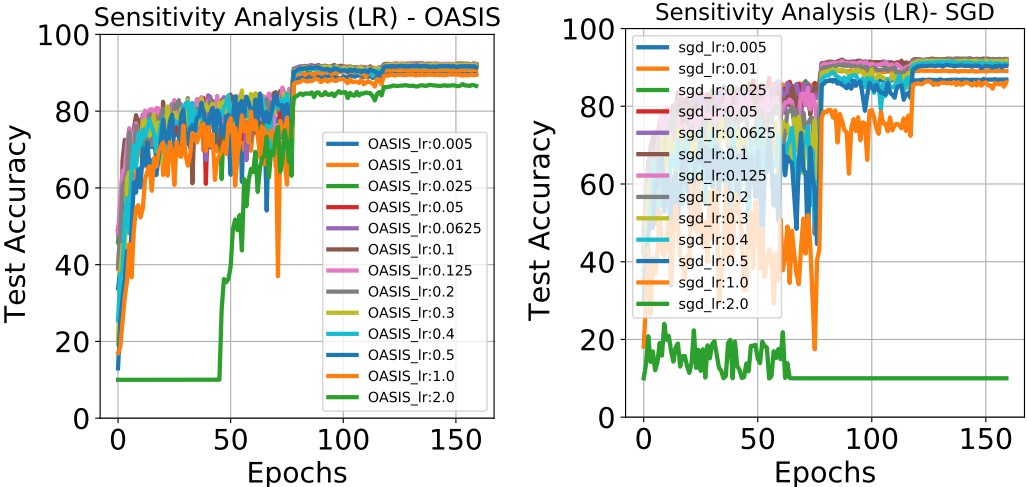

Figure 18: Sensitivity of `OASIS` vs. SGD w.r.t. learning rate $\eta$, `CIFAR10` on `ResNet20`.

We also provide sensitivity analysis for the image classification tasks. These problems are in stochastic setting, and we need to tune some of the hyperparameters in `OASIS`. As is clear from the following figures, `OASIS` is robust with respect to different settings of hyperparameters, and the spectrum of changes is narrow enough; implying that even if the hyperparameters aren't properly tuned, we still get acceptable results that are comparable to other state-of-the-art first- and second-order approaches with less tuning efforts. Figure 18 shows that `OASIS` is robust for different values of learning rate (left figure), while SGD is completely sensitive with respect to learning rate choices. One possible reason for this is because OASIS uses well-scaled preconditioning, which scales each gradient component with regard to the local curvature at that dimension, whereas SGD treats all components equally.

Furthermore, in order to use an adaptive learning rate in the stochastic setting, an extra hyperparameter, $\gamma$, is used in `OASIS` (similar to (Malitsky & Mishchenko, 2020)). Figure 19 shows that `OASIS` is also robust with respect to different values of $\gamma$ (unlike the study in (Malitsky & Mishchenko, 2020)). The aforementioned figures are for `CIFAR10` dataset on the `ResNet20` architecture; the same behaviour is observed for the other network architectures.

Finally, we show here that the performance of `OASIS` is also robust with respect to different values of $\beta_2$. Figure 20 shows the robustness of `OASIS` across a range of $\beta_2$ values. This figure is for `CIFAR100` dataset on the `ResNet18` architecture.

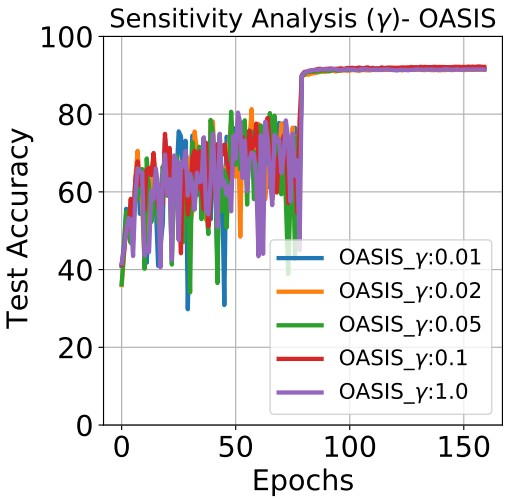

Figure 19: Sensitivity of `OASIS` w.r.t. $\gamma$, `CIFAR10` on `ResNet20`.

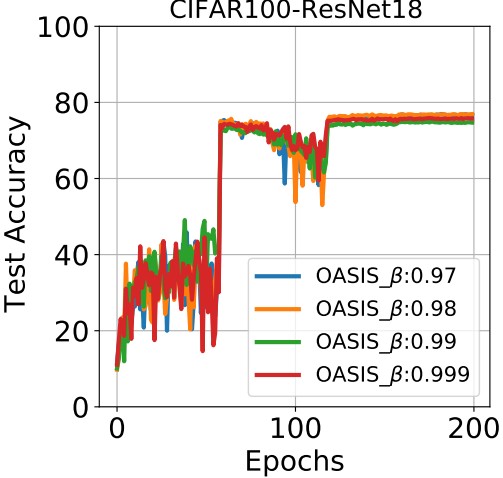

Figure 20: Sensitivity of `OASIS` w.r.t. $\beta_2$, `CIFAR100` on `ResNet18`.

