# OpenReview forum: "Doubly Adaptive Scaled Algorithm for Machine Learning Using Second-Order Information"
_ICLR.cc/2022/Conference — ICLR 2022 Poster_

### Official Review · Reviewer_524y · 2021-10-28

**Correctness:** 3
**Technical Novelty And Significance:** 2
**Empirical Novelty And Significance:** 2
**Recommendation:** 5
**Confidence:** 4

**Main Review:**

The paper is nicely written and easy-to-follow. The topic on how to effectively leverage the Hessian-vector oracle in large scale machine learning tasks is definitely important and interesting.

For the main ideas, the authors show in Figure 1 that OASIS approximates the diagonal of Hessian much more accurate than the Hessian momentum in AdaHessian, which is the main point made in the paper (I have the feeling that the Hessian momentum is not solely for approximation? like the first-order momentum vector may not be an accurate approximation for the gradient vector, but is effective for acceleration). Another point is that OASIS incorporates the adaptive stepsize in (Mishchenko & Malitsky, 2020), which allows it to adapt to the local Lipschitz constant (wrt a weighted Euclidean norm) and thus reduces the tuning effort. However, it seems to me that these ideas are a bit straightforward and not particularly novel. From my perspective, AdaHessian is a ''diagonal-Hessian-variant'' of Adam and OASIS is the corresponding variant of RMSProp. It seems that the adaptive learning rate can also be incorporated into AdaHessian by choosing a different weighted norm.

For the theory part, I appreciate the thorough analysis of OASIS under various settings. However, I was hoping for more insightful discussion on these results, such as how the theorems would suggest a better parameter choice. Currently, they are only convergence guarantees, which could be far from the practical performance. The theorems in Section 4.1 generalize the results in (Mishchenko & Malitsky, 2020) in the deterministic setting while there seems to be no theoretical advantage of such generalization (BTW, is there any bound on the scale of $Q_k$ in Theorem 4.6? It seems that it can be of the order $O(k)$ which kills the convergence).

For the empirical results, the authors considered various machine learning tasks and the deviation is also plotted in the figures, which are appreciated. However, the improvement in most of the results seems marginal to me, and thus may not be appealing to practitioners especially since the Hessian-vector oracle is around twice as expensive as the gradient oracle (for neural nets). Moreover, OASIS still requires a learning rate scheduler as shown in the CIFAR results, which makes the statement "Our methodology does not require the tedious task of learning rate tuning" not well-supported.

Minor comments:
- Equation (7) is not centered.
- Typo in the citation "Adaptive gradient descent without descent.  In 37th International Conference on Machine Learning (ICLM 2020), 2020"
- I think Lemma A.1 is covered by Theorem 2.1.5 in Nesterov's updated book "Nesterov, Y. (2018). Lectures on convex optimization (Vol. 137). Berlin, Germany: Springer International Publishing".

**Summary Of The Paper:**

This work proposes OASIS, a second-order method which approximates the diagonal of Hessian matrix and uses the information to rescale the gradient vector. The main difference between OASIS and the existing method AdaHessian (Yao et al., 2020) is on the ways they approximate the diagonal of Hessian, that AdaHessian uses a formula similar to Adam and OASIS uses an exponential average. Moreover, OASIS also incorporates the adaptive stepsize in (Mishchenko & Malitsky, 2020). The authors established the convergence guarantees of OASIS under various settings including convex, strongly convex and nonconvex cases, using various learning rate schedulers such as the adaptive learning rate, fixed learning rate and line search. Empirical results on various machine learning tasks are provided to evaluate OASIS.

**Summary Of The Review:**

I appreciate the authors' efforts on the comprehensive analysis and empirical evaluations of the proposed OASIS. The paper is also very well written. However, both the theoretical and practical results seem incremental to me. The construction in OASIS also seems a bit straightforward. Moreover, OASIS still requires parameter tuning in some of the experiments, and thus is not "fully adaptive".

---

> ### Author Response · Authors · 2021-11-18
> **Reply to Reviewer 524y (minor comments)**
>
> $\textbf{C1)}$ Equation (7)
>
> $\textbf{A1) }$ Great catch, thank you. We have fixed it in the updated version.
> ****
> $\textbf{C2)}$ Typo in the citation
>
> $\textbf{A2) }$ Thank the reviewer for this comment. We have fixed it in the updated version.
> ****
> $\textbf{C3)}$ Theorem 2.1.5 in Nesterov's updated book
>
> $\textbf{A3) }$ Great comment. We thank the reviewer for referring Theorem 2.1.5 in Nesterov's updated book. Nesterov proved Theorem 2.1.5 for any norm, and thus, Lemma A.1 can be seen as a special case of Theorem 2.1.5 w.r.t. the weighted Euclidean norm. We  provided the proof for completeness. In the updated version of our paper, we have clarified it and mentioned the point in this comment.

---

> ### Author Response · Authors · 2021-11-18
> **Reply to Reviewer 524y (Hessian-vector cost and learning rate scheduler)**
>
> Thank you for these great comments. We agree with the reviewer that the Hessian-vector oracle is around twice as expensive as the gradient oracle, and we mentioned it in the text. However, $\texttt{OASIS}$ method, similar to the methods utilizing curvature information, is robust and less sensitive with respect to changes in the hyper-parameters in comparison to first-order methods (i.e., methods using just gradient information). In other words, $\texttt{OASIS}$ method requires less time for hyper-parameter tuning which definitely compensate the cost of Hessian-vector products, while tuning of the hyper-parameters for first-order methods is indeed a big bottleneck and requires a lot of time and energy.
>
> Furthermore, for the deterministic regime, we do not need to tune the hyper-parameters specially the learning rate.
> Regarding the reviewer's comment about the learning rate scheduler, we did $\textbf{NOT}$ spent any time playing with the learning rate schedulers, we just picked the ones that have been tuned for SGD type algorithms (similarly as was done in AdaHessian) that shows our $\texttt{OASIS}$ is robust with respect to the hyper-parameters (learning rate scheduler is less sensitive hyper-parameter in comparison to the learning rate itself, and again our $\texttt{OASIS}$ method performs very well even without tuning the learning rate scheduler). In addition,
> Figures 16, 17, 18, 19 and 20 (in Appendix) shows the robustness of $\texttt{OASIS}$ with respect to the changes in hyper-parameters.

---

> ### Author Response · Authors · 2021-11-18
> **Reply to Reviewer 524y (convergence guarantees and Theorem 4.6.)**
>
> We thank the reviewer for these comments. Most of our theoretical results solely provides the convergence guarantees which is completely normal and in the same page as in the analysis of second-order methods such as quasi-Newton methods (please see the discussion regarding the bounds in Section A.13 in Appebdix). Some of our theoretical guarantees such as Theorem 4.6 suggests to chose a value for $\beta_2$ close to 1 (as we utilized it in practice) in order to converge to a small neighborhood of stationary point(s).
>
> Moreover, regarding the reviewer's point about the scale of $Q_k$, please note that there is the term $(1-\beta_2)\Gamma \dfrac{Q_k}{k}$
> in Theorem 4.6., which essentially highlights that the parameter $\beta_2$ controls the size of the neighborhood of optimal solution(s). In other words, if $\beta_2$ is very close to 1 (which is the case for our algorithm), the convergence neighborhood would be very small. Meaning that, with the assumption regarding bounded iterates, our algorithm converges to a very small neighborhood of optimal solution as $\beta_2$
> is very close to 1 (while in SGD the step-size controls the size of this neighborhood and accordingly, the step-size needs to go zero in order to converge to the small neighborhood of the optimal solution(s)).
> Finally, Theorem 4.6. is related to the case with adaptive learning rate (i.e., eliminating the need for tuning one important hyper-parameter), thus it is normal if the rate would be a bit slower as we gain useful properties for our method.

---

> ### Author Response · Authors · 2021-11-18
> **Reply to Reviewer 524y (novelty of OASIS, and AdaHessian with adaptive learning rate)**
>
> We thank the reviewer for mentioning these comments. As the reviewer truly stated that the $\texttt{OASIS}$ Hessian diagonal approximation is more well-scaled than AdaHessian method. The reason that we originally did not analyze AdaHessian with adaptive learning rate is due to the point that the Hessian diagonal approximation by AdaHessian method is far from the true scale (please see Fig. 1). Thus, we proposed a new method that firstly, approximate the Hessian diagonal in a well-scaled manner; and secondly, our method incorporates the adaptive learning rate idea on top of the well-scaled diagonal of Hessian.
> The idea behind our well-scaled preconditioning matrix is simple yet very effective and keeps the scale of the diagonal of the Hessian (unlike AdaHessian). It should be considered that there is $\textbf{NOT}$ any convergence guarantee for AdaHessian algorithm (please see (Yao et al., 2020) for more info), while our study presents convergence guarantees on a comprehensive collection of optimization problems, including convex, strongly convex, and nonconvex problems, in both deterministic and stochastic regimes for a variety of learning rate scenarios (adaptive, fixed, using line search, and diminishing ).
>
>  Respectfully, we disagree with the comment of reviewer that ``OASIS is the corresponding variant of RMSProp." The preconditoning matrix for RMSProp is as follows ($j =k-1$):
>
> $$
> \hat{D}_{k} =\sqrt{\beta_2 \hat{D}^2_j +(1-\beta_2)\texttt{diag}\big( g_k\odot g_k \big)},
> $$
> while the $\texttt{OASIS}$ preconditioning matrix is constructed using decaying exponential average as follows ($j =k-1$):
>
> $$
> \hat{D}_{k} = (\beta_2 D_j + (1-\beta_2)v_k) (\text{with truncation step}).
> $$
> Clearly, there are noticeable differences among the above two  preconditioning matrices and the way they are constructed.
> Table 3 (in Section B in Appendix) gives a very good big picture for comparison of the considered methods with their corresponding preconditioning matrices. In addition, it should be noted that incorporation of adaptive learning rate (Mishchenko \& Malitsky, 2020) with some second-order methods is not trivial at all, and not all methods can benefit from it. Further, the $\textbf{analysis for the adaptive learning rate}$ case is far from simple (the results in Section 4.4 with their corresponding proofs). Fortunately, as we showed in our analysis and stated in the paper, our adaptive learning rate technique (with weighted norm) can benefit any method with a positive-definite preconditioning matrix and bounded eigenvalues. That said, if one can show that the eigenvalues of AdaHessian preconditioning matrix are positive and uniformly bounded above and below, then the analysis can be straightforward. To put everything a nutshell, we believe that the idea behind $\texttt{OASIS}$ is novel enough with supportable convergence guarantees, and $\texttt{OASIS}$ is a promising direction and hope researchers would be motivated by it and take it to the next level.

---

> ### Author Response · Authors · 2021-11-18
> **Thank you Reviewer 524y**
>
> Thank you for finding our paper "nicely written" and "easy-to-follow", and for taking the time to carefully read our paper, and mentioning that our paper considers fundamental ML problems, provides detailed theoretical conclusions, and is beneficial in the DL domain. We'll go over each of your great questions and comments in detail in the following. We would be grateful to hear whether this addresses your concerns, thanks.

---

> ### Author Response · Authors · 2021-11-21
> **Are your concerns addressed?**
>
> Dear Reviewer 524y,
>
> We hope our responses have addressed your valuable comments and questions. We were especially wondering if your concerns regarding the novelty of $\texttt{OASIS}$, theoretical results (Theorem 4.6), and learning rate scheduler are addressed.
>
> Please let us know if there are any remaining questions. We will be more than happy to hear what you think.
>
> Thank you!
>
> Authors

---

> ### Comment · Reviewer_524y · 2021-11-21
> **Response to Authors**
>
> Thank you very much for the detailed reply. I am still not sure about the point on learning rate scheduler. What is the difference between tuning the learning rate and choosing different learning rate schedulers? Why is the learning rate scheduler a less sensitive hyper-parameter? Clearly, there are exotic learning rate schedulers such as ExponentialLR, CosineAnnealingLR implemented in PyTorch, which significantly affect the performance (see, e.g., "Loshchilov, I., & Hutter, F. (2016). Sgdr: Stochastic gradient descent with warm restarts. arXiv preprint arXiv:1608.03983").

---

> > ### Author Response · Authors · 2021-11-22
> > **Reply to Reviewer 524y (learning rate scheduler)**
> >
> > We indeed thank the reviewer for checking our answers. Tuning the learning rate basically means choosing the initial learning rate, and different learning rate schedulers are used in order to modify the original learning rate (e.g., multiplying the original learning rate by a constant (let's say 0.1) in different epochs (indexed by 80 and 120)); we hope the difference between tuning the learning rate and choosing different learning rate schedulers are clear now. Furthermore, let us highlight that one of the key reasons for using learning rate scheduler is due to the variance in the stochastic gradient. We do not need to use the learning rate scheduler if we use additional combination with variance reduced stochastic gradient techniques, or we use overparametrized models.
> >
> > Also, the main reason for mentioning the less-sensitivity of the learning rate scheduler in comparison to the learning rate was solely based on the well performance of $\texttt{OASIS}$ even without tuning the used learning rate scheduler. However, it depends on applications, and can be different case by case. Please let us know if there are any remaining questions. We will be more than happy to hear what you think.

---

> > > ### Author Response · Authors · 2021-11-26
> > > **Are all your concerns addressed?**
> > >
> > > Dear Reviewer 524y,
> > >
> > > We hope our responses have addressed your valuable comments and questions. We were especially wondering if you have any further comments/questions.
> > >
> > > Thank you!
> > >
> > > Authors

---

> > > ### Author Response · Authors · 2021-11-27
> > > **Follow up**
> > >
> > > We thank the reviewer 524y again for their time and efforts, as well as their valuable comments. We were wondering whether the valuable comments/questions in your last message have been addressed.
> > >
> > > Please let us know if you have any further questions/comments, and we will be more than happy to address them during the rebuttal period till 29 Nov. We tried our best to comprehensively address all valuable reviews that we received, and will be happy to further discuss, should there be a need (as we did with Reviewer doAo which resulted in increasing their score to 8).
> > >
> > > Thanks,
> > >
> > > Authors

---

> > > ### Author Response · Authors · 2021-11-29
> > > **Follow Up**
> > >
> > > We thank the reviewer 524y for their time and efforts, as well as their valuable comments.
> > >
> > > That would be appreciated if you can check our responses to your message(s) and we would be happy to answer any more questions/comments. we believe we have addressed all your concerns, and we kindly ask you to re-evaluate the paper and reconsider the score.
> > >
> > > Best,
> > >
> > > Authors

---

> > > ### Author Response · Authors · 2021-12-01
> > > **Follow Up**
> > >
> > > Dear reviewer 524y,
> > >
> > > We appreciate your valuable comments and questions once more.
> > >
> > > In our humble idea, we comprehensively addressed your great questions/comments. Please let us know if there are any confusions that would further need any response from our side, we would be happy to explain more.
> > >
> > > As the reviewers doAo and QUCj kindly evaluated our answers, which addressed their valuable concerns and resulted in an increase in their scores, we respectfully request you to re-evaluate the paper and reconsider the score.
> > >
> > > Best,
> > >
> > > Authors

---

### Official Review · Reviewer_QUCj · 2021-11-01

**Correctness:** 4
**Technical Novelty And Significance:** 4
**Empirical Novelty And Significance:** Not applicable
**Recommendation:** 6
**Confidence:** 4

**Main Review:**

Strengths:
- The paper considers a fundamental problem in ML, i.e., minimizing a separable function.
- The algorithm and its convergence are proven for many cases, i.e., deterministic, stochastic, convex, non-convex.
- Empirical evidence is given that the algorithm outperforms comparable approaches like AdaHessian, etc.
- No need to tune a learning rate since the step lengths are determined by the curvature of the function. This can be really a huge advantage in the stochastic setting, i.e., in deep learning.

Weakness:
- The empirical evidence/experiments are rather limited.
  Deterministic case: Only two experiments are provided (logistic regression, non-linear least-squares) and only two data sets. Furthermore, a comparison to other minimization methods would be very beneficial in this case, and not only to AdaHessian and AdGD. (Yes, it is stated in the paper that comparison to only diagonal preconditioners is made but in general, there are many more methods to solve this case, e.g., quasi-Newton methods or trust region Newton-CG methods which are also used for computing the optimum in the provided code. These methods make use of the same information as the presented method and hence, a comparison to these methods would also be useful for a better global picture.)

  Stochastic case: Again, only a very limited number of experiments is provided here. Having not to tune the learning rate is an enormous plus here and it would be nice to verify the algorithm's robustness on a number of different problems/nets. The experiments suggest that OASIS would be a viable replacement for SGD, Adam, etc. But for such a bold statement, more experiments are needed.

**Summary Of The Paper:**

The paper designs and analyses an algorithm for minimizing a separable function. It provides deterministic as well as stochastic versions, which either fully compute the gradient or sample it. The algorithm estimates the diagonal elements of the Hessian via Hessian-vector products. The algorithm makes use of this information for finding a better search direction and the step length, eliminating the need for a line search. It provides convergence guarantees and a number of experiments on classical ML problems as well as on deep nets.

**Summary Of The Review:**

I like the paper, the algorithm, and its versatility. Especially that one does not need to tune a learning rate can be very beneficial. I did not fully read the convergence proofs though they seem sound. According to theory and experiments, one should always use this algorithm. It would be nice to justify this claim by a more comprehensive study, e.g., more problems, datasets, and other algorithms in the deterministic case and more nets and data sets in the stochastic setting. Only then one can tell if it is superior to state-of-the-art approaches. If such experiments were provided in the paper I would have given a higher score.

---

> ### Author Response · Authors · 2021-11-18
> **Reply to Reviewer QUCj (empirical expriments in stochastic case)**
>
> We thank the reviewer for this comment. For the stochastic setting, we used 3 datasets
> $\textit{MNIST}$, $\textit{CIFAR10}$, $\textit{CIFAR100}$
> and 4 networks $\textbf{NetDNN}$, $\textbf{ResNet20}$, $\textbf{ResNet32}$ and $\textbf{ResNet18}$ (as appropriate); please see Figures 12, 13, 14, and 15 for additional results and Figures 18, 19 and 20 for sensitivity analysis showing our $\texttt{OASIS}$ algorithm is robust w.r.t. changes in hyper-parameters. We believe that our numerical results demonstrated a promising direction and we hope our paper could help to push the optimization for ML closer to the goal of having less tuning when training some ML models. Here, we provide some related optimization papers with the same/fewer experiments than our study [1, 2, 3, 4] (and there are many more).
> ****
>  [1] Practical Quasi-Newton Methods for Training Deep Neural Networks Donald Goldfarb, Yi Ren, Achraf Bahamou, NeurIPS 2020
>
>  [2] RSN: Randomized Subspace Newton (Robert Gower, Dmitry Kovalev, Felix Lieder, Peter Richtarik) NeurIPS 2019
>
>  [3] Adaptive Newton method for empirical risk minimization to statistical accuracy (Aryan Mokhtari, Hadi Daneshmand, Aurelien Lucchi, Thomas Hofmann, Alejandro Ribeiro), NeurIPS 2016
>
>  [4] A Multi-Batch L-BFGS Method for Machine Learning (Albert S. Berahas, Jorge Nocedal, Martin Takáč), NeurIPS 2016
> ****

---

> ### Author Response · Authors · 2021-11-18
> **Reply to Reviewer QUCj (empirical expriments in deterministic case)**
>
> We thank the reviewer for these comments. Regrading the $\textbf{Deterministic Case}$, we provided comprehensive experiments for binary classification on $5$ popular binary class datasets, $\textit{ijcnn1, rcv1, news20, covtype}$ and $\textit{real-sim}$ (which cover different scenarios for number of features and samples) for both strongly convex case (regularized logistic regression) and nonconvex case (non-linear least squares). Please see Section C.3 in Appendix for additional results (Figures 6,7,8,9,10,11 are related additional results regarding more datasets, and Figures 16 and 17 are for sensitivity analysis showing our $\texttt{OASIS}$ algorithm is robust w.r.t. changes in hyper-parameters). Moreover, the main reason that we compare our method with AdaHessian and AdGD is due the point that these methods are also using the information of the Hessian $\textbf{diagonal}$ approximation (i.e., they belongs to the same class of algorithms) unlike quasi-Newton and Newton methods that the preconditiong matrix is either $\textbf{full}$ Hessian or its approximation (completely different class of methods).
>  Let us highlight that the difference between these two classes (the ones with Hessian diagonal approximation and the (approximation of the) full Hessian) is due to the point that the elements on the vector $m_k$ (or gradient) would be just scaled (in the first class), but would be both rotated and scaled (in the second class) by the inverse of preconditioning matrix to generate the search direction ($\hat{D}_{k}^{-1} m_k$) as shown in eq. (2) in the text. As an interesting avenue for future research, one can combine the idea of $\texttt{OASIS}$ with Newton-CG. To be precise, the efficient and well-scaled $\texttt{OASIS}$ preconditioning matrix can be used as the suitable preconditioner in Newton-CG to converge faster which is a very interesting direction to pursue.

---

> > ### Comment · Reviewer_QUCj · 2021-11-22
> > **Comparison to other methods**
> >
> > Dear authors,
> >
> > thank you very much for answering my questions. I highly appreciate this.
> >
> > As for the comparison to other methods: Yes, the comparison is made within one class of algorithms (Hessian approximation by a diagonal matrix.) But there is no good reason for not comparing it with other methods, that make use of the same information. The only reason I can see is for efficiency reasons, i.e., when the time for computing the search direction is much larger for the other methods and would dominate the optimization process. But this is not the case here. From a practical point of view, one would just pick the method that provides the best/fastest convergence for a problem, regardless of which class of method it belongs to. Especially, if the other methods (Adam, quasi-Newton, Netwon-CG, ... ) are readily available as in this case here.
> >
> > (If you allow a momentum term in your method, then you also implicitly allow rotations.)

---

> > > ### Author Response · Authors · 2021-11-22
> > > **Response to "Comparison to other methods"**
> > >
> > > Dear reviewer,
> > > we understand your point of view, but there is also a bit issue with comparing with e.g., quasi-Newton, Netwon-CG as suggested. To be honest, the class of problems we are solving could be also solved by quasi-Newton methods (also Newton+CG, Trust-Region, ....), but traditionally people do not compare with these methods when they work on first-order methods. One can even say that Quasi-Netwon methods just use Gradient Information, so why people do not always compare also with e.g., L-BFGS?
> > > The rationale of why not doing it is that we want to study the effect of using the Hutchinson method to obtain good diagonal scaling and hence we just study the effect of doing this algorithmic modification.
> > > We believe that this, by itself, is showing a big promise. As mentioned above, this could now be combined with e.g, the Trust-Region method or use as preconditioning in Newton+CG (PCG) which is another project we are currently working on, but it is not possible to include everything in one paper.

---

> > > > ### Author Response · Authors · 2021-11-26
> > > > **Are all your concerns addressed?**
> > > >
> > > > Dear Reviewer QUCj,
> > > >
> > > > We were wondering if you have any other questions as we responded to your prior and important feedback.
> > > >
> > > > Thank you!
> > > >
> > > > Authors

---

> > > > ### Author Response · Authors · 2021-11-27
> > > > **Follow up**
> > > >
> > > > We thank the reviewer QUCj again for their time and efforts, as well as their valuable comments. We were wondering whether the valuable comments/questions in your last message have been addressed.
> > > >
> > > > Please let us know if you have any further questions/comments, and we will be more than happy to address them during the rebuttal period till 29 Nov. We tried our best to comprehensively address all valuable reviews that we received, and will be happy to further discuss, should there be a need (as we did with Reviewer doAo which resulted in increasing their score to 8).
> > > >
> > > > Thanks,
> > > >
> > > > Authors

---

> > > > ### Author Response · Authors · 2021-11-29
> > > > **Follow Up**
> > > >
> > > > We thank the reviewer QUCj for their time and efforts, as well as their valuable comments.
> > > >
> > > > That would be appreciated if you can check our responses to your message(s) and we would be happy to answer any more questions/comments. we believe we have addressed all your concerns, and we kindly ask you to re-evaluate the paper and reconsider the score.
> > > >
> > > > Best,
> > > >
> > > > Authors

---

> ### Author Response · Authors · 2021-11-18
> **Thank you Reviewer QUCj**
>
> Thank you for taking the time to carefully read our paper, and mentioning that our paper considers fundamental ML problems, provides detailed theoretical conclusions, and is beneficial in the DL domain. We'll go over each of your great questions and comments in detail in the following. We would be grateful to hear whether this addresses your concerns, thanks.

---

> ### Author Response · Authors · 2021-11-21
> **Are your concerns addressed?**
>
> Dear Reviewer QUCj,
>
> We hope our responses have addressed your valuable comments and questions. We were especially wondering whether the experiments in $\textbf{Figures 6-20}$ with respect to the different datasets and different networks in both deterministic and stochastic regimes have addressed your concerns.
>
>
> Please let us know if there are any remaining questions. We will be more than happy to hear what you think.
>
> Thank you!
>
> Authors

---

> ### Author Response · Authors · 2021-12-01
> **Thanks Reviewer QUCj**
>
> Dear reviewer QUCj,
>
> We indeed thank your thoughtful comments and questions. Furthermore, we thank you for checking our responses, for re-evaluating our paper, and more importantly for raising your score, which means a lot to us.
>
> Best,
>
> Authors

---

### Official Review · Reviewer_doAo · 2021-11-01

**Correctness:** 3
**Technical Novelty And Significance:** 2
**Empirical Novelty And Significance:** 3
**Recommendation:** 8
**Confidence:** 3

**Main Review:**

------ Pros ------

The paper has the following strengths.

1) Very well written paper providing a clear motivation for the problem considered.

2) The theoretical results involving the convergence analysis of the method are rigorous, and cover both convex and nonconvex settings.

3) The empirical evaluation is extensive and provides a good indication of how the method performs in practice.

------ Cons ------

The paper has the following weaknesses.

1) There is currently not much discussion on the interpretation of the bounds appearing in the convergence analysis.
For instance, how do these results compare with those for existing second-order methods (e.g., AdaHessian)?

2) It would have been helpful to have provided some kind of proof sketch of the theoretical results, or atleast an
overview of the key steps which I believe might be common to more than one theorem. At the moment, no such explanation is
provided for any of the theoretical results.

------ Further remarks -------

1) The setup in Fig. 1 is really not clear to me. What is meant by number of samples (x axis of left plot)? Is there an underlying optimization problem considered here such as a quadratic function with matrix A? Some more detailed explanation on the experimental setup considered for the figure will be very helpful.

2) In Fig. 2, the parameter $\lambda$ is not clearly defined, I believe this occurs much later in the experiments section.

3) In eq. (6), is the matrix $D_k$ formed by just generating a Rademacher $z$ and forming $D_k = z \circ \nabla^2 F(w_k) z$? Because in the AdaHessian paper, they also consider a spatial averaging step for better estimation of the Hessian diagonal. Also, should $z$ be $z_k$ as in eq. (8) later?

4)  As mentioned on pg 4 in the discussion on literature for adaptive LR, the present paper draws upon ideas from the literature on first order methods for adaptive LR. So couldn't one do the same analysis for AdaHessian for adaptive LR?

5) It wasn't clear to me why AdaHessian (eq. 6 and 7) doesn't approximate the diagonal well, while  OASIS (eq. 8 and 9) does a better job.
Because we see a (temporal) average in eq. 7 as well, which means AdaHessian should also smooth out the Hessian noise over  iterations. Is there any conceptual reason behind this?

6) In Section 3.2, shouldn't the distribution from which the sets $\mathcal{I}_k, \mathcal{J}_k$ are sampled be specified (e.g., uniformly at random)? Or is it the case that the conditions on the distribution are subsumed by assumptions 4.14-4.16?. Also, in assumption 4.16: the sentence ''where the samples $\mathcal{I}$ are drawn independently'' should be removed since there is a single random variable $\mathcal{I}$.

7) Since $z_k$ is random, one would imagine that this randomness is accounted for in the convergence analysis, which doesn't seem to be the case. For instance the theorems 4.6, 4.9 seem to be worst case bounds. Moreover, the bound in theorem 4.9 depends on $\hat{D}_k$ which is a random variable. This point needs further clarification.

8) In theorem 4.17, its better to write $\eta_k = \eta$ for consistency of notation.

9) Both theorems 4.17, 4.18 show convergence to neighborhoods of stationary points, and not to the stationary points themselves. There is a discussion after theorem 4.18 regarding this aspect, but it seems a bit strange why this (e.g. decaying learning rate) is not accounted for in the analysis to begin with?

**Summary Of The Paper:**

The paper presents a novel adaptive second order method: ''OASIS'', for large scale optimization problems (convex and nonconvex). The search direction is obtained by preconditioning the gradient information with a matrix obtained by approximating the Hessian diagonal matrix (via Hutchinson's method with a momentum term). The learning rate is updated adaptively by approximating the Lipschitz smoothness parameter. On the theoretical front, convergence analysis is provided for the adaptive learning rate case, for the convex and strongly convex setups. Similar analysis is also provided for the fixed learning rate (LR) case, for the strongly convex and nonconvex settings. Finally, extensive empirical results are provided which show that the proposed method achieves comparable, and sometimes better results than other existing methods.

**Summary Of The Review:**

The paper is written in a very clean manner, and is easy to follow. Sufficient background is provided in the introduction which gives the reader a good context to understand the problem setting and contributions. The preconditioning step is a modification of an existing method AdaHessian, and the adaptive LR part builds on techniques used for deriving adaptive LR rules for first order methods (Mishchenko and Malitsky 2020). So the novelty aspect is a bit limited in that respect. The theoretical results are outlined rigorously, although it is not clear what is the novelty of the theoretical results compared to those for other second order methods. The empirical evaluation is quite extensive and satisfactory in my view. I am giving it a 6 at the moment since I have other comments (in ''Further remarks'') which I hope can be addressed during the rebuttal phase.

------------- Post rebuttal ---------

As mentioned in the comments, I am satisfied with the author's response to my concerns and I am happy to increase my score to 8.

---

> ### Author Response · Authors · 2021-11-17
> **Thank you Reviewer doAo**
>
> Thank you for your feedback and especially  for finding our paper "well written," with comprehensive theoretical results and extensive empirical experiments. In the following we address your valuable questions and comments in detail. We would be grateful to hear whether this addresses your concerns, thanks.

---

> ### Author Response · Authors · 2021-11-17
> **Reply to Reviewer doAo (reported cons)**
>
> $\textbf{Cons 1)}$ More interpretation of the bounds appearing in the convergence analysis. For instance, comparing with those for existing second-order methods (e.g., AdaHessian)
>
> $\textbf{A-C1)}$ Thanks for mentioning this comment. In the updated version, we've provided more explanation to the bounds that appear in the convergence analysis (please see Section A.13 in Appendix) and compared our convergence bounds with the bounds in some quasi-Newton methods. Furthermore, it is noteworthy to mention that there is $\textbf{NOT}$ any convergence guarantee for AdaHessian algorithm (please see (Yao et al., 2020) for more info) while we have provided comprehensive theoretical results for OASIS for a variety of cases.
>
> $\textbf{Cons 2)}$ Adding proof sketch of the theoretical results
>
> $\textbf{A-C2)}$ Thank you for this comment. The reason that we did not provide the proof sketch is due to the limited space in the main body of the paper. In our humble idea, the proof itself is the best source for understanding the provided analysis, and the proof sketch is almost similar to the provided proofs. If the reviewer still thinks that adding the proof sketch in the appendix can be necessary, we can add a section regarding that in the appendix.

---

> ### Author Response · Authors · 2021-11-17
> **Reply to Reviewer doAo (further remarks 5-9)**
>
> $\textbf{Further Remarks}$
>
> $\textbf{A5)}$ Great question -- thank you. AdaHessian idea is based on the second moment of $D_k$, while $\texttt{OASIS}$ is based on (decaying exponential) average of $D_k$ which  keeps the scale of the diagonal approximation by the Hutchinson's approach (where $D_k \approx \texttt{diag}( \mathbb{E}[z_{k} \odot \nabla^2F(w_k)z_{k}]$). Please note that, in AdaHessian approach, the square root of the second moment does not cancel out the magnitude of the second moment. Thus, as shown in Fig. 1, the diagonal approximation of matrix $A$ by $\texttt{OASIS}$ (with decaying exponential average) is well-scaled, while AdaHessian does not scale well.
>
> $\textbf{A6)}$ Thanks for this comment. In Section 3.2, it is not required to specify the distribution in which the sets $\mathcal{I}_k$ and $\mathcal{J}_k$ sampled. We have made it more clear in the paper. Moreover, as the reviewer suggested it is not required to mention ``where the samples $\mathcal{I}$ are drawn independently."
>
> $\textbf{A7)}$ Thanks the reviewer for these great question and comments. That is true that $z_k$ is random, however, the main point is that the eigenvalues of $\hat{D}_k$ in $\texttt{OASIS}$ are uniformly bounded above and below (Remark 4.10). Therefore, we can get the convergence results with bounded $\hat{D}_k$; please see the proofs of Theorems 4.6 and 4.9 for more information.
>
> $\textbf{A8)}$ Good call, thanks. We have added $\eta_k = \eta$ to be consistent.
>
> $\textbf{A9)}$ We thank the reviewer for this great comment. The extension of Theorems 4.17 and 4.18 with decaying learning rate is added in $\textbf{Section A.12 in Appendix}$ in the updated version of our paper (the corresponding proof simply follows from the proofs of Theorems 4.17 and 4.18 in our paper and Theorems 4.7 and 4.9 in [1] which is for SGD with diminishing learning rate for strongly convex and nonconvex cases). Moreover, showing a convergence to a neighborhood
> of optimal solution is still interesting and many papers provide similar results (especially in the case when inexact computation is done or e.g. for SGD when step-size is not diminishing). See for example [Theorem 4, 2],
> [Lemmas 1\&2,3], [Theorem 3.1,4],
> [Theorems 1--4, 5],
> [Theorems 3.1;3.4;3.6;3.8, 6],
> [Theorem 3.2, 7]
> or [Theorem 4.6, 8]
> ****
> [1] Optimization methods for large-scale machine learning (Léon Bottou, Frank E. Curtis, \& Jorge Nocedal), Siam Review.
>
> [2] An Accelerated Communication-Efficient Primal-Dual Optimization Framework for Structured Machine Learning,
> (Chenxin Ma, Martin Jaggi, Frank E. Curtis, Nathan Srebro, \& Martin Takáč), Optimization Methods and Software
>
> [3]  First-order methods of smooth convex optimization with inexact oracle, (Olivier Devolder,
> Fran{\c{c}}ois Glineur, Yurii Nesterov), Mathematical Programming
>
> [4] SGD: General Analysis and Improved Rates, (Robert M. Gower, Nicolas Loizou, Xun Qian, Alibek Sailanbayev, Egor Shulgin, \&  Peter Richtarik), ICML
>
> [5] Random Reshuffling: Simple Analysis with Vast Improvements, (Konstantin Mishchenko, Ahmed Khaled, \& Peter Richtárik), NeurIPS
>
> [6] Stochastic Polyak Step-size for SGD: An Adaptive Learning Rate for Fast Convergence, (Nicolas Loizou, Sharan Vaswani, Issam Laradji, \& Simon Lacoste-Julien), AISTATS
>
> [7] A Multi-Batch L-BFGS Method for Machine Learning (Albert S. Berahas, Jorge Nocedal, \& Martin Takáč), NeurIPS
>
> [8] A Stochastic Quasi-Newton Method for Large-Scale Optimization (R.H. Byrd, S.L. Hansen, J. Nocedal, \& Y.Singer),  SIAM Journal on Optimization
> ****

---

> ### Author Response · Authors · 2021-11-17
> **Reply to Reviewer doAo (further remarks 1-4)**
>
> $\textbf{Further Remarks}$
>
> $\textbf{A1)}$ Thank you for this comment. Fig. 1 is related to the diagonal approximation by AdaHessian and $\texttt{OASIS}$ which the Hutchinson's method is utilized differently in both methods. Hutchinson's method is a stochastic method in order to approximate the diagonal of a given matrix $A$ as $ \texttt{diag}( \mathbb{E}[z \odot Az])$ where $z$ is a random vector with Rademacher distribution. The $x$-axis in the left plot in Fig. 1 is essentially shows the number of random vector $z$ which is sampled from the Rademacher distribution in order to approximate the diagonal of matrix $A$. We have made it more clear in the updated version of our paper.
>
> $\textbf{A2)}$ As is mentioned in the caption of Figure 2, $\textcolor{red}{\lambda}$ is the strong convexity parameter for the regularized $\texttt{Logistic Regression}$ problem, i.e., the loss function is $F(w):= \tfrac{1}{n} \textstyle{\sum}_{i=1}^n f_i(w)$ , where $
>     f_i(w) = \log(1+e^{-y_ix_i^Tw}) + \frac{\textcolor{red}{\lambda}}{2}\|w\|^2$. We have made it more clear by removing $\lambda$ in the caption of Figure 2, and simply said ``(\texttt{Logistic Regression} with strong-convexity parameter $\textcolor{red}{\cancel{\lambda =}} \frac{1}{n}$ over $\texttt{rcv1}$ dataset)" in the updated version.
>
> $\textbf{A3)}$ Thank you for this comment. In $\texttt{OASIS}$, in order to construct $D_0$, we do the warmstarting step discussed in Section 3.3. In the warmstarting phase, we
> sample some predefined number of Hutchinson’s estimates before the training process. For constructing $D_k, \, \forall k \geq 1$, just $\textbf{one}$ Rademacher $z$ is required. Further, our $\texttt{OASIS}$ idea is based on decaying exponential average which is simple yet very effective, and keeps the scale of the diagonal of the Hessian and, more importantly, smooth out the Hessian noise over iterations. Our method, unlike AdaHessian, does not require any extra steps such as spatial averaging. In eq. 6, either $z$ or $z_k$ represents the idea of AdaHessian. We have changed $z$ to $z_k$ in the modified version of our paper, that shows its dependency to iteration $k$.
>
> $\textbf{A4)}$ Great comment $-$ thank you. Please note that the reason that we originally did not analyze AdaHessian with adaptive learning rate is due to the point that the Hessian diagonal approximation by AdaHessian method is far from the true scale (please see Fig. 1). Thus, we proposed a new method that firstly, approximate the Hessian diagonal in a well-scaled manner; and secondly, our method incorporates the adaptive learning rate idea on top of the well-scaled diagonal of Hessian to make it completely practical.
> Also, it is noteworthy to mention that there is $\textbf{NOT}$ any convergence guarantee for AdaHessian algorithm even for the fixed learning rate. Therefore, it is not clear how the idea of adaptive learning rate can be incorporated with AdaHessian algorithm.
> Fortunately, as we showed in our analysis and stated in the paper, our adaptive learning rate technique (with weighted norm) can benefit any method with a positive-definite preconditioning matrix and bounded eigenvalues. That said, if one can show that the eigenvalues of AdaHessian preconditioning matrix are positive and uniformly bounded above and below, then the analysis can be straightforward.

---

> ### Author Response · Authors · 2021-11-21
> **Are your concerns addressed?**
>
> Dear Reviewer doAo,
>
> We hope our responses have addressed your valuable comments and questions. We were especially wondering whether the questions and comments in the "$\textbf{Further Remarks}$" are answered.
>
> Please let us know if there are any remaining questions. We will be more than happy to hear what you think.
>
> Thank you!
>
> Authors

---

### Decision · Program_Chairs · 2022-01-20

**Decision:**

Accept (Poster)

**Comment:**

The paper presents a novel approximate second order optimization method for convex and nonconvex optimization problems. The search direction is obtained by preconditioning the gradient information with a diagonal approximation of the Hessian via Hutchinson's method and exponential averaging. The learning rate is updated using an estimate of the smoothness parameter.

The merit of the paper has to be evaluated from the theoretical and empirical point of view.

From the internal discussion, the reviewers agreed that the new algorithm is a mix a known methods, mainly present in AdaHessian, with a small tweak on the exponential average. Moreover, the theoretical guarantees do not seem to capture the empirical performance of the algorithm nor they provide any hint on how to set the algorithm's hyperparameters. For example, in Theorem 4.6 the optimal setting of $\beta_2$ is 1. That said, the most important theoretical contribution seems to lie in the fact that AdaHessian did not have any formal guarantee. Hence, this paper is the first one to show a formal guarantee this type of algorithms.

From the empirical point of view, the empirical evidence is very limited for the today standards in empirical machine learning papers. The reviewers and me do not actually believe that the proposed algorithm dominates the state-of-the-art optimization algorithms used in machine learning. However, in the internal discussion we agreed that the algorithm has still potential and it should be added to the pool of optimization algorithms people can try.

Overall, considering the paper in a holistic way, there seems to be enough novelty and results to be accepted at this conference.

That said, I would urge the authors to take into account reviewers comments (and I also add some personal ones here). In particular, a frank discussion of current theoretical analysis and empirical evaluation is needed.

Some specific comments:
- AdaGrad was proposed by two different groups at COLT 2010, so both papers should be cited. So, please add a citation to:
McMahan and Streeter. Adaptive bound optimization for online convex optimization. COLT 2010.
- Remark 4.7, second item: Neither Reddi et al.(2019) nor Duchi et al. (2011) *assume* bounded iterates, that must be proved not assumed. Instead, they explicitly project onto a domain that they assumed to be bounded.
- The convergence of the gradient to zero does not imply convergence to a critical point. To prove convergence to a critical point you should prove that the iterates converge, that in general is false even for lower bounded functions. Indeed, consider $f(x)=log(1+exp(-x))$, the iterates would actually diverge while the gradient still go to zero.